# Nonlinear machine learning pattern recognition and bacteria-metabolite multilayer network analysis of perturbed gastric microbiome

Claudio Durán [1,14], Sara Ciucci[1,14], Alessandra Palladini[1,2,3,14], Umer Z. Ijaz [4], Antonio G. Zippo [5], Francesco Paroni Sterbini [6], Luca Masucci [6], Giovanni Cammarota[7], Gianluca Ianiro[7], Pirjo Spuul[8], Michael Schroeder[9], Stephan W. Grill[9,10], Bryony N. Parsons [11], D. Mark Pritchard [11,12], Brunella Posteraro[6], Maurizio Sanguinetti [6], Giovanni Gasbarrini[7], Antonio Gasbarrini[7] & Carlo Vittorio Cannistraci [1,13 ✉]

The stomach is inhabited by diverse microbial communities, co-existing in a dynamic balance. Long-term use of drugs such as proton pump inhibitors (PPIs), or bacterial infection such as *Helicobacter pylori*, cause significant microbial alterations. Yet, studies revealing how the commensal bacteria re-organize, due to these perturbations of the gastric environment, are in early phase and rely principally on linear techniques for multivariate analysis. Here we disclose the importance of complementing linear dimensionality reduction techniques with nonlinear ones to unveil hidden patterns that remain unseen by linear embedding. Then, we prove the advantages to complete multivariate pattern analysis with differential network analysis, to reveal mechanisms of bacterial network re-organizations which emerge from perturbations induced by a medical treatment (PPIs) or an infectious state (*H. pylori*). Finally, we show how to build bacteria-metabolite multilayer networks that can deepen our understanding of the metabolite pathways significantly associated to the perturbed microbial communities.

[1] Biomedical Cybernetics Group, Biotechnology Center (BIOTEC), Center for Molecular and Cellular Bioengineering (CMCB), Center for Systems Biology Dresden (CSBD), Cluster of Excellence Physics of Life (PoL), Department of Physics, Technische Universität Dresden, Dresden, Germany. [2] Paul Langerhans Institute Dresden, Helmholtz Zentrum Munchen, Carl Gustav Carus, Technische Universität Dresden, Dresden, Germany. [3] German Center for Diabetes Research (DZD e.V.), Neuherberg, Germany. [4] Department of Infrastructure and Environment University of Glasgow, School of Engineering, Glasgow, UK. [5] Institute of Neuroscience, Consiglio Nazionale delle Ricerche, Milan, Italy. [6] Institute of Microbiology, Università Cattolica del Sacro Cuore, Rome, Italy. [7] Internal Medicine and Gastroenterology Unit, Università Cattolica del Sacro Cuore, Rome, Italy. [8] Department of Chemistry and Biotechnology, Division of Gene Technology, Tallinn University of Technology, Tallinn 12618, Estonia. [9] Biotechnology Center (BIOTEC), Center for Molecular and Cellular Bioengineering (CMCB), Technische Universität Dresden, Dresden, Germany. [10] Max Planck Institute of Molecular Cell Biology and Genetics, Dresden, Germany. [11] Department of Cellular and Molecular Physiology, Institute of Translational Medicine, University of Liverpool, Liverpool, UK. [12] Department of Gastroenterology, Royal Liverpool and Broadgreen University Hospitals NHS Trust, Liverpool, UK. [13] Center for Complex Network Intelligence (CCNI) at Tsinghua Laboratory of Brain and Intelligence (THBI), Department of Biomedical Engineering, Tsinghua University, Beijing, China. [14]These authors contributed equally: Claudio Durán, Sara Ciucci, Alessandra Palladini. ✉email: kalokagathos.agon@gmail.com

The gastric environment with its microbiota is the active gate that regulates access to the whole gastrointestinal tract, and therefore it has a remarkable impact on the correct functionality of the entire human organism. Recent studies have revealed that many orally administered drugs can perturb the elegant balance of the gastric microbiota[1,2]. However, not all of them cause permanent adverse effects and particular attention should be addressed to drugs that are frequently prescribed and administered for long periods. They can cause permanent unbalance of the gastric microbiota that might generate adverse side effects for the patient's health. Since the introduction of proton pump inhibitors (PPIs) into clinical practice more than 25 years ago, PPIs have become the mainstay in the treatment of gastric-acid-related diseases[3]. PPIs are potent agents that block acid secretion by gastric parietal cells by binding covalently to and inhibiting the hydrogen/potassium ($H^+/K^+$)-ATPases (or proton pumps), and additionally they can bind non-gastric $H^+/K^+$-ATPases, both on human cells and on bacteria and fungi, such as Helicobacter pylori (H. pylori)[4-6].

PPIs are drugs of first choice for peptic ulcers (PU) and their complications (e.g. bleeding), gastroesophageal reflux disease (GERD), nonsteroidal anti-inflammatory drug (NSAID)-induced gastrointestinal (GI) lesions, Zollinger-Ellison syndrome and dyspepsia[3,7,8]. In particular, dyspepsia is a common clinical problem characterized by symptoms (e.g. epigastric pain, burning, postprandial fullness or early satiation) originating from the gastroduodenal region[9]. The potent gastric-acid suppression drugs PPIs can treat the most frequent causes of dyspepsia including GERD, medication-induced gastritis, and PU, thus minimizing the need for costly and invasive testing, and moreover are currently recommended to eradicate H. pylori infection, in combination to antibiotics[7,9,10]. Nevertheless, some patients are resistant or partial responders to empiric PPI therapy, and continue to have dyspepsia[7].

Additionally, there is growing evidence that these medications are associated with increased rates of pharyngitis and upper and lower respiratory tract infections[11]. Their long-term over-utilization has been associated with potential adverse effects. For instance: the development of corpus predominant atrophic gastritis in H. pylori-positive patients (that is a precursor of gastric cancer), enteric infections (especially Clostridium difficile-associated diarrhoea), increased risk of fundic gland polyps, hypomagnesaemia and hypocalcaemia, osteoporosis and bone fractures, vitamin and mineral deficiency, pneumonia, acute interstitial nephritis and increased risk of drug–drug interactions, among others[7,12-15].

Consumption of such acid-suppressive medications has also been associated with changes in microbial composition and function of gut microbiota. More recent studies relying on amplicon-based metagenomic approaches, have shown that PPIs exert an effect on gastric, oropharyngeal, and lung microflora in children with a chronic cough[11], and have a significant impact on the gut microbiome in healthy subjects, with an increase of oral and pharyngeal bacteria and potential pathogenic bacteria[16,17]. Furthermore, another study by Tsuda et al.[18] revealed that PPIs influence the bacterial composition of saliva, gastric fluid and stool in a cohort of adult dyspeptic patients. However, this latter study highlights how the influence of PPI administration on the fecal and gastric luminal microbiota is still controversial and further investigation is required to understand the interaction between PPIs and non-H. pylori bacteria. Hence, this represents the first reason that motivates the present study.

In fact, by irreversibly blocking $H^+/K^+$-ATPases, PPIs inhibit gastric-acid secretion by gastric parietal cells, which results in a higher intragastric pH, meaning the microenvironment of this niche changes, hence allowing more bacteria to survive the gastric-acid barrier[4,5,16]. The use of PPIs and higher gastric pH were indeed correlated with the overgrowth of non-H. pylori bacterial microflora in the stomach of patients with gastric-reflux and PPIs were shown to aggravate gastritis because of co-infection with H. pylori and non-H. pylori bacterial species[4,14,19,20]. However, PPIs may also affect the gastro-intestinal microbiome through pH-independent mechanisms, by directly targeting the proton pumps of naturally occurring bacteria by binding P-type ATPases (e.g. H. pylori)[4,6].

Attempts to detect patterns of PPI-related gastrointestinal changes have been made in different studies[21,22] through linear multidimensional analysis techniques, such as Principal Component Analysis (PCA) and Multidimensional Scaling (MDS), also called Principal Coordinates Analysis (PCoA). Nevertheless, they failed to detect the effect of PPIs on gastric fluid samples[21], nor any significant PPI-related modification in esophageal[21] and gastric[22] tissue samples. This represents the second reason that motivates our investigation. Are these controversial results due to complex patterns that cannot be detected using linear analysis?

In this study, we show that: unlike linear approaches, Minimum Curvilinear Embedding (MCE)[23], which is a technique for nonlinear dimension reduction, discriminated both the esophageal and the gastric tissue microbial profiles of patients taking PPI medications from untreated ones when re-analyzing the data published in the abovementioned studies. This finding demonstrates the importance of routinely integrating the use of nonlinear multidimensional techniques into clinical metagenomic studies, since addressing nonlinearity could significantly modify the results and conclusions. Indeed, the absence of separation by means of linear transformations does not imply absence of separation in general, and nonlinear techniques could prove it, especially in complex datasets such as the ones generated in metagenomics 16 S rRNA. As a matter of fact, the high throughput profiling of bacteria is frequently used in clinical studies, thus posing a challenge to efficient information retrieval: understanding how microbial community structure affects health and disease can indeed contribute to better diagnosis, prevention and treatment of human pathologies[24].

The common practice in unsupervised dimension reduction data analysis is to consider only the first two (or three, less used) dimensions of mapping, and the goal is to visually explore the distribution of the samples and the incidence of significant patterns[25]. This type of analysis is advantageous to validate hypothesis or to generate new ones. In addition, this procedure is particularly useful in case of studies with small-size datasets[23], or for imbalance class samples, to obtain unbiased (the labels are not used) confirmation of the separation between groups of samples for which diversity is theorized or expected.

In addition, we will provide an analysis with two nonlinear algorithms for dimensionality reduction often used in literature, namely Isomap[26] and t-SNE[27,28]. These methods, although unsupervised, need hyperparameters optimization. Indeed, Isomap needs as input a parameter related to '$k$' number of neighbours to construct a network, whereas t-SNE needs the perplexity and number of dimensions (or components). Different values of these parameters may lead to different results, which represent a challenge in an unsupervised scenario where automatic and label-free selection of the best solution is wished. This is the reason why this study will focus mainly on parameter-free dimensionality reduction techniques, whereas Isomap and t-SNE results will be shortly considered for a specific dataset in the result section.

Here, we will specifically analyse the many aforementioned 16 S rRNA amplicons datasets to address the following pattern-recognition questions: (1) Is PPI treatment affecting change on the microbiota of esophageal and gastric tissues in dyspeptic patients, regardless of the initial pathological infection due to H.

*pylori*? (2) Is this PPI-induced change so dominant as to result in a discernible pattern in the first two dimensions of mapping by unsupervised dimension reduction? (3) Are linear techniques sufficient to bring out patterns in complex microbial data?

Furthermore, using differential network analysis we will address from the systems point of view these other questions: (4) How is PPI affecting the microbiota in the gastric environment in dyspeptic patients? (5) What is the effect of *H. pylori* infection on gastric mucosal microflora? Both factors (PPI treatment and *H. pylori* infection) can influence the composition of the gastric microbiota, and this further analysis will help to understand the general (overall) behaviour of the microbial ecosystem under these conditions and their impact on bacteria-associated metabolic pathways. Ultimately, this means that we will try to clarify and visualize via a bacteria-metabolite multilayer network representation how the bacteria-metabolite cooperative organization is systemically altered either by the use of this acid suppressant drug in the gastric environment under dyspepsia, or by *H. pylori* infection in the gastric mucosa.

## Results

To answer the five questions stated in the Introduction section, we analysed the abovementioned 16 S rRNA gene sequencing datasets with information on PPI consumption in dyspeptic patients, following the flowchart shown in Fig. 1. Our study is innovative at two different levels. At the more general 'methodological level', we introduce a computational data mining pipeline (Fig. 1), which explains how to overcome the limits of current multivariate analysis of small-size microbial data. At the more specific 'technical level', we propose solutions in each of the five steps that composes this pipeline: dimension reduction, clustering, PC-corr networks, multilayer bacteria-metabolite networks and metabolic network pathways analysis. In the dimension reduction section, we illustrate the benefits to apply minimum-curvilinear nonlinear machine learning methods for dimension reduction. In the clustering section, we propose MC-MCL, which represents the nonlinear version of Markov clustering. In the PC-corr section, we show how to extract valuable and robust information (that would otherwise be missed using standard procedure of analysis) across several (4 in total) small-size microbial datasets. In the fourth and fifth step we clarify how to enhance the biomedical interpretation with the aim to increase the impact of the findings on the scientific community.

It is important to underline that, in one of the three initially analysed datasets (in Paroni Sterbini et al.[22]), we have the additional information on positivity or negativity to *H. pylori* infection. A fourth dataset (Parsons et al.[29]) is used only for the validation of the PC-corr network results and it contains not only information on PPI consumption but also additional information on positivity or negativity to *H. pylori* infection.

Unsupervised approaches were chosen for dimension reduction, and clustering because supervised (constrained) methods have been shown to perform poorly on small datasets, as explained in the paper by Smialowski et al.[30] and the work by Zagar and colleagues[31].

First, we performed unsupervised dimension reduction, both linear and nonlinear (described in the 'Methods- PCA, MDS and LDA' and 'Methods- Minimum Curvilinear Embedding') and we focused on the first two dimensions of embedding as they are significantly related with the treatment/infection response (Supplementary Data 1). As we will show, linear techniques will fail to bring out the patterns in the microbial datasets related to PPI treatment. Instead, nonlinear dimension reduction will reveal the presence of hidden patterns related to PPI treatment. In particular, in the gastric biopsies dataset (Paroni Sterbini et al.[22]),

nonlinear dimension reduction will point out the evidence of PPI perturbation. Second, clustering algorithms were applied to the studied datasets to confirm that the hidden patterns detected by nonlinear dimension reduction are well posed. Furthermore, the PC-corr algorithm[32] is used to find the bacteria community (features) that make the difference between the patterns or groups, allowing our understanding of the PPI-induced and *H. pylori*-induced microbial perturbations. Finally, bacteria-metabolic networks are displayed addressing possible metabolic alterations produced by the respective perturbed bacteria.

**Data exploration and visualization: the reason for unsupervised dimension reduction.** The main reason to perform an unsupervised dimension reduction is to explore and visualize the most relevant sample patterns that should emerge in the first two dimensions of embedding (which represent the information of higher variability in the data) from the hidden multidimensional space of a dataset. The fact that the sample labels (if known) are not used for the data projection makes the analysis unsupervised. The advantage of performing an unsupervised analysis is both for data quality checking and to gather the main trends hidden in the data, independently from any hypothesis or knowledge available on the samples. This is particularly useful to discover the presence of interesting sub-groups inside the studied cohort or to detect the influence of confounding factors.

A final interesting advantage offered by unsupervised analysis is in small-size datasets, where the number of samples $n$ is significantly lower that the number of features $p$, a condition that unfortunately occurs in several metagenomic studies. When $n \ll p$ the application of supervised approaches can become problematic, because the supervised procedure of parameter learning can suffer from overfitting[23,30,33].

Below, we report some of the PCA major advantages and drawbacks, that were pinpointed in a recent study on multidimensional population genomics[34], and of other conventional dimensional reduction techniques employed for the analysis of metagenomic data.

PCA is time-efficient, parameter-free and straightforward to interpret, yet it strives to resolve structure in datasets with few samples and highly numerous features, which enclose nonlinear patterns. Therefore, PCA can occasionally fail to reveal differences among samples, even when differences are known a-priori, which means it can also miss represent hidden nonlinear relations among the samples in the feature space. For instance, see the illustration of the PCA two-dimension reduction mapping of the Tripartite-Swiss-Roll dataset in Supplementary Fig. 1b. PCA clearly fails to unfold and reveal the structure of the three separated groups of samples (Supplementary Fig. 1a). MDS, on the other hand, preserves the sample distances in a 2D space based on the calculation of a distance matrix. Nonetheless, given by the distance matrix, it can too strive to resolve nonlinear structures (Supplementary Fig. 1c,d).

**Gastric tissue dataset unsupervised analysis.** According to the questions formulated in our study, we are interested in an unsupervised approach to verify whether PPI drugs cause a major change in the gastric tissue microbiota of dyspeptic patients regardless of the initial pathological infection due to *H. pylori*[22].

In our first analysis, we focused on the Paroni Sterbini et al. dataset[22] and, to facilitate the visualization of the sample separations in the 2D reduced space, we assigned: red colour to untreated dyspeptic patients without *H. pylori* infection (H-); green colour to untreated dyspeptic patients with *H. pylori* infection (H+); and blue colour to patients treated with PPI regardless of their *H. pylori* infection (P). However, to help to

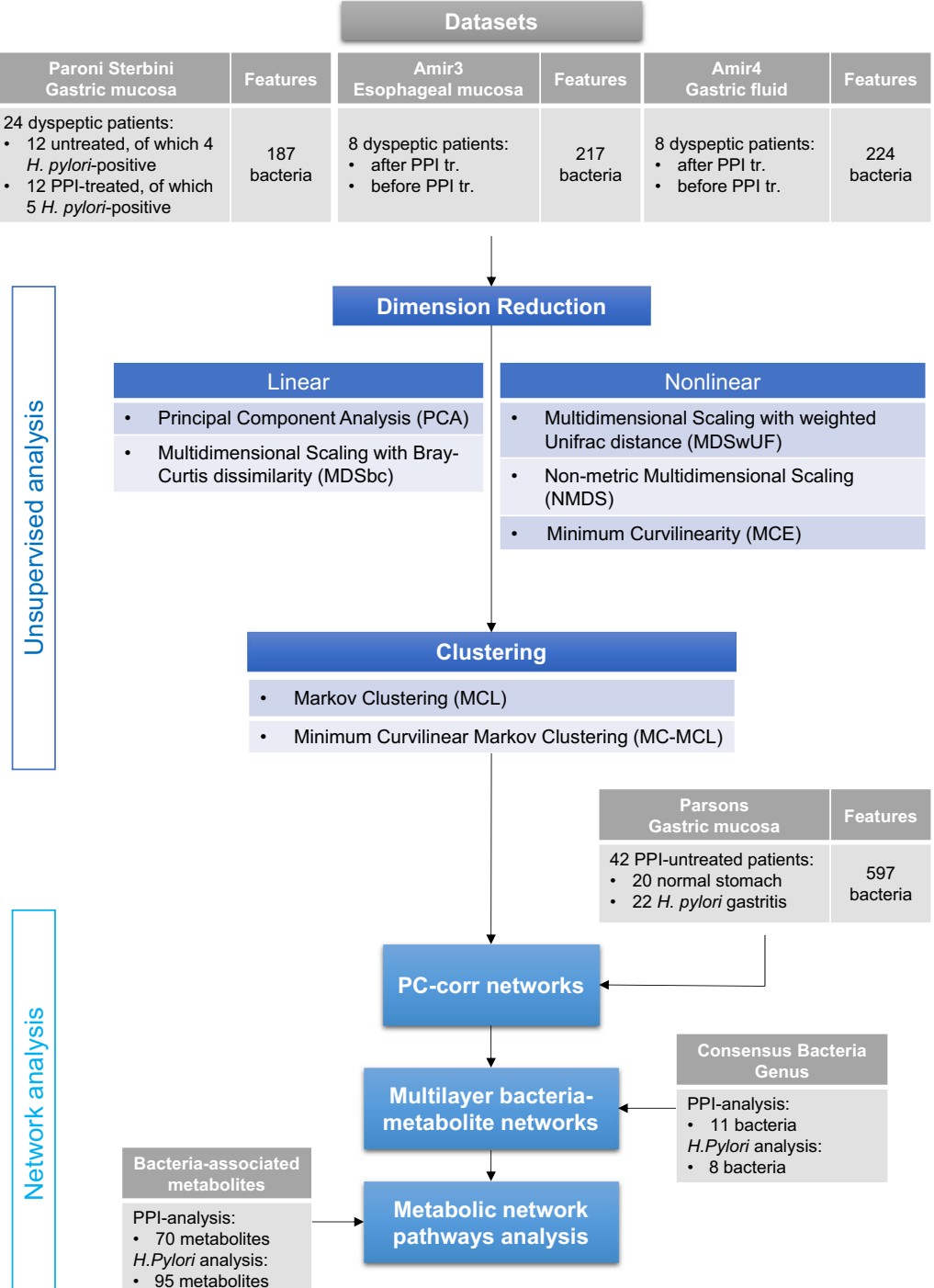

**Fig. 1 Flowchart of the data analysis.** To answer the five questions under investigation in our study, we implemented a workflow based on machine learning tools. Following the flowchart shown in the figure, we analysed three 16 S rRNA gene sequencing datasets with information on PPI use in dyspeptic patients; for one of the datasets (Paroni Sterbini et al.[22]), patients were also determined to be positive or negative to *H. pylori* infection. First, we performed unsupervised dimension reduction, both linear and nonlinear, in the first two dimensions of embedding. Nonlinear dimension reduction will show the presence of hidden patterns, in the form of sample groups. Secondly, nonlinear clustering was applied to confirm the well-possedeness of the hidden patterns found by nonlinear dimension reduction. Furthermore, our workflow ends with the network analysis. It starts with the use of the PC-corr algorithm, that reveals which combination of bacteria (features) are responsible for the identified differences between the groups of samples. A fourth dataset (Parsons et al.[29]) is used only for the validation of the PC-corr network results and it contains information of PPI treatment and *H. pylori* infection. From the consensus bacteria found in each PC-corr network, a bacteria-metabolite multilayer analysis that lastly end with the metabolite pathway enrichment analysis that introduces evidence to possible perturbed biological mechanisms.

detect also the effect of the *H. pylori* infection for the P patients, we reported the labels close to each sample with a '&H+' indicating the infection (P&H+) or a '&H−' indicating the absence of infection (P&H−). Finally, we also tested whether this separation into three main groups (H−, H+, P) is more truthful, from the metagenomics data standpoint, than the one in four groups (H−, H+, P&H−, P&H+).

Figure 2 shows the results of the multivariate techniques widely employed in metagenomic studies, PCA (Fig. 2a), MDSbc (Fig. 2b), MDSwUF (Fig. 2c) and NMDS (with Sammon Mapping) (Fig. 2d) (for more detail see the corresponding method section; the plots represents the best results based on PSI-PR in Supplementary Data 2), which could only differentiate the group of untreated *H. pylori*-positive samples (green dots) with respect to the group of untreated *H. pylori*-negative samples (red dots), and PPI-treated samples (blue dots), and no further separation is significantly detectable. Considering the PSI results, the values are high (Table 1 and Fig. 2) (evaluated in the 2D embedding space, for details see 'Procedure to evaluate the performance of the dimension reduction algorithms'). PCA (PSI-ROC = 0.85, PSI-PR = 0.91) and NMDS (PSI-ROC = 0.85, PSI-PR = 0.90) exhibit the highest PSI-ROC and PSI-PR values, followed by MDSwUF (PSI-ROC = 0.84, PSI-PR = 0.88) and MDSbc (PSI-ROC = 0.81, PSI-PR = 0.86). Indeed, in all the plots there is a visible trend of separation between PPI-treated (blue dots) and untreated (red and green dots) samples, but this is not sufficient to declare the presence of the complete separation, and a manifest 'crowding problem'[33] mixes the two cohorts together (blue and red dots). According to this output, the dataset appears to be strongly influenced by the presence of *H. pylori*, which is the predominant taxon (abundance > 50%, Supplementary Data 3, percent abundance sheet) in four of the untreated *H. pylori*-positive patients: where *H. pylori* is predominant, sample groups are quite close to one another and far from all the other samples in all four multivariate analyses (Fig. 2a–d). Thus, PCA and MDS mainly show us that these 16 S rRNA amplicons separate according to *H. pylori* abundance, and there is no treatment-related pattern.

Noncentred MCE (Fig. 2e, DCS normalization) was the best performing technique, with a PSI-ROC of 0.91 and PSI-PR of 0.96 (Table 1) (for details see Supplementary Data 2). It even outperforms the nonlinear methods NMDS (Sammon Mapping) and MDSwUF, since MCE is automatically able to unsupervisedly infer from data the underlying (hierarchical) phylogenetic relationship among the bacteria. MCE does not receive in input any phylogenetic information but directly infers it from the bacterial abundance of the dataset by performing a hierarchical embedding, as already shown in the study of Alanis-Lobato et al.[34] (see 'Supplementary Note 1—MCE to unsupervisedly infer and visualize phylogenetic (hierarchical) relations'). The gain in performance compared with the rest of the dimensionality reduction techniques is relevant.

Indeed, the PSI-ROC improvement from 0.85 (PCA and NMDS) to 0.91 is not trivial. We want to stress that in general offering an AUC-ROC result that is higher than 0.9 is considered relevant in all scientific literature. Furthermore, as suggested by Ammirati et al.[35], the same level of increase becomes more significant when being close to perfect segregation. For details, see the Supplementary Note 2—Relative performance improvement.

Furthermore, the MCE performance does not depend on its centring/noncentring, in fact the centred MCE version resolves the nonlinearity in the data too. Whereas, PCA regardless of being centred or noncentred does not resolve the nonlinearity in the data.

While MDS and PCA are confounded by the mixture of factors characterizing the samples and do not manage to resolve the

differences between treated and untreated samples, noncentred MCE is the only technique that visibly separates samples by ordering them along the second dimension into three groups, detecting a treatment-related structure in the data (Fig. 2f). This is plausible, because in any noncentred embedding the first dimension points towards the centre of the manifold[33], while the second dimension in the case of noncentred MCE represents the direction of higher topological nonlinear extension of the manifold. Interestingly, untreated *H. pylori*-negative samples (red dots, H−) gather in the upper tail of the samples' distribution, while treated samples (blue dots, P), both *H. pylori* test positive (P&H+) and negative (P&H−), are mixed and show no other internal discernible groups. Untreated *H. pylori*-positive samples (green samples, H+) gather at the bottom of the plot (Fig. 2e). Unlike the other approaches, noncentred MCE detects a treatment-related structure in the data and separates patients into three, not four, groups: PPI treated, untreated *H. pylori*-negative and untreated *H. pylori*-positive. This last group appears as a subgroup marginally discriminating from the PPI-treated group and the topology of the samples seems to suggest that PPI treatment modifies the gastric microbiota of *H. pylori*-negative patients with dyspeptic symptoms and gastric mucosa inflammation, shifting their gastric ecosystem in the same direction of PPI-treated *H. pylori*-positive patients. We speculate that the fact that PPI treatment and *H. pylori* infection determine the samples to gather in a similar position (i.e. out of the PPI-untreated/HP-negative group) in the noncentred MCE reduced space, indicates that both the PPI drugs and *H. pylori* induce an ecological change in the stomach, which might be driven by similar mechanisms. As a matter of fact, *H. pylori* can colonize the acidic lumen of the stomach thanks to its ability to hydrolyse urea into carbon dioxide ($CO_2$) and ammonia ($NH_3$)[36], thus increasing the intragastric pH. On the other hand, PPIs obtain the same result through the inhibition of acid secretion in gastric parietal cells, which blocks $H^+/K^+$ -ATPases. Both processes are therefore shifting the gastric environment towards an alkaline condition. Thus, MCE provides an ordering of the groups along the second dimension that is related to pH increment (from H− to P&H+).

Furthermore, we contrast MCE performance on this challenging dataset versus two baseline algorithms for nonlinear dimension reduction: t-SNE and Isomap. The results are shown in Supplementary Fig. 2. t-SNE (PSI-ROC: 0.90, PSI-PR: 0.94) and Isomap (PSI-ROC: 0.87, PSI-PR: 0.94) performances are lower than MCE performances, displaying difficulty to resolve the difference between treated and untreated samples, mostly for the cases of treated patients (blue points) and untreated patients without *H. Pylori* infection (red points). For more details see the Supplementary Note 3—Nonlinear dimension reduction techniques t-SNE and Isomap.

On the other hand, and similarly to the Paroni Sterbini et al. microbial dataset, all dimensionality reduction techniques were compared in two artificial scenarios (for more details see the Supplementary Note 4—Artificial datasets). These analyses confirm anew that certain methods will fail to uncover hidden nonlinear structures whilst methods tailored for this purpose will not (for more details see the Supplementary Note 5—Dimensionality Reduction analysis in artificial datasets).

For the Paroni Sterbini dataset, we also performed a supervised linear approach for dimension reduction, LDA (Supplementary Fig-. 3), yet the cross-validation test showed that this constrained technique could re-assign samples to their groups with 54% of error (ldaCVErr in Supplementary Data 4), confirming its statistical invalidity for the small-size dataset problem.

Moreover, the clustering algorithms MCL and MC-MCL, that is the minimum-curvilinear version of MCL were applied to the Paroni Sterbini et al. dataset and the best results (highest

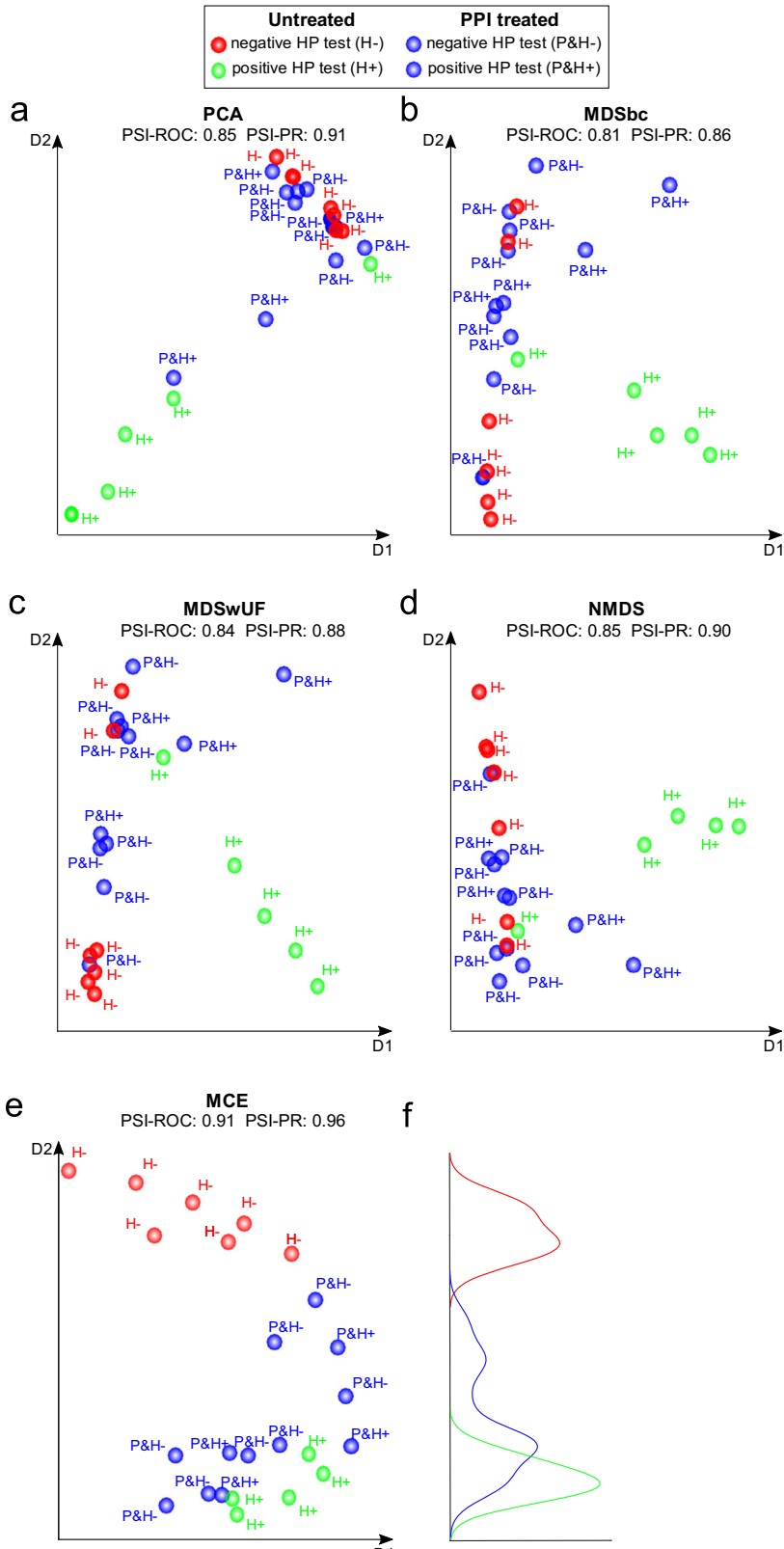

**Fig. 2 Dimension reduction techniques applied to the Paroni Sterbini dataset.** The plots represent the best dimension reduction results based on PSI-PR projection-based separability index (PSI) for the three different labels (P-treated, untreated H+ and untreated H−), evaluated in the 2D embedding space. Moreover, also the average values of all pairwise PSI-ROC are reported as overall estimators of separation between the groups in the 2D reduced space. **a** PCA; **b** MDS with Bray-Curtis dissimilarity (MDSbc); **c** MDS with weighted UniFrac distance (MDSwUF); **d** nonmetric MDS with Sammon Mapping (NMDS); **e** MCE. Blue dots represent PPI-treated samples, while red and green dots are the untreated samples which resulted either negative (red) or positive (green) to the H. pylori test (histological observation and urease test). **f** The curves in three different colours (red, blue and green) highlight the different distributions of the three groups on the second dimension for the MCE plot (**e**).

**Table 1 Results of unsupervised dimension reduction techniques on the real dataset.**

| Method | Paroni Sterbini | Trust | Amir3 | Trust | Amir4 | Trust | mean |
|---|---|---|---|---|---|---|---|
| PSI-ROC | | | | | | | |
| HD | 0.88 | 0.0036 | 0.95 | 0.0009 | 0.98 | 0.0009 | 0.94 |
| MDSwUF | 0.84 | 0.0089 | 1.00 | 0.0009 | 0.88 | 0.0329 | 0.90 |
| MCE | 0.91 | 0.0036 | 0.88 | 0.0329 | 0.91 | 0.0009 | 0.90 |
| PCA | 0.85 | 0.0063 | 0.91 | 0.0009 | 0.86 | 0.0169 | 0.87 |
| MDStyc | 0.84 | 0.0076 | 0.88 | 0.0009 | 0.84 | 0.0249 | 0.85 |
| NMDS | 0.85 | 0.0036 | 0.86 | 0.0169 | 0.84 | 0.0089 | 0.85 |
| MDSbc | 0.81 | 0.0183 | 0.86 | 0.0089 | 0.84 | 0.0189 | 0.84 |
| PSI-PR | | | | | | | |
| HD | 0.94 | 0.0009 | 0.96 | 0.0009 | 0.99 | 0.0009 | 0.96 |
| MDSwUF | 0.88 | 0.0036 | 1.00 | 0.0009 | 0.90 | 0.0089 | 0.93 |
| MCE | 0.96 | 0.0009 | 0.89 | 0.0089 | 0.92 | 0.0039 | 0.92 |
| PCA | 0.91 | 0.0039 | 0.90 | 0.0009 | 0.88 | 0.0089 | 0.90 |
| MDStyc | 0.88 | 0.0116 | 0.90 | 0.0009 | 0.88 | 0.0089 | 0.89 |
| MDSbc | 0.86 | 0.0116 | 0.89 | 0.0009 | 0.90 | 0.0009 | 0.88 |
| NMDS | 0.90 | 0.0036 | 0.87 | 0.0089 | 0.87 | 0.0009 | 0.88 |

Best results of unsupervised dimension reduction techniques according to the PSI indices for sample separation in the space of the first two dimensions of embedding. HD (no dimension reduction) represents the reference results to see how good the separability present in the high-dimensional space is preserved by dimension reduction techniques. Results are ordered from the best (top) to the worst (bottom) method. For the Paroni Sterbini dataset, we show the results for three different labels (PPI treated, untreated H+ and untreated H−). For the Amir datasets, the PSI measures were computed for two groups, identified by the presence or absence of PPI treatment. For each PSI value, a respective trustworthiness was calculated.
All PSI-ROC and PSI-PR values can be found in Supplementary Data 2.
*HD* high dimension, *MCE* Minimum Curvilinear Embedding, *MDSbc* Multidimensional Scaling with Bray-Curtis dissimilarity, *MDSwUF* Multidimensional Scaling with weighted UniFrac distance, *NMDS* Nonmetric Multidimensional Scaling, *MDStyc* Multidimensional Scaling with Theta-YC distance, *PCA* Principal Component Analysis, *PSI-ROC* Projection Separability Index measured by Area Under the Curve, *PSI-PR* Projection Separability Index measured by Area Under the Precision Recall, *Trust* trustworthiness.

accuracies) are shown in Table 3 (for more details see the methods' sections 'From Markov Clustering to Minimum Curvilinear Markov Clustering' and 'Procedure to evaluate the performance of clustering algorithms'). MC-MCL performs better than the MCL (both for three and four clusters), even if their accuracies are not remarkably high, confirming that difficulties in pattern-recognition arise also from the presence of three clusters in the high-dimensional space. In addition, the hypothesis of three clusters seems more congruous than four clusters, because both MC-MCL and MCL decrease their accuracies in detecting four clusters.

While MC-MCL represents the minimum-curvilinear version of MCL, MCE is the minimum-curvilinear version of PCA, particularly valuable for small sample size datasets. The principle behind them is MC[23], that suggests that curvilinear (nonlinear) distances between samples may be estimated as pairwise distances over their Minimum Spanning Tree (MST) (constructed according to a selected distance). In fact, as explained in ref. [37], to approximate nonlinear (curvilinear) distances between the points of the manifold it is not necessary to reconstruct the nearest-neighbour graph. Indeed, a greedy routing process (that exploits a norm, for instance Euclidean) between the points in the multidimensional space is enough to efficiently navigate the hidden network that approximates the manifold in the multi-dimensional space. And a preferable greedy routing strategy, at the basis of MC-kernel, is the minimum spanning tree (MST).

Overall, we can conclude that both MCE in dimensionality reduction and MC-MCL in clustering perform better than the respective non-MC-based versions, and this result confirms the presence of nonlinear complexity in this dataset, generated by a three-body interaction (presence of three clusters). In addition, when considering correlation-based distances, they do not react to the presence of compositionality, since pairwise correlations are computed between samples. Compositionality instead is a problem that arises when the correlations are computed between OTUs (features) from metagenomics abundance data (which are normalized by diving each OTU count to the total sum of counts in the sample[38,39]), which yields unreliable results due to dependency of microbial relative abundances.

To discover the nonlinearity of the data, a pairwise group analysis applied to the Paroni Sterbini gastric biopsy dataset (Supplementary Fig. 4) and the Tripartite-Swiss-Roll (Supplementary Fig. 5) revealed that the nonlinearity was indeed associated to the presence of three hidden clusters. For more information see the Supplementary Note 6—Origin of the Paroni Sterbini data nonlinearity.

In conclusion, the results confirm that linear techniques, even if supervised like LDA, are not able to resolve the differences in the data due to the presence of nonlinear complexity generated by the three-body interaction (H−, H+ and P). Once the complexity is reduced to a two-body interaction, the problem tends to vanish and PCA can detect significant differences between the groups, as shown by the PCA pairwise comparisons (Supplementary Fig. 4 and Supplementary Fig. 5). However, the presence or absence of *H. Pylori* does not seems to heavily affect patients with PPI treatment (Supplementary Fig. 6)

Hence, the results of unsupervised analysis on Paroni Sterbini et al. dataset show that PPI treatment causes a major change in gastric mucosal communities of dyspeptic patients, regardless of the initial pathological infection due to *H. pylori*.

**Comparison of unsupervised analysis in three gastroesophageal datasets.** We compared the performance of unsupervised analysis (dimensional reduction and clustering) in the Paroni Sterbini dataset[22] (gastric biopsies) and two additional datasets by Amir and colleagues[21], that investigated the PPI influence on the eso-phageal microbiota (Amir3) and gastric fluid (Amir4).

Table 1 shows the best results in performance of unsupervised dimension reduction (PCA, MDSwUF, MDSbc, NMDS, MCE, for details see 'Methods—PCA, MDS and LDA' and 'Methods—Minimum Curvilienar Embedding') according to PSI based on AUC and AUPR, on the three different datasets (for more details on the PSI see 'Methods—Procedure to evaluate the performance of the dimension reduction algorithms'). Just the space of the first two dimensions of embedding were here used since they are the ones related with the treatment/infection-related structures (Supplementary Data 1). The mean performance across all datasets is shown in the last column of the Table 1 for each method. The corresponding ranked performance for each

**Table 2 Ranked performance of unsupervised dimension reduction techniques on the real datasets.**

| Method | Paroni Sterbini | Amir3 | Amir4 | mean |
|---|---|---|---|---|
| PSI-ROC | | | | |
| HD | 2 | 2 | 1 | 1.67 |
| MCE | 1 | 4 | 2 | 2.33 |
| MDSwUF | 5 | 1 | 3 | 3.00 |
| PCA | 3 | 3 | 4 | 3.33 |
| NMDS | 3 | 6 | 5 | 4.67 |
| MDStyc | 5 | 4 | 5 | 4.67 |
| MDSbc | 7 | 6 | 5 | 6.00 |
| PSI-PR | | | | |
| HD | 2 | 2 | 1 | 1.67 |
| MCE | 1 | 5 | 2 | 2.67 |
| MDSwUF | 5 | 1 | 3 | 3.00 |
| PCA | 3 | 3 | 5 | 3.67 |
| MDStyc | 5 | 3 | 5 | 4.33 |
| MDSbc | 7 | 5 | 3 | 5.00 |
| NMDS | 4 | 7 | 7 | 6.00 |

The table shows the ranked performance of unsupervised dimension reduction techniques according to the PSI indices for sample separation (PSI-ROC and PSI-PR) in the space of the first two dimensions of embedding, for the three studied datasets (Paroni Sterbini, Amir3 and Amir4). Each rank is related to the results obtained in Table 1. The results are ordered by the mean performance (fourth column) from the best (top) to the worst (bottom) method. *HD* high dimension, *MCE* Minimum Curvilinear Embedding, *MDSbc* Multidimensional Scaling with Bray-Curtis dissimilarity, *MDSwUF* Multidimensional Scaling with weighted UniFrac distance, *NMDS* Nonmetric Multidimensional Scaling, *MDStyc* Multidimensional Scaling with Theta-YC distance, *PCA* Principal Component Analysis, *PSI-ROC* Projection Separability Index measured by Area Under the Curve, *PSI-PR* Projection Separability Index measured by Area Under the Precision Recall.

**Table 3 Results of clustering on the real dataset.**

| Method | Paroni Sterbini | Amir3 | Amir4 | mean |
|---|---|---|---|---|
| Accuracy | | | | |
| MC-MCL | 0.71 (0.58) | 0.81 | 0.75 | 0.76 |
| MCL | 0.67 (0.63) | 0.69 | 0.75 | 0.70 |

Best results of clustering (highest accuracies, regardless of the normalization and type of correlation) MCL and MC-MCL, in each of the three studied datasets (Paroni Sterbini, Amir3 and Amir4), and the mean performance (mean of the highest accuracies) across all the datasets.
For Paroni Sterbini dataset, we show the results for three clusters (PPI treated, untreated H+ and untreated H−) and in brackets the results for four clusters (P&H+, P&H−, untreated H+ and untreated H−). Instead, for Amir datasets, the accuracies were computed for two groups, identified according to the presence or absence of PPI treatment. All accuracies can be found in Supplementary Data 5.
*MCL* Markov Clustering, *MC-MCL* Minimum Curvilinear Markov Clustering.

what is the best method, we considered an evaluation based on ranking (Table 2). It is important to note that MCE was the dimension reduction approach that ranked first in performance across all the datasets, followed by MDSwUF (Table 2). Hence, the results of sample separability suggest the presence of hidden patterns that emerge by applying nonlinear dimension reduction techniques like MCE and MDSwUF.

Then, clustering algorithms, MCL and its Minimum Curvilinear version (for more information see 'Methods—From Markov Clustering to Minimum Curvilinear Markov Clustering'), were used to confirm the well-possedeness of the hidden patterns that were recognized by nonlinear dimension reduction. The best results as highest accuracies in each dataset and the mean performance across all the datasets are exhibited in Table 3. As already discussed in the previous section, the minimum-curvilinear version of MCL (MC-MCL, acc = 0.71) outperforms the MCL clustering algorithm (acc = 0.67) in the Paroni Sterbini dataset, confirming the presence of underlying nonlinear complexity in the data. However, the accuracy doesn't reach high values, because of the difficulty in pattern recognition generated by the three-body problem in the HD space. Curiously, the accuracies for four clusters (H−, H+, P&H−, P&H+) drop to 0.58 for MC-MCL and to 0.63 for MCL, supporting the hypothesis that three clusters are more congruous than four clusters. Notably in Amir3, MC-MCL attains high clustering accuracy (acc = 0.81), compared to MCL (acc = 0.69). This is the dataset for which, surprisingly, Amir and collaborators did not find significant changes in the esophageal tissue microbiota following PPI treatment, using classical MDS unsupervised multivariate method with unweighted UniFrac distance[21]. Instead, in the gastric fluid dataset (Amir4), MC-MCL and MCL got the same accuracy of 0.75, where a significant separation of samples according to PPI consumption was already proved in the original article[21].

Other normalizations besides DRS, DCS and log transformation could potentially improve the performance of the unsupervised analysis. Therefore, we analysed the data with two regularly employed normalizations in microbiome studies: (1) applying a Variance-Stabilization Transformation (VST) (results in Supplementary Tables 3–5 and Supplementary Data 9 and 10) and (2) rarefying the OTU table (results in Supplementary Tables 6–8 and Supplementary Data 11 and 12). As discussed in the Supplementary Note 7—Normalizations applied in microbiome studies, these normalizations tend to linearize the data structure but at the cost of information loss.

**Network analysis clarifies the effect of PPI treatment on the gastric microbiota.** Five major phyla have been detected in the normal gastric microbiota: *Firmicutes*, *Bacteroidetes* and

method, based PSI-ROC and PSI-PR, is presented instead in Table 2. For the Paroni Sterbini dataset, we show the results for three different labels (untreated H−, untreated H+ and PPI treated). For Amir datasets, the PSIs were computed for two groups, identified by the presence or absence of PPI treatment. The PSIs were also applied to the data in the original high-dimensional (HD) space, as a reference to see how good the unsupervised dimension reduction approaches are in preserving the group separability in the HD. Moreover, the PSI-ROC and PSI-PR best results with trustworthiness and standard error on the real datasets, when applying leave-one-out-cross-validation (LOOCV), are shown in Supplementary Table 2.

For the Paroni Sterbini dataset, the PSI evaluation in the first two dimensions of embedding identifies MCE as the best dimension reduction technique that is able to preserve the group separability in the HD space. Surprisingly, MCE (presented in Fig. 2e, PSI-ROC = 0.91, PSI-PR = 0.96) outdoes HD in sample separation in three groups (for HD, PSI-ROC = 0.88, PSI-PR = 0.94). Similarly, in Amir4, MCE (PSI-ROC = 0.91, PSI-PR = 0.92) succeeds in preserving the separability of the original HD space (in HD, PSI-ROC = 0.98, PSI-PR = 0.99), better than the other dimension reduction methods. Finally, dimension reduction analysis on the Amir3 dataset shows that esophageal biopsies were significantly different before and after PPI treatment, as shown by MDSwUF results (PSI-ROC = 1=PSI-PR), that surpass the PSI-ROC and PSI-PR values in HD space (PSI-ROC = 0.95, PSI-PR = 0.96). Markedly, MDSwUF reaches a value of AUPR and AUC of 1, meaning perfect classification of the samples.

Overall, when averaging across all datasets, the two metrics based on PSI-ROC and PSI-PR pointed out that MDSwUF (PSI-ROC = 0.90, PSI-PR = 0.93) gave the best results of separability compared to HD (PSI-ROC = 0.94, PSI-PR = 0.96), followed by MCE with close results (PSI-ROC = 0.90, PSI-PR = 0.92). Then PCA is the third best result (PSI-ROC = 0.87, PSI-PR = 0.90), followed by MDStyc, NMDS and MDSbc. However, to conclude

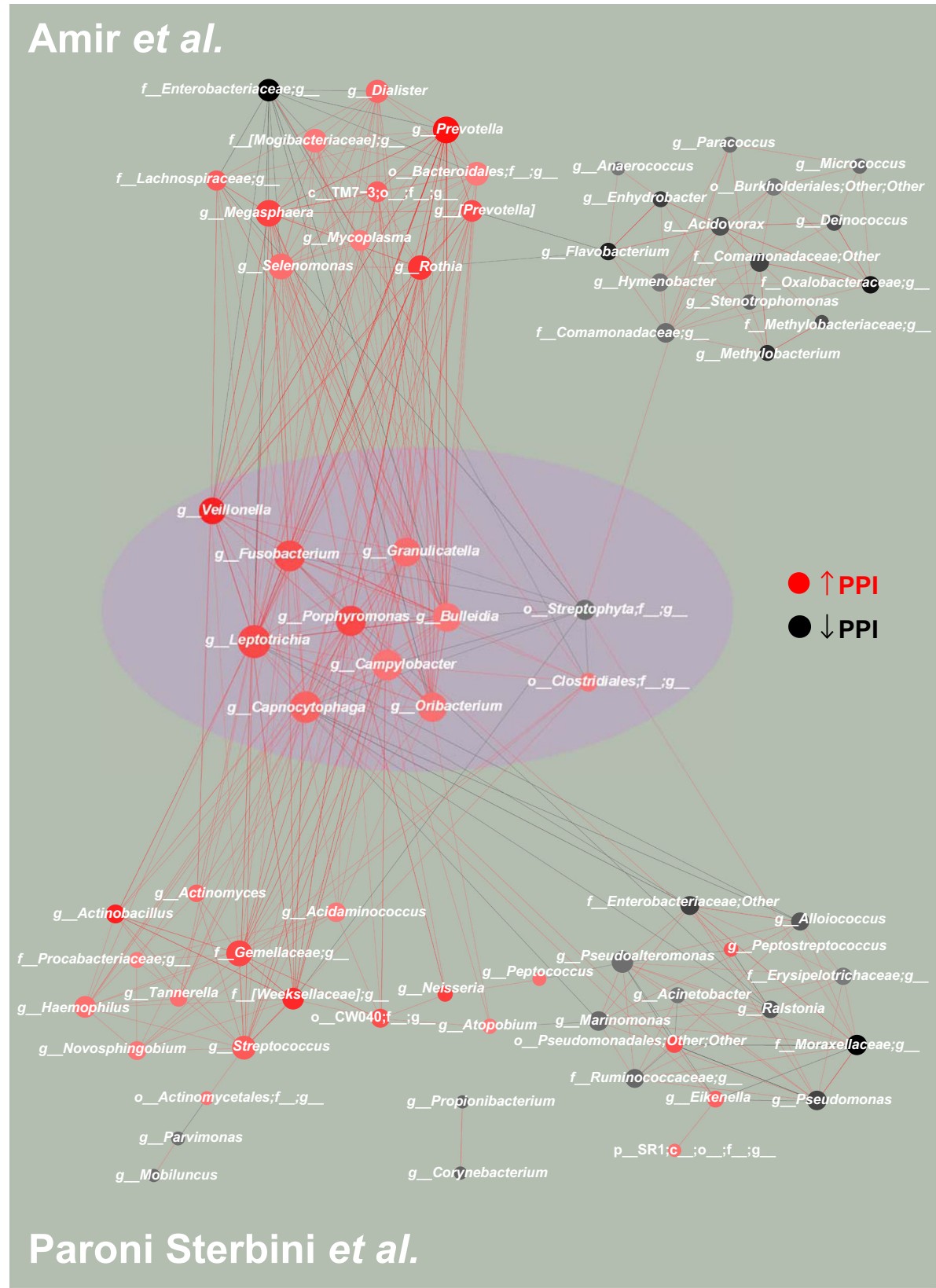

Actinobacteria dominate the gastric fluid samples, while Fusobacteria and Proteobacteria are the most abundant phyla in gastric mucosal samples[1].

However, the composition and abundance of gastric microbiota may be affected by many factors, such as dietary habits, H. pylori infection, diseases and drugs, including PPIs[1].

Yet, although recent studies have highlighted the potential of these antacid drugs to affect the gastric microbiota, more knowledge needs to be gained about the association between PPI usage and the non-H. pylori bacteria in the stomach.

Since we wanted to investigate the effect of PPI intake on gastric microbiota in dyspepsia, we analysed: Amir4 for gastric

**Fig. 3 PC-corr method to unveil how PPI is affecting the microbiota in gastric environment in dyspeptic patients.** (Middle panel) To investigate the effect of PPIs on the gastric microbiota in dyspeptic patients, we constructed the conserved PC-corr network at 0.5 cut-off, by merging the PC-corr networks obtained from the gastric mucosa (Paroni Sterbini et al.[22]) and the gastric fluid (Amir et al.[21]). To do so, we firstly considered the union of the two PC-corr networks obtained from the gastric tissue dataset and then we intersected it with the PC-corr network from the gastric fluid dataset. All the bacteria spotted in the conserved PC-corr network (violet circle) were found increased with PPI use. In both the two studied datasets, red nodes indicate bacteria whose abundance is increased with PPI treatment, while black nodes indicate bacteria with lower abundance following treatment with this acid suppressing medication. The common bacteria that showed an opposite trend in the two datasets, i.e. microbial abundance increased in one dataset and decreased in the other dataset, were removed from the network. (Top panel) The top panel shows the obtained Amir4's network, not in common with the Paroni Sterbini's network. The module on the left side (except Enterobacteriaceae) include bacteria more abundant following PPI treatment in Amir4's data, while the module on the right (and Enterobacteriacea) is composed of decreased bacteria in abundance under PPI therapy in Amir4's data. (Bottom panel) The bottom panel represents the part of Paroni Sterbini's network (union of the two PC-corr network), that is not shared with Amir4's one. As in the top and middle panels, the colour of the nodes represents if the bacteria display higher (red nodes) or lower abundance (black nodes) in PPI-treated samples of Paroni Sterbini's dataset.

fluid microbiota[21] and Paroni Sterbini et al. dataset[22] for gastric mucosal microflora, in the latter case restricting to PPI-treated *H. pylori*-negative (P&H−) and untreated *H. pylori*-negative patients (H−). In both studies, the samples from dyspeptic patients were analysed using the same next-generation sequencing technologies for direct sequencing of 16 S rRNA gene amplicons, 454 Pyrosequencing.

For this purpose, we employed PC-corr algorithm, that was discussed in the Methods section named: 'PC-corr network'. In brief, PC-corr discloses the discriminative network of features that are associated to a sample separation along a principal component direction. Hence, we expect that the PC-corr network of bacteria will offer a view on how the community of bacteria respond to PPI-treatment perturbation in the gastric niche (environment), in dyspeptic patients. For more explanation about the usage of PC-corr on the microbiome data please see the Supplementary Note 8—From nonlinear data to linear analysis.

In Amir4 (gastric fluid), PCA revealed that gastric fluid samples were separated into two groups according to PPI treatment along PC2 and their difference is significant (*p*-value < 0.01) (Supplementary Fig. 7). Hence, we built the PC-corr network[32] using the loadings of PC2 at cut-off 0.5 (Supplementary Fig. 8).

Similarly for the Paroni Sterbini dataset (gastric mucosa), PCA (Supplementary Fig. 9) could (significantly or close to significance) separate PPI-treated *H. pylori*-negative patients from untreated *H. pylori*-negative patients along PC2 and PC15 (*p*-value along PC2 = 0.014, *p*-value along PC15 = 0.054). Therefore we built the PC-corr network for both PC2 and PC15 discriminating dimension using 0.5 cut-off (Supplementary Fig. 10, panel a and b).

Subsequently, to investigate how PPI is affecting the microbiota in the gastric environment, we considered the conserved PC-corr network as an indication of bacteria behavior robustness. It is obtained as the union of the two PC-corr networks (obtained for PC2 and PC15) derived from the Paroni Sterbini gastric mucosa dataset intersected with the PC-corr network derived from the Amir4 gastric fluid dataset. The resulting conserved network displays the bacteria with same trend in the two datasets, i.e. either increased or decreased abundance for patients with PPI treatment, respectively in red and black colour, as emphasized by the violet circle at the centre of Fig. 3. Figure 4 is the same as Fig. 3 but here the nodes are coloured according to phylum-level taxonomy. The conserved network which arises at the overlap between the two PC-corr networks (union of Paroni Sterbini networks intersected with the Amir4 network) is statistically significant (*p*-value = 1.00E-04), as a result of the statistical test based on trying to obtain the same conserved network by random resampling the bacteria in the two networks (Supplementary Fig. 11), implying the difficulty of generating this intersection

simply at random (since this intersection lies to the right of the critical value at the 0.05 level in the distribution of overlap). This is an important result because it confirms the robustness of the detected conserved network as a microbiota signature perturbed by PPI treatment. The top and bottom panels in Figs. 3 and 4 show instead the remaining part of Amir4's network (top panel) and of Paroni Sterbini's network (bottom panel) that are not in the intersection, and therefore might be more specific for the gastric fluid and mucosa, respectively. The PPI-perturbed conserved network is characterized by a main interconnected module with nine bacteria of four different phyla (*Bacteroidetes, Fusobacteria, Proteobacteria, Firmicutes*) that are positively associated (red edges) and by two single bacteria order without interactions (*Streptophyta, Clostridiales*), all being increased following PPI treatment, except *Streptophyta* that is instead decreased with PPI treatment (Figs. 3 and 4). Note that a mix between genera, phyla and order of bacteria can be found in the networks. The reason behind it is the availability of detail information regarding different bacteria. Some of the spotted bacteria (*Veillonella, Clostridiales, Campylobacter*) were already observed in previous studies. The genus *Veillonella* was found increased in relation to PPI use[16] in the gut microbiome and has been associated with increased susceptibility to *Clostridium difficile* infection[40]. These Gram-negative anaerobic cocci with lactate fermenting abilities are abundant in the human microbiome and are normally found in the intestines and oral mucosa of humans[41]. Interestingly, they favour nitrite accumulation in the stomach during nitrate reduction, promoting a carcinogenic effect[1]. In addition, the order *Clostridiales*, that is associated to *Clostridium difficile* infection, was also seen significantly changed in the gastrointestinal tract; however; Freedberg et al.[4] found it significantly decreased during PPI use, in contrast to our results. PPIs use also increases the risk of other enteric infections, apart from *C. difficile* infection, such as campylobacteriosis, as reported in[42,43]. Moreover, half of the bacteria present in the network normally colonize the human oral cavity. Indeed, it is the main purpose of PPI treatment to increase the stomach pH, and the higher pH of treated patients is known to favour the growth of bacteria that usually reside in the mouth and esophagus and are not adapted to survive the normal gastric acidity[6,20]. Among genera usually reported as part of the normal microbiota of the gastrointestinal tract, only *Veillonella* is found regularly at other sites, like the mouth[44]. *Leptotrichia* species mostly colonize the oral cavity and they were isolated from various human infections, suggesting that they are emerging human pathogens[45,46]. *Oribacterium* also inhabits the mouth, besides the upper respiratory tract[47]. *Prevotella* is a genus of Gram-negative bacteria that tend to colonize the human gut, mouth and vagina, and may cause infections, mostly observed in the oral cavity (odontogenic infections)[46]. *Porphyromonas* has been found by[48] as part of the

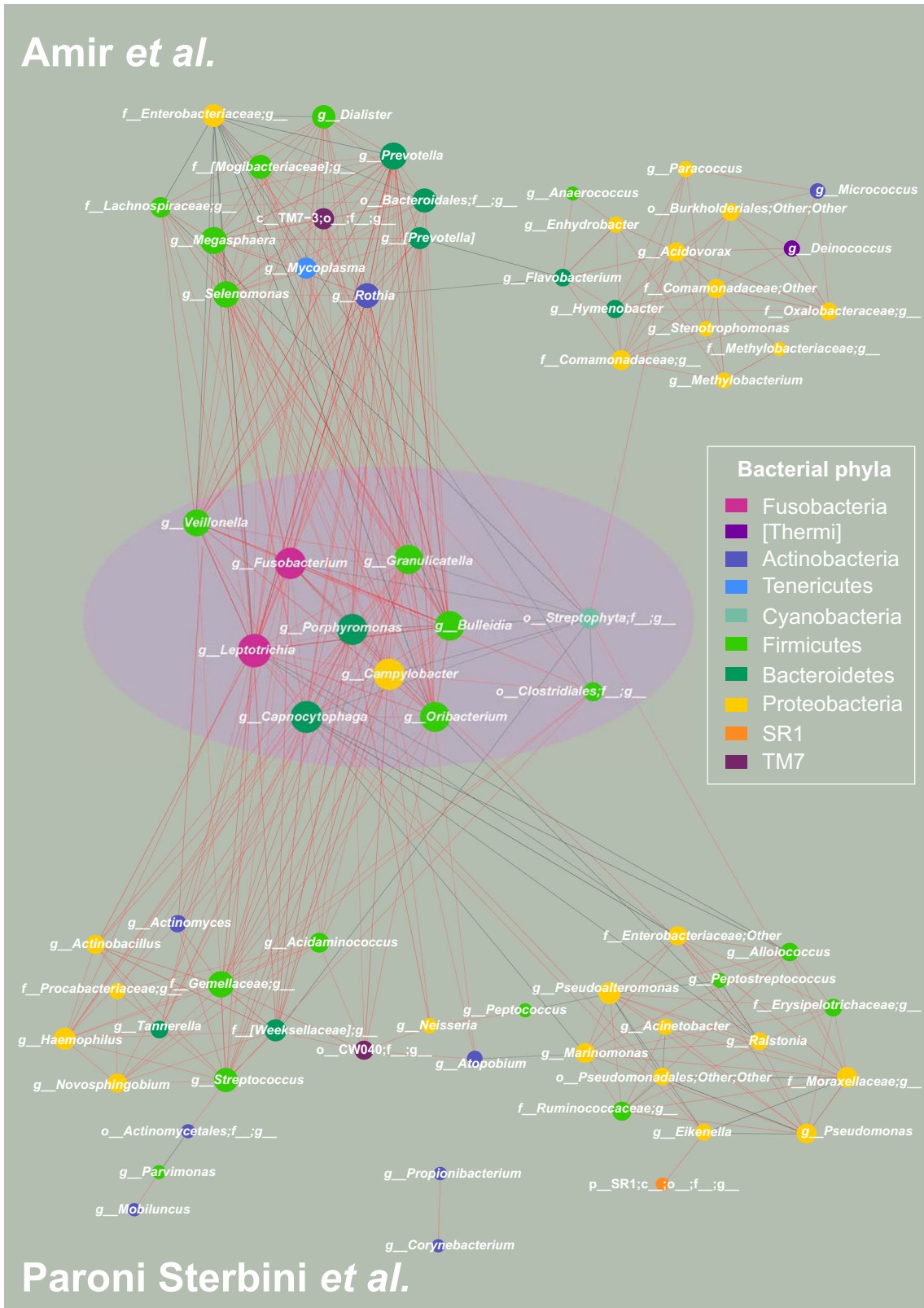

salivary microbiome. Both *Prevotella* and *Porphyromonas* contribute to the formation of abscesses and soft tissue infections in various part of the body and they can cause infections, including periodontal and endodontal diseases[49]. *Capnocytophaga* are inhabitants of the oral cavity too, and these opportunistic pathogens can cause infections (both in immunocompromised and immunocompetent hosts), the severity of which depend on the immune status of the host[50,51]. As well, *Granulicatella* are Gram-positive cocci normally found in the oral microbiota and are uncommon causes of infections, nevertheless they can cause infections, including bloodstream infection and infective endocarditis[52]. Besides, the genus *Fusobacterium* inhabits the mucosal membranes of humans and all its species are parasites of humans[53], and some species are found in the oral cavity. The

**Fig. 4 PC-corr networks to unveil how PPI is affecting the microbiota in gastric environment in dyspeptic patients, coloured according to phylum-level taxonomy.** To investigate the effect of PPIs on the gastric microbiota in dyspeptic patients, we constructed the conserved PC-corr network at 0.5 cut-off, by merging the PC-corr networks obtained from the gastric mucosa (Paroni Sterbini et al.[22]) and the gastric fluid (Amir et al.[21]). To do so, we first considered the union of the two PC-corr networks obtained from the gastric tissue dataset and then we intersected it with the PC-corr network from the gastric fluid dataset. All the bacteria spotted in the conserved PC-corr network (violet circle) were found increased with PPI use. (Top panel) The top panel shows the obtained Amir4's network, not in common with the Paroni Sterbini's network. The module on the left side (except Enterobacteriaceae) include bacteria more abundant following PPI treatment in Amir4's data, while the module on the right (and Enterobacteriacea) is composed of decreased bacteria in abundance under PPI therapy in Amir4's data. (Bottom panel) The bottom panel represents the part of Paroni Sterbini's network (union of the two PC-corr network), that is not shared with Amir4's one. As in the top and middle panels, nodes are coloured according to bacterial phylum level.

remaining bacteria (*Campylobacter, Bulleidia*) do not belong to the oral microbiota[49]. The genus *Campylobacter* was increased in relation to PPI use and the increased abundance of these Gram-negative bacteria has the potential to cause diseases and infections in humans (most commonly diarrhoea). Due to the induced increase of pH, PPI is hypothesised to facilitate gastrointestinal infections and a study by Brophy et al.[54] reported an increased risk of *Campylobacter* infection following PPI therapy. Moreover Campylobacteriosis, mostly caused by eating undercooked foods derived from poultry or other warm-blooded animals or contact with contaminated water or ice[55], has been shown by the Dutch National Institute for Public Health and the Environment to noticeably increase in incidence when PPI use grows[42].

Altogether, PC-corr approach was applied on gastric fluid and gastric mucosal datasets (in the latter case, excluding the samples positive to *H. pylori* infection) to investigate how PPI is affecting the gastric microbiota (both gastric fluid and gastric mucosal microbiota), because of PC-corr's ability to pinpoint the combination of bacteria that play a major role in the discrimination of the samples, in this case according to PPI intake. The PC-corr conserved network identified eleven genera and order of bacteria, which belong to the phyla (*Bacteroidetes, Fusobacteria, Proteobacteria, Firmicutes*) commonly found in the stomach which, with exception of *Streptophyta*, demonstrated increased abundance following PPI treatment. Mostly all the found bacteria were not reported in previous studies, except *Veillonella, Clostridiales* and *Campylobacter*, but they were found as inhabitants of the oral cavity and/or possible cause of infections and diseases in humans. Hence, and in concordance to previous studies[6,20], these results point out that PPI treatment, by increasing the intragastric pH, favours the growth of bacteria that usually reside in the mouth and survive through the harsh acidic conditions of the stomach. Furthermore, the results suggest that PPI-associated increase of some bacterial populations may lead to infections and diseases or increase susceptibility for other bacterial infections (like *Veilonella*) or promote a carcinogenic effect (like *Veilonella*). Previous studies have highlighted that PPI intake is associated with decreased bacterial richness[16,18,56,57], increased risk of enteric and other infections (e.g. caused by *Salmonella, Clostridium difficile, Shigella, Listeria*)[17,58], increase in the abundance of oral and upper GI tract commensals and potential pathogenic bacteria (e.g. *Enterococcus, Streptococcus, Staphylococcus* and *Escherichia coli*)[16,17] in the gut microbiota. Nevertheless, our analysis by means of PC-corr does not spot single bacteria perturbed in the gastric environment by PPI treatment, but a community of bacteria is altered in abundance by PPIs and their inter-specific bacterial interactions in the gastric niche.

Therefore our study will ground the basis for further investigations that could better clarify the effect of PPI treatment on the human gastric microbiota and additionally verify the identified altered bacteria, as PPIs may have possible side effects, including increased risks of different infections and diseases.

**Network analysis clarifies the effect of *Helicobacter pylori* infection on gastric mucosal microbiota.** The stomach was long thought sparsely colonized by bacteria due to the gastric microbicidal acidic barrier (pH < 4.0)[59]. This view dramatically changed with the discovery of the Gram-negative bacterium *H. pylori* in the 1980's by Warren and Marshall[60], that is a carcinogenic bacterial pathogen infecting the stomach of more than one-half of the world's human population. This human pathogen is able to survive in the highly acidic environment within the stomach by producing cytoplasmic urease that, by catalysing the hydrolysis of urea into $CO_2$ and $NH_4$, produces a neutralizing ammonia cloud around it[19,61,62]. However, most *H. pylori* avoid the acidic environment of the gastric lumen by swimming towards the mucosal cell surface (using their polar flagella and chemotaxis mechanisms) and may adhere and invade the gastric mucosal epithelial cells[63,64]. Hence, it doesn't represent a dominant species in gastric fluid microbiota[65], but was found to generally reside in the gastric mucosae[5,63,66].

Persistent (chronic) infection with this Gram-negative bacterium induces changes in gastric physiology and immunology, e.g. reduced gastric acidity and parietal cell mass, perturbed nutrient availability, local innate immune responses[67,68], that most probably induces shift in gastric microbiota composition[67]. Although *H. pylori* colonization usually persists in the human stomach for many decades without adverse effects, the infection of this bacteria is associated with increased risk for several diseases, including peptic ulcers, chronic gastritis, mucosa-associated lymphoid tissue lymphoma, gastric adenocarcinoma[69,70] and dyspepsia[71,72]. The potential alterations induced by the *H. pylori* can in turn lead to dysbiosis and may cause aberrant proinflammatory immune responses[73], susceptibility to bacterial pathogens and increased risk of gastric disease, including cancer[1,74]. However, the effect of *H. pylori* infection on overall composition of gastric microbiota at genus level and the bacterial interplay in presence of this widespread human infection remain unclear.

Similar to the PPI treatment network analysis in the previous section, in order to investigate the influence of *H. pylori* infection on the gastric mucosal microbiota by means of PC-corr, we analysed: (1) Paroni Sterbini et al.[22] considering only PPI-untreated dyspeptic patients, either infected (H+) or not by *H. pylori* (H−); (2) Parsons et al.[29] restricting to PPI-untreated patients from: (i) normal stomach group with no evidence of *H. pylori* infection; (ii) *H. pylori* gastritis group with evidence of *H. pylori* infection. Even though the same technology is important for a comparative study, unfortunately in the literature there was no such data available like Paroni Sterbini's one, that is 16 S rRNA gene pyrosequencing data (derived from gastric mucosal microflora in dyspeptic untreated patients either positive or negative for *H. pylori*). Despite this, the two studied datasets, obtained with two different next-generation sequencing technologies for direct sequencing of 16 S rRNA gene amplicons (454 Pyrosequencing for Paroni Sterbini et al. and Illumina MiSeq for Parsons et al.)[75], both contain community profiling of gastric

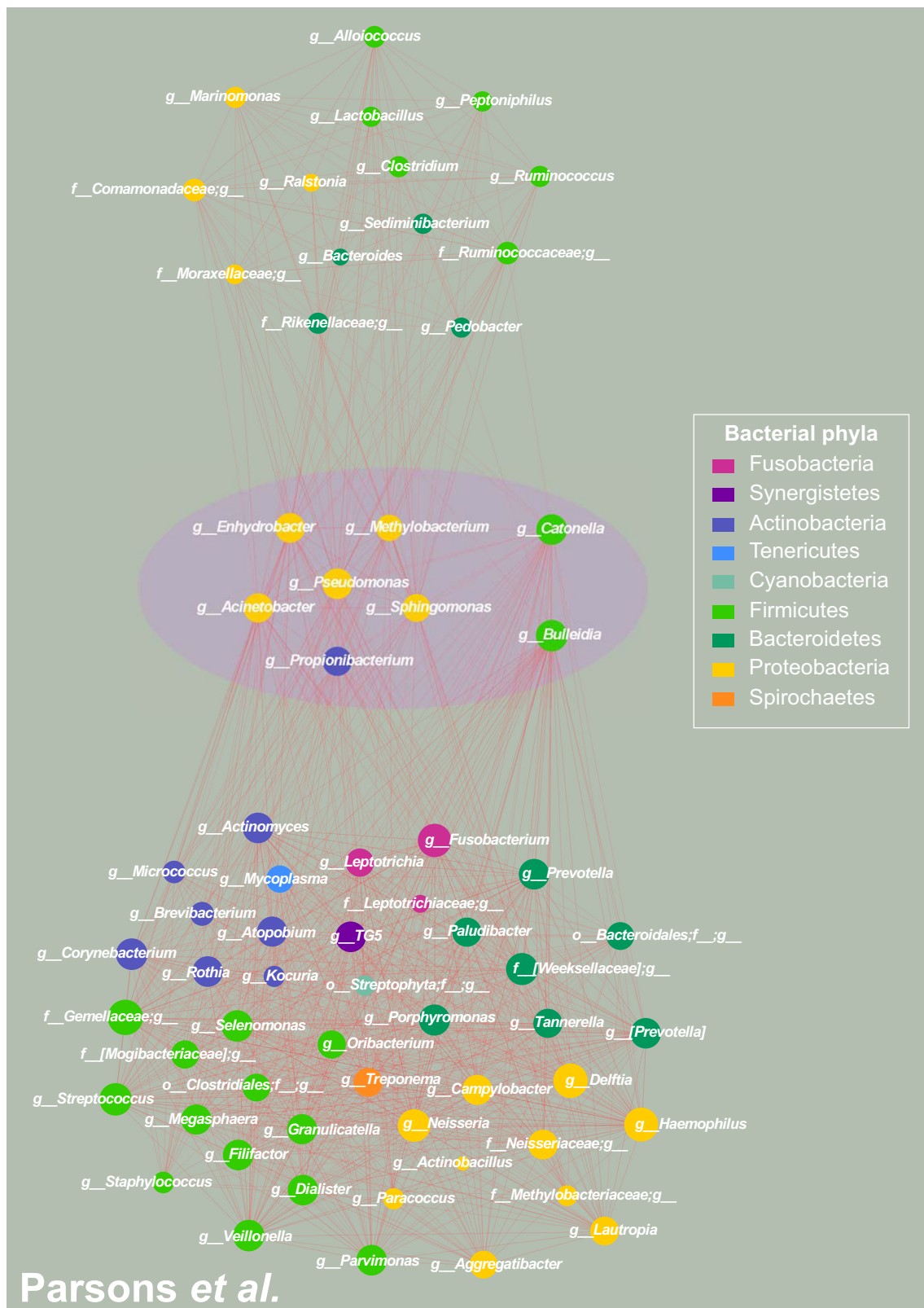

Parsons *et al.*

mucosa-associated microbiota in PPI-untreated *H. pylori*-negative and -positive subjects. However, for the sake of clarity, we have to specify a difference: while in Paroni Sterbini's dataset the gastric mucosal biopsy specimens were collected from patients with dyspepsia, this is not the case for Parsons's data.

To enhance the understanding of the *H. pylori*-triggered microbial perturbation in this ecological niche, we employed again PC-corr algorithm (for details see 'Methods-PC-corr network'). The analysis of the 16 S rRNA sequencing data was restricted only the overlapping OTUs, excluding *Helicobacter* because our goal is to investigate its impact on the rest of the microbial network.

In Paroni Sterbini's dataset, since PCA could significantly separate gastric mucosal biopsy samples of PPI-untreated patients

**Fig. 5 PC-corr network to investigate the effect of *H. pylori* infection on the gastric mucosal microbiota, coloured according to phylum-level taxonomy.**
(Middle panel) To investigate the effect of *H. pylori* infection on the gastric mucosal microbiota, we constructed the conserved PC-corr network at 0.5 cut-off, by intersecting the PC-corr networks obtained from Paroni Sterbini et al.[22] and Parsons et al.[29] dataset. All the bacteria spotted in the conserved PC-corr network (violet circle) were found decreased in abundance with *H. pylori* infection. The common bacteria that showed an opposite trend in the two datasets, i.e. microbial abundance increased in one dataset and decreased in the other dataset, were removed from the network. (Top panel) The top panel show the obtained Paroni Sterbini's network, not in common with the Parsons's network. It contains all bacteria whose abundance is decreased in *H.pylori*-positive patients in Paroni Sterbini et al. dataset. (Bottom panel) The bottom panel represent the part of Parsons's network that is not shared with Paroni Sterbini's one. As in the top and middle panels, it includes bacterial communities decreased in *H. pylori*-infected patients.

according to *H. pylori* positivity (*p*-value = 0.01) along PC2 (Supplementary Fig. 12), the PC-corr network was constructed from PC2 loadings at 0.5 cut-off (Supplementary Fig. 13). Similarly, for Parsons' dataset, since PCA (Supplementary Fig. 14) could significantly separate patients from the normal stomach group with no evidence of *H. pylori* infection and PPI-untreated (Control) from *H. pylori* gastritis group positive to *H. pylori* infection and not using PPIs (HPGas) along PC1 (*p*-value along PC1 < 0.01), the PC-corr network was constructed from this discriminating dimension at 0.5 cut-off (Supplementary Fig. 15). The obtained microbial differential networks (Fig. 5, coloured according to phylum level) pinpointed, from the system point of view, the bacteria affected by *H. pylori* infection in the gastric mucosa, that are precisely bacteria whose abundance is decreased in *H. pylori*-positive patients. A presumable explanation of this trend is already pointed out in literature, where the presence of *H. pylori* leads to a reduced gastric microbial diversity[76–78]. Nevertheless, in some cases the diversity increases again, because of diverse factors that allow survival and colonization of bacteria in the stomach[1,79]. Then, the preserved network of gastric mucosa microbiota was constructed by intersecting the two PC-corr networks obtained from Paroni Sterbini's and Parsons's dataset. Figure 5, middle panel, shows the conserved network (violet circle), which presents the common bacteria coloured according to phylum level and their associations. The spotted bacteria display decreased abundance with *H. pylori* infection (i.e. increased in *H. pylori-negative* subjects) in both the two 16 S rRNA gene sequencing data. By performing a statistical test based on random resampling of the bacteria in the two networks, we verified that the shown bacterial conserved network is statistically significant and difficult to be generated at random (*p*-value = 1.00e-04), because getting this intersection at random is very rare (Supplementary Fig. 16). The top and bottom panels in Fig. 5 show instead the remaining part of Paroni Sterbini's network (top panel) and of Parsons's network (bottom panel) that are not in the intersection. At the genus level, a study by Klymiuk et al.[80] identified *Actinomyces, Granulicatella, Veillonella, Fusobacterium, Neisseria, Helicobacter, Streptococcus* and *Prevotella* as significantly different between the *H. pylori*-positive and *H. pylori*-negative gastric samples. These bacteria do not emerge in the conserved network, while they all (except *Neisseria*) appear altered (decreased) during *H. pylori* infection in the study by Parsons and colleagues (present in the bottom panel of Fig. 5).

Our analysis pinpoints a conserved network from two independent 16 S rRNA gene sequencing data that reveals microbial communities altered by *H. pylori* infection and their interactions in the gastric mucosa. It revealed a main core of six associated bacteria (with positive association, red edges) and two single nodes without any interaction with the main module, from three different phyla (*Proteobacteria, Firmicutes, Actinobacteria*) all resulting decreased in *H. pylori*-infected subjects (that is increased in noninfected subjects). The decreased abundance of the phyla *Firmicutes* and *Actinobacteria* in *H. pylori*-positive patients with respect to *H. pylori*-negative subjects was already shown in a previous study by Maldonado-Contreras et al.[81]. In

addition, other studies have demonstrated an increased colonization of *Proteobacteria* in *H. pylori*-positive patients[81,82], while the obtained conserved PC-corr network shows that the bacteria from this phylum are instead decreased in those individuals. Among the spotted bacteria, *Methylobacterium* is a genus of facultative methylotrophic bacteria that are commonly found in diverse natural environments (such as leaf surfaces, soil, dust and fresh water) and in hospital environment due to contaminated tap water. *Methylobacterium* species can cause health care-associated infections (mainly catheter infection), especially in immunocompromised patients[83]. In addition, *Sphingomonas* plays a role in human health, as some of the sphingomonads (in particular *Sphingomonas paucimobilis*) are the cause of a range of mostly nosocomial, non-life-threatening infections. *Sphinhomonas* species are widely spread in nature, having been isolated from many sources, from water habitats to clinical settings[84].

*Pseudomonas*, due to its great metabolic versatility, can also colonize different types of niches[85], including soil and water, in addition to plant and animal associations, and includes pathogenic species in humans[86]. *Acinetobacter* species are instead common, free-living saprophytes found in soil, water, sewage and foods and are ubiquitous organisms in hospitals. They have been increasingly identified as a key source of infection in debilitated patients in hospitals, due to their rapid development of resistance to antimicrobials[87]. In particular, one species, *Acinetobacter lwoffi*, can trigger gastritis, apart from *H. pylori*[88]. *Propionibacterium*, so named for their unique ability to synthesize propionic acid by using unusual transcarboxylase enzymes[89], are primarily facultative pathogens and commensals of humans, living on the skin, while other members are widely employed for synthetizing vitamin $B_{12}$, tetrapyrrole compounds, and propionic acid, as well as used as probiotics[90]. *Catonella* is another node in the network and this bacterial genus is obligative anaerobic, non-spore-forming and nonmotile, with one known species (*Catonella morbi*) from the human gingival crevice[91,92], that has been associated with periodontitis[91] and endocarditis[93]. Besides, the bacterial genus *Enhydrobacter* so far contains a single species, *Enhydrobacter aerosaccus*, a Gram-negative nonmotile bacterium that is both oxidase and catalase positive and shows gas vacuoles[94,95]. *Bulleidia*, a Gram-positive, non-spore-forming, anaerobic and nonmotile genus, has one known species too (*Bulleidia extructa*)[96].

In conclusion, by means of the PC-corr approach, we determined the combination of bacteria responsible for the difference between *H. pylori*-positive and *H. pylori*-negative gastric mucosa of untreated patients and their microbe-microbe interactions. All the bacteria, both in the conserved network and not, were decreased in *H. pylori*-infected individuals (i.e. increased in *H. pylori*-negative group). *H. pylori*, like acid suppressing medications (for the treatment of dyspepsia), alters the population structure of the gastric and intestinal microbiota[97] and regularly, this bacterium constitutes most of the gastric microbiota[79], literally depleting bacterial biodiversity. Moreover, most of the identified bacteria represent bacteria of potential health concern, as agents of diseases and infections.

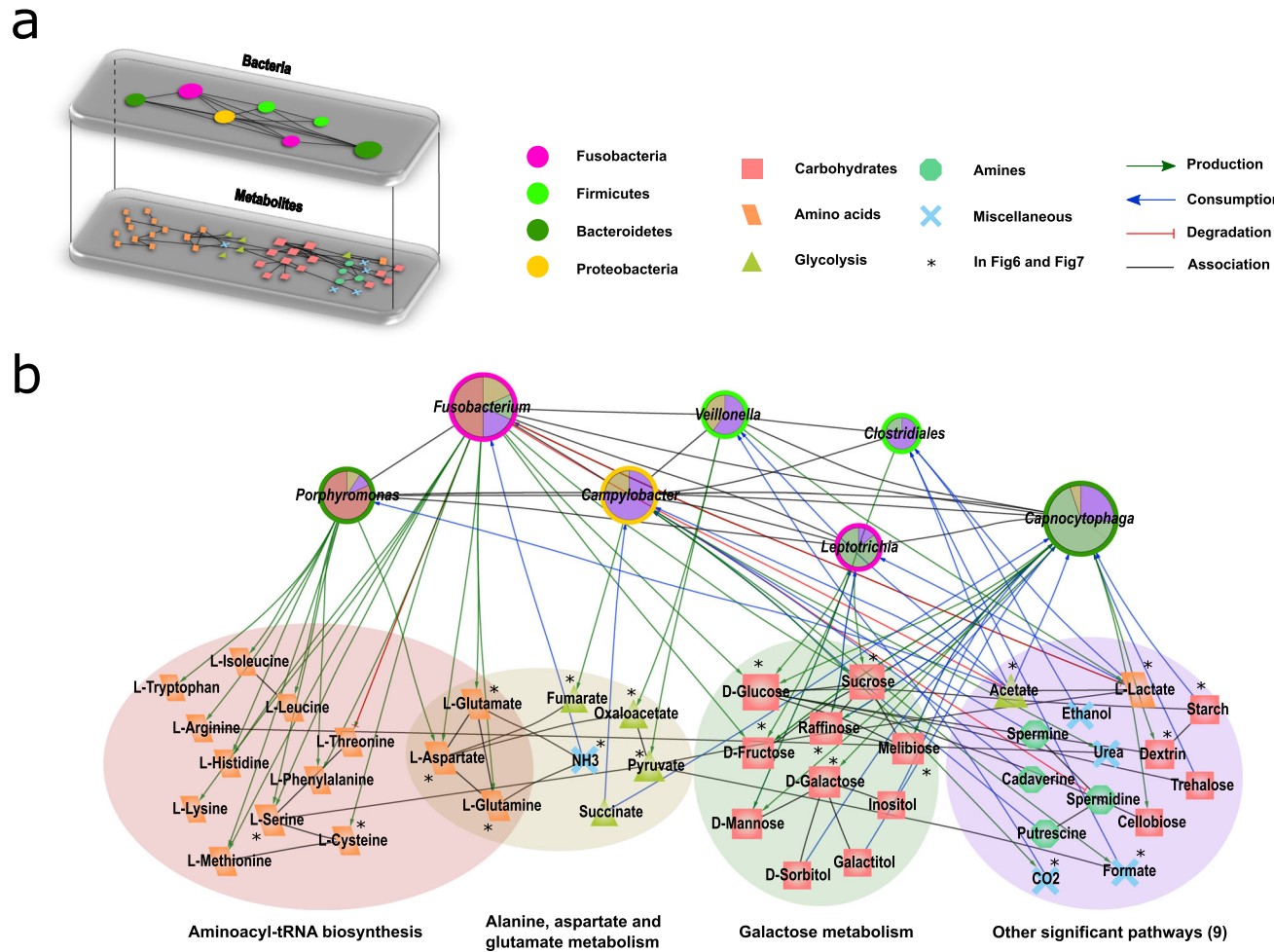

**Fig. 6 PPI-affected bacteria-metabolite network in gastric environment of dyspeptic patients. a** Multilayer (bacteria-metabolite) network representation: the first layer is derived from Fig. 4 and represents the consensus network (confirmed in two datasets: gastric mucosa from Paroni Sterbini et al.[22] and gastric fluid from Amir et al.[21]) with PPI-affected bacteria nodes that present information on metabolite interaction in ref. [148]. The second layer represents the network whose nodes are the metabolites in ref. [148] interacting with the bacteria network in the first layer; different node shapes and colours refer to different metabolite classes (carbohydrates, amino acids, glycolysis, amines, miscellaneous). **b** In depth visualization of the bacteria-metabolite network interactions. The metabolites are grouped according to their involvement in significant pathways. For discernibility, the metabolites are arranged according to three significant pathways ($p < 0.05$ after Benjamini correction as result of a metabolite pathway enrichment analysis by a one-sided test) and a fourth group that encloses altogether nodes associated to other significant pathways (please refer to the method section: Bacteria-metabolite multilayer network construction and metabolite pathway analysis); note that only metabolites present in significant pathways are here displayed. For more information, please refer to Supplementary Fig. 17 and Supplementary Data 6. The bacteria node stroke color is associated to the phyla information as in Fig. 4, whereas the different colours in the inner fill are associated to the different pathways and their extent is proportional to the number of metabolites that the bacterium connects with in the different displayed pathways.

**Bacteria-metabolite multilayer network analysis associates possible metabolic pathways perturbations**. The relation between bacteria and metabolites is fundamental both to deepen the understanding of mechanisms associated to diseases dysfunction and drugs action, and to foster their biomedical interpretation[98–100]. For this reason, we made a quantum leap in our investigation from bacteria to metabolites and we built two bacteria-metabolite multilayer networks: one (Fig. 6) was derived from the PPI-affected bacteria network in Fig. 4, the other (Fig. 7) was derived from the *H. pylori*-affected bacteria network in Fig. 5. The methodological procedure to build those multilayer networks is provided in the Methods (see section: Bacteria-metabolite multilayer network construction). Remarkably, by applying metabolic pathway enrichment analysis, we found that the metabolite layer of the PPI-affected (Fig. 6b) and *H. pylori*-affected (Fig. 7b) networks contain metabolites significantly involved ($p < 0.05$ after Benjamini correction) in important

pathways (see full list in Supplementary Data 6 and 7) associated with obesity[101], symptomatic atherosclerosis[102], functional dyspepsia[103], gestational diabetes mellitus[104] Wilson's disease[105], among others. To simplify the visualization and interpretation (for the methodology of selection see Method section: Bacteria-metabolite multilayer network construction) we displayed the three most significant and relevant pathways in both PPI-affected (Fig. 6b) and *H. pylori*-affected (Fig. 7b) networks. Interestingly, the bacteria *Porphyromonas* and *Fusobacterium* are highly contributing for possible perturbations on the Aminoacyl-tRNA biosynthesis pathway for alterations produced by PPI (Fig. 6b), while *Methylobacterium* does it on the *H. Pylori* infection side (Fig. 7b). Besides, N-Acetylneuraminic acid (Supplementary Fig. 17) is a sialic acid that has been associated also with pathogenic enteric bacteria[106,107] and tumors[108]. Overproduction of nitrites and nitrates by the observed anaerobic bacteria have been already noticed in diverse parts of the gastrointestinal tract in

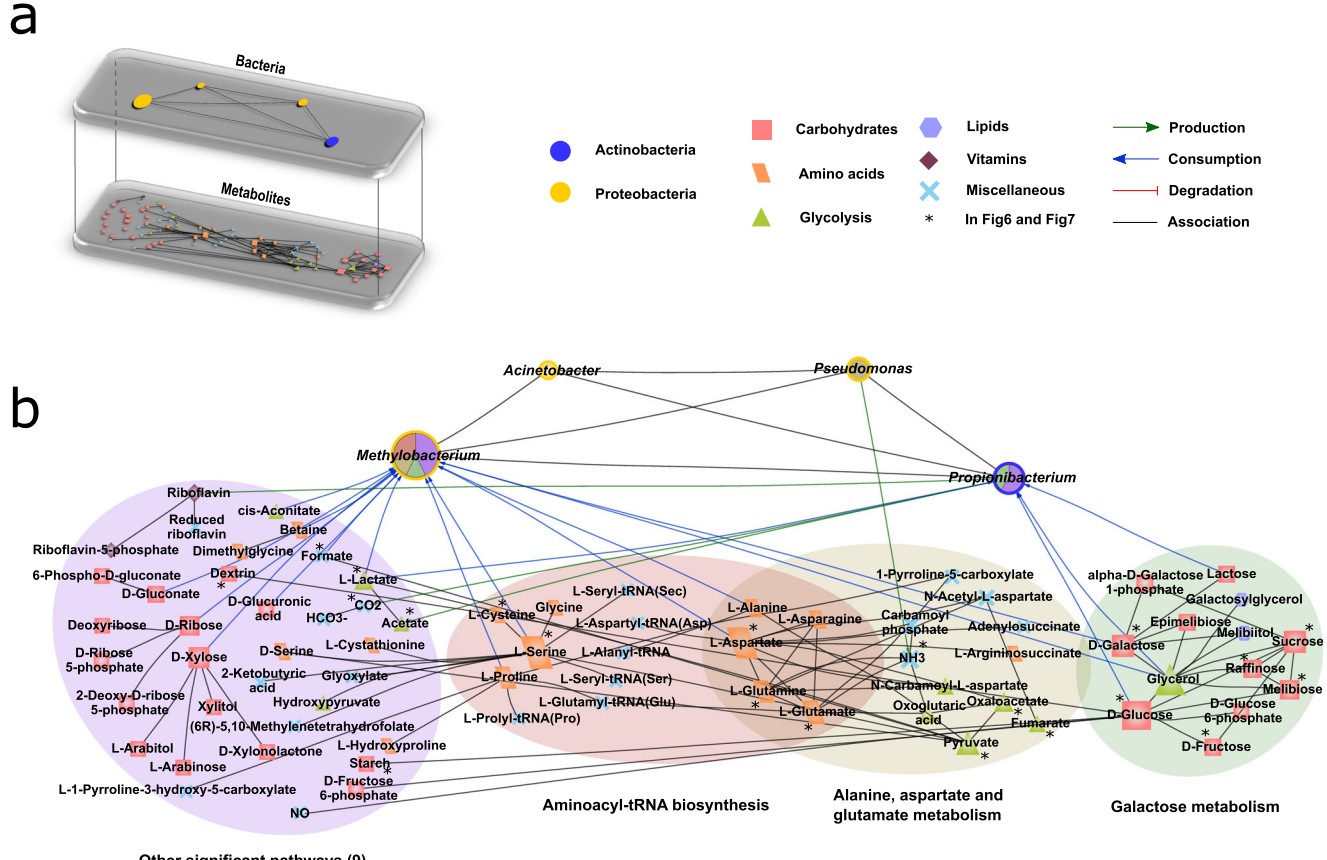

**Fig. 7 *H. pylori*-affected bacteria-metabolite network in gastric environment of dyspeptic patients. a** Multilayer (bacteria-metabolite) network representation: the first layer is derived from Fig. 5 and represents the consensus network (confirmed in two different datasets of gastric mucosa: Paroni Sterbini et al.[22] and Parsons et al.[29]) with *H. pylori*-affected bacteria nodes that present information on metabolite interaction in ref. [148]. The second layer represents the network whose nodes are the metabolites in ref. [148] interacting with the bacteria network in the first layer; different node shapes and colours refer to different metabolite classes (carbohydrates, amino acids, glycolysis, lipids, vitamins, miscellaneous). **b** In depth visualization of the bacteria-metabolite network interactions. The metabolites are grouped according to their involvement in significant pathways. For discernibility, the metabolites are arranged according to three significant pathways ($p < 0.05$ after Benjamini correction as result of a metabolite pathway enrichment analysis by a one-sided test) and a fourth group that encloses altogether nodes associated to other significant pathways (please refer to the method section: Bacteria-metabolite multilayer network construction and metabolite pathway analysis); note that only metabolites present in significant pathways are here displayed. For more information, please refer to Supplementary Fig. 18 and Supplementary Data 7. The bacteria node stroke color is associated to the phyla information as in Fig. 5, whereas the different colours in the inner fill are associated to the different pathways and their extent is proportional to the number of metabolites that the bacterium connects with in the different displayed pathways.

patients suffering from migraine[109], intestinal dysbiosis and colorectal cancer[110,111], an effect already associated with the use of PPIs such as omeoprazole[112].

## Discussion

This study indicates the necessity of including nonlinear multidimensional techniques into clinical studies based on 16 S metagenomic sequencing data, since drawing a study's conclusions by solely relying on linear techniques, such as PCA and MDS, can lead to data misinterpretation and impair the translational path from research to diagnostic. In the era of post-genomics and systems approaches, nonlinear dimension reduction and clustering by MCE and MC-MCL can offer new insights into complex clinical 16 S metagenomics data, like the ones studied in this article or the presence of clinical sub-types, and serve as a valuable tool in the run towards precision medicine. Moreover, this study shows how it is possible to complement multivariate analysis by means of network analysis employing PC-corr algorithm, that accounts for the bacteria responsible for the sample discrimination and their co-occurrence relationships. Precisely, from the system point of view the obtained microbial differential networks

pinpointed marked bacteria–bacteria interactions and modules affected by PPI treatment in the gastric environment in dyspepsia and by *H. pylori* infection in the gastric mucosa. Moreover, we elucidated via bacteria-metabolite multilayer networks, possible metabolic alterations produced by the perturbed bacteria communities and the respective metabolic pathways involved in those changes. The fact that we find significant metabolic pathways associated to the discriminative bacteria networks, which are detected by PC-corr, is a nontrivial finding that suggests the reliability and impact of the integrated machine learning/network biology methodology we propose. However, some limitations frequently present in integrative systems biology also affect our study. For instance, when we adopt protein interaction networks in drug repositioning[113] or in disease analysis[114], we are aware that further information such as the contextualization of the network to the peculiar organ, tissue, cell or cell-compartment would allow more accurate results. The same is valid for our study, where we have to adopt a generic bacteria-metabolite gut network, because it is the most updated resource currently available in the field. This means that when—hopefully in future—more specialized bacteria-metabolite networks will be available for the gastric

mucosa/fluid and even in specific areas of the stomach, then our analysis—such as many other omic analyses in integrative network biology—will benefit of this quantum leap in the data quality and contextualization. Hence, we suggest that our findings can be an important starting point to design therapies that consider not only *H. pylori* infection but also the directly associated microbial alterations as well as the indirect alterations due to the drugs used for *H. pylori* eradication such as PPI.

## Methods

**Dataset description**. In the case of Amir3 (esophageal mucosa), the 16 S rRNA gene sequences were generated by Amir and colleagues[21] and are publicly available via the MG RAST database (http://metagenomics.anl.gov/linkin.cgi?project=5767). The dataset was obtained from 16 esophageal mucosal biopsies of eight individuals before and after eight weeks of PPI treatment. Two patients with heartburn presented normal oesophagogastroduodenoscopy (H) indicating that they present healthy oesophageal tissues but are exposed to gastric refluxate, four patients had oesophagitis (ES) and two had Barrett's oesophagus (BE). Metagenomics data were obtained by pyrosequencing 16 S rRNA gene amplicons on the GS FLX system (Roche). Data were processed by replicating the bioinformatics workflow followed by Amir and colleagues[21] in order to obtain the matrix of the bacterial absolute abundance: sequence reads were analysed with the pipeline Quantitative Insights into Microbial Ecology (QIIME) v. 1.6.0[115] using default parameters (sequences were removed if shorter than 200 nt, if they contained ambiguous bases or uncorrectable barcodes, or if the primer was missing). Operational Taxonomic Units (OTUs), that are clusters of sequences showing a pairwise similarity no lesser than 97%, were identified using the UCLUST algorithm (http://www.drive5.com/usearch/). The most abundant sequence in each cluster was chosen as the representative of its OTU, and this representative set of sequences was then used for taxonomy assignment by means of the Bayesian Ribosomal Database Project classifier[116] and aligned with PyNAST[117]. Chimeras, that are PCR artefacts, were identified using ChimeraSlayer[118] and removed. The Greengenes database, which was used for the annotation of the reads, additionally identifies groups of bacteria that are supported by whole-genome phylogeny, but are not yet officially recognized by the Bergeys taxonomy, which is the reference taxonomy and is based on physiochemical and morphological traits. This results in a special annotation for some taxa, like *Prevotella*, that thus appears both with the general annotation, that is *Prevotella*, and with the special annotation, that is between square brackets, [*Prevotella*]. The list of primers used and its sequence can be found in Supplementary Table 9.

For Amir4 (gastric fluid), the dataset was generated in the same study as for Amir3 by Amir and colleagues[21], and is public and available in the MG RAST database (http://metagenomics.anl.gov/linkin.cgi?project=5732). It comprises eight patients, whose gastric fluid was sampled at two different time points, that is before PPI treatment and after eight weeks of PPI treatment, for a total of 16 samples. The patients are the same described in Amir3. Metagenomics data were obtained by pyrosequencing fragments of the 16 S rRNA gene amplicons on the GS FLX system (Roche). Then the data were processed by replicating the same bioinformatics workflow followed by Amir and colleagues[21] that was described in the previous data description (Amir3), in order to obtain the matrix of the bacterial absolute abundance. As for Amir3, the Greengenes database was used for the annotation of the reads. The list of primers used and its sequence can be found in Supplementary Table 9.

The Paroni Sterbini (gastric mucosa) dataset was generated by Paroni Sterbini and colleagues[22], and is public and available in the NCBI Sequence Read Archive (SRA) (http://www.ncbi.nlm.nih.gov/sra, accession number SRP060417), where all details pertaining the sequencing experimental design are also reported. It contains 24 biopsy specimens of the gastric antrum from 24 individuals who were referred to the Department of Gastroenterology of Gemelli Hospital (Rome) with dyspepsia symptoms (i.e. heartburn, nausea, epigastric pain and discomfort, bloating, and regurgitation). Twelve of these individuals (PPI1 to PPI12) had been taking PPIs for at least 12 months, while the others (S1–S12) were not being treated (naïve) or had stopped treatment at least 12 months before sample collection. In addition, nine (five treated and four untreated) were positive for *H. pylori* infection, where *H. pylori* positivity or negativity was determined by histology and rapid urease tests. Metagenomics data were obtained by pyrosequencing fragments of the 16 S rRNA gene amplicons on the GS Junior platform (454 Life Sciences, Roche Diagnostics). Then the sequence data were processed by replicating the bioinformatics workflow followed by Paroni Sterbini et al.[22]—which is in principle the same procedure as for the Amir datasets—in order to obtain the matrix of the bacterial absolute abundance. The list of primers used and its sequence can be found in Supplementary Table 9.

The Parsons (gastric mucosa) dataset was generated by Parsons and colleagues[29], and is public and available in the EBI short-read archive (the European Nucleotide Archive, ENA) (https://www.ebi.ac.uk/ena, accession number PRJEB21104). In the original study, the authors focused on the analysis of gastric biopsy samples of 95 individuals (in groups representing normal stomach, PPI treated, *H. pylori*-induced gastritis, *H. pylori*-induced atrophic gastritis and

autoimmune atrophic gastritis), selected from a larger prospectively recruited cohort patients who underwent diagnostic upper gastrointestinal endoscopy at Royal Liverpool University Hospital[29]. RNA extracted from gastric corpus biopsies was analysed using 16 S rRNA sequencing (MiSeq). Then the sequence analysis was performed, as described by the authors in the supplementary methods of the original article[29], by first cleaning the sequences employing Sickle 1.200 to remove reads shorter than 10 bp (available in QIIME[115]), assembles of paired-end reads was processed via FLASH[119] and samples were compared using BLASTn[120]. Subsequently, same procedures were carried out for the obtainment of OTU tables as in Amir datasets. Here we focused on the analysis of gastric biopsy specimens (in total 42 samples) from normal stomach group (20 patients) and belonging to the *H. pylori* gastritis group (22 patients). As described in ref. [29], patients in the normal stomach group showed normal endoscopy, no evidence of *H. pylori* infection by histology, rapid urease test or serology, were not treated by PPI and were normogastrinaemic. Patients in the *H. pylori* gastritis group were instead positive to *H. pylori* infection by urease test, histology and serology, were not taking PPI medication and were normogastrinaemic. The list of primers used and its sequence can be found in Supplementary Table 9.

**PCA, MDS and LDA**. The mainstream multivariate methods to unsupervisedly explore data patterns in metagenomic studies are based on linear dimension reduction, in particular PCA[121,122] and MDS[123,124], also known as PCoA, methods that have been used to explore and visualize data structure in many metagenomic studies, from sponge[125,126] to gastric tissue microbiota[22]. These tools perform a dimension reduction of the data either by multidimensional variance analysis (for instance PCA) or dissimilarity embedding (for instance MDS/PCoA). PCA collects uncorrelated variance in the multidimensional space, creating new synthetic orthogonal variables, which are linear combinations of the original ones, then plots the samples in a reduced space using the new variables that embody the largest orthogonal variances. MDS computes dissimilarities between every pair of samples, plotting the Euclidean part of these dissimilarities as distances between every pair of points (MDS) in a reduced space, in this way the linear part of the sample relations can be represented.

In ecology, distance (or dissimilarity) matrices are a major way to transpose the ecological information of samples in terms of their species composition and abundance[127,128]. In this article we will consider classical MDS (which uses Euclidean distance and is in practice equivalent to PCA[129,130]), and nonmetric MDS (NMDS) obtained according to Sammon's Mapping[131]. In the latter, the elements of the multivariate space are mapped onto a lower dimensional space while retaining the original inter-point dissimilarities, by means of a nonlinear, but monotonic transformation (Sammon Mapping). Since it respects the ranking of dissimilarities, it tends to linearize the relationships between the samples. In addition, MDS will be performed also according to Bray-Curtis (MDSbc) dissimilarity and weighted UniFrac (MDSwUF) distance because they are considered the reference in metagenomics studies. Bray-Curtis dissimilarity quantifies how dissimilar two sites (samples) are based on counts (bacterial abundances), where 0 means two samples are identical and 1 means that the two samples do not share any taxa[132,133]. Dissimilarly, the UniFrac distance, either unweighted (qualitative) or weighted (quantitative), is the most popular phylogenetic distance measure for the microbial community diversity between different samples (also known as β-diversity[134]) and, differently from the previous discussed methods, uses the phylogenetic information (which is an external knowledge not contained in the dataset) on the taxa to compare samples. In particular, its weighted-version weights the branches of a phylogenetic tree based of the taxa abundance information[135–138]. Hence the weighted UniFrac distance directly accounts for differences in the abundance of different kinds of bacteria, and can be crucial to describe community changes[136] in the studied samples.

We need to specify that both MDSwUF and NMDS are in practice nonlinear methods and weighted UniFrac is not a classical unsupervised technique like the others. In fact, MDSwUF adopts a distance that combines the information given by the bacterial abundance of the dataset with the supervised prior (external) knowledge regarding the known hierarchical phylogenetic relationship among the bacteria. However, like PCA, MDS can fail to detect patterns if data are not properly linearized[139]. For instance, see Supplementary Fig. 1c-d where MDSbc and NMDS respectively fail to resolve the Tripartite-Swiss-Roll dataset. When we consider clinical 16 S rRNA amplicons data, this failure potentially reduces the chances of correctly pinpointing samples, which may represent clinical subspecies, and thus remain undetected and undiagnosed. In brief, these methods are not efficient to perform hierarchical embedding directly from the abundance value, since hierarchies preserve tree-like structures, and tree-like structures follow a hyperbolic, thus nonlinear, geometry[140–142]. Only MDSwUF is able to account for nonlinear hierarchical organization, yet this is not directly inferred from the abundance values, but rather forced as a constraint of prior supervised knowledge on the phylogeny of bacteria. For this reason we cannot offer a test on the Tripartite-Swiss-Roll dataset.

In our analysis of the Paroni Sterbini dataset, we also showed the results of a supervised technique, Linear Discriminant Analysis (LDA), which uses the labels to perform dimension reduction. LDA aims to separate the samples into groups based on hyperplanes and describe the differences between groups by a linear classification criterion that identifies decision boundaries between groups[123]. This

technique is not congruous (and sometimes statistically invalid) for small sample size datasets. The reason is that given the reduced sample size we cannot divide the dataset in a training and test set, which is a fundamental requirement of supervised methods such as LDA.

**Minimum Curvilienar Embedding**. In 2010, Cannistraci et al.[23] introduced the centred version of Minimum Curvilinear Embedding (MCE), which provided notable results in: (i) visualisation and discrimination of pain patients in peripheral neuro-pathy, and the germ-layer characterisation of human organ tissues[23]; (ii) discrimination of microbiota in sea sponges[125]; (iii) embedding of networks in the hyperbolic space[141]; (iv) stage identification of embryonic stem cell differentiation based on genome-wide expression data[31]. In this fourth example, MCE performance ranked first on 12 different tested approaches (evaluated on 10 diverse datasets). More recently in 2013[33], the noncentred version of the algorithm, named ncMCE, has been used: (i) to visualise clusters of ultra-conserved regions of DNA across eukaryotic species[143]; (ii) as a network embedding technique for predicting links in protein interaction networks[33], outperforming several other link prediction techniques; (iii) to unsu-pervisedly reveal hidden patterns related with gender difference and metabolic-disease risk-factors in lipidomic profiles extracted from human plasma samples[144]; (iv) to unsupervisedly infer and visualize phylogenetic (hierarchical) relations directly from individual SNP profiles in human population genetics[34]. Finally, also applications in nonbiological problems such as the unsupervised discrimination of bad from good radar signals[33], represented a proof of concept of the universality of MCE for addressing nonlinear investigation of data and signals in general. Also in the case of the metagenomics studies targeting sea sponges,[125,126] both MCE and its noncentred variant[23,33] once again proved successful in detecting structure where PCA and MDS could not, or hardly find any. This is mainly because MCE/ncMCE are unsupervised and parameter-free topological machine learning for nonlinear dimensionality reduction and multivariate analysis, that are able to perform a hierarchical embedding (For more information see Supplementary Note 1—MCE to unsupervisedly infer and visualize phylogenetic (hierarchical) relations).

This study stems from the intuition that MCE/ncMCE analysis could successfully reveal undetected patterns also in esophageal and gastric metagenomics data, where only unsupervised linear methods or classical nonlinear methods such as NMDS and MDSwUF had been used and had failed to achieve any clear-cut result[21,22].

Minimum Curvilinearity (MC)[23], the principle behind MCE and ncMCE, was invented with the aim to reveal nonlinear data structures also, and especially, in the case of datasets with few samples and many features. MC principle suggests that curvilinear (nonlinear) distances between samples may be estimated as pairwise distances over their Minimum Spanning Tree (MST), constructed according to a selected distance (Euclidean, correlation based, etc.) in a multidimensional feature space (here the metagenomic data space). In this study, we considered Pearson- and Spearman-correlation-based distance (refer to ref. [23] for details on the way to compute the distance for the MST). The collection of all nonlinear pairwise distances forms a distance matrix called the MC-distance matrix or MC-kernel, which can be used as an input in algorithms for dimensionality reduction, clustering, classification and generally in any type of machine learning. In MCE and ncMCE, the MC-kernel (which is noncentred for ncMCE) is followed by dimensionality reduction using singular value decomposition (SVD), and then by the projection of the samples onto a two-dimensional space for visualisation and analysis. Thus, MCE/ncMCE is a form of nonlinear and parameter-free kernel PCA[33]. In the rest of the article we will simply use the name MCE to indicate both MCE and ncMCE, since the centring transformation is related to the specific data pre-processing and will be specified for each dataset as a technical detail in the respective results' tables.

**Procedure to evaluate the performance of the dimension reduction algo-rithms**. The performance of the mentioned dimension reduction algorithms is evaluated as the ability to separate the samples in the first two dimensions of embedding since, as discussed, they are related with the treatment/infection response. In order to quantitatively evaluate the performance, we use a recently proposed index termed Projection Separability Index (PSI) used for sample separation[145]. This index can be defined for any separation-measure and in this study we considered well-known measures: Area Under the ROC-Curve (PSI-ROC) and Area Under the Precision-Recall curve (PSI-PR), that are regularly used to quantitatively measure the performance of a binary predictor.

More precisely, in the 2D space a line is drawn between the centroids of the two groups that are compared, subsequently all the points are projected on this line and then AUC and AUPR are computed for the projected points. This index can actually be applied not only in a 2D space, but in any N dimensional space. For the calculation of the centroids we consider the 2D-median of each cluster/class's group. In case more than two groups are present in a dataset, all the AUC and AUPR values between the possible pair-groups are computed. Then, the following formula is applied: $E/(1 + \delta)$, where E is the mean of the pairwise PSI values and $\delta$ their standard deviation. Thus, the standard deviation works as a penalization in case of outliers PSI values, ensuring that the overall PSI is high only when all pairwise PSI values are close to the mean. The computed values are finally chosen as an overall estimator of separation between the groups in the 2D reduced space. This case applies only to the Paroni Sterbini dataset, which is composed of three or,

possibly, four groups of samples. All the other datasets are instead composed of two groups.

It is important to note that the PSIs were also applied to the data in the original high-dimensional (HD) space, as a reference to see how good the unsupervised dimension reduction approaches are in preserving the original group separability of the HD space.

All the algorithms were tested considering (when allowed by the dimension reduction method) data centring or noncentring. In addition, multiple normalization options were investigated and the datasets were considered under a certain type of normalization: division by the column—which reports the OTU—sum (indicated by DCS); division by the row—which reports the sample-sum (indicated by DRS); function $\log_{10}(1 + x)$ applied to the dataset (indicated by LOG).

In order to verify that the performances obtained by our evaluations using PSI on the DR techniques are not obtained by chance, we calculate a measure termed trustworthiness, which exploits a resampling technique based on label-reshuffling to build a null model (Supplementary Fig. 19). The labels are reshuffled uniformly at random on the embedded points whose location is maintained unaltered in the reduced space. For each random reshuffling (the total number of reshuffling is a resampling parameter decided by the user, we adopted 1000 realizations), a PSI measure value is computed. The collection of all these values is used to draw the null model distribution. This distribution is employed to compute the probability to get at random a separation equal or higher than the one detected by using the original labels.

**From Markov Clustering to Minimum Curvilinear Markov Clustering**. MCL is an unsupervised algorithm for the clustering of weighted graphs based on simulations of (stochastic) flow in graphs[146] (http://micans.org/mcl/). By varying a single para-meter called inflation (with values between 1.1 and 10), clustering patterns on different scales of granularity can be detected. For clustering samples of a multidimensional dataset, the workflow starts with the computation of correlations (generally Pearson correlations) between the samples, and creates an edge between each pair of samples, where the edge-weight assumes the value of the respective pairwise positive sample correlation, or values zeros in case of negative correlations. This generates a weighted correlation graph (network), which is used as a map to simulate stochastic flows and detect the structural organization of clusters in the graph.

With the purpose of creating and testing a nonlinear variant of the MCL algorithm, we adopt an innovative algorithm, which was recently proposed and called MC-MCL[147]. The idea is the following. The MC-kernel—discussed above in the MCE section—is a nonlinear distance matrix (or kernel) that expresses the pairwise relations between samples as a value of distance: small samples distance indicates sample similarity, while large samples distance indicates sample dissimilarity. Here we reverse (using the following function: $f(x) = 1 - x$ and after this we put to zero the negative values—strategy already applied in the original MCL algorithm—of the MC-distance kernel to get a MC-similarity kernel, where small values (close to zero) indicate low sample similarity and large values (close to one) indicate high sample similarity. A technical detail: for the computation of the MC-distance kernel, it is necessary to firstly square root the original distances (correlation based) between the samples. As already investigated in ref. [23], this attenuates the estimation of large distances and amplifies the estimation of short distances; consequently it helps to regularize the nonlinear distances inferred over the MST in order to subsequently use them for message passing[23] (such as affinity propagation) or flow simulation (such as MCL) clustering algorithms.

Then, the standard stochastic flow simulations of MCL algorithm runs on the graph weighted with the values of the MC-similarity kernel (which collects pairwise nonlinear associations between samples) instead of the Pearson-correlation kernel (which collects pairwise linear associations between samples). In practice, this is an algorithm for clustering that is a nonlinear version (based on the MC-kernel) of the classical MCL. The goal of the MC-MCL analysis is to verify whether the use of the MC-kernel improves performance, by solving nonlinearity, not only in dimension reduction (such as in MCE) but also in clustering (such as in MC-MCL).

**Procedure to evaluate the performance of clustering algorithms**. The cluster-ing algorithms MCL and MC-MCL were applied to the datasets, either raw, or after the same normalization procedures used before dimensionality reduction (DCS: division by column (OTU) sum; DRS: division by row (sample) sum; LOG: function $\log_{10}(1 + x)$ applied to the dataset) and their performance was evaluated by means of accuracy. The accuracy is computed as the ratio of the number of samples assigned to the correct clusters over the total number of samples. For both MCL and MC-MCL, we tested Pearson and Spearman correlations to build the similarity measure to feed into the clustering methods. The Spearman correlation can also detect a subclass of nonlinear associations (which have monotonic shape function) or correct for outliers. Differently from what suggested for large gene datasets with thousands of samples in ref. [146] (http://micans.org/mcl/), in this study we had to consider the whole set of original positive correlations without applying any threshold (cut-off) to the values. This was compulsory, since we considered datasets with few samples. In our case, to keep the graph connected, with one unique connected component, we could not introduce any kind of threshold that would otherwise alter the real graph connectivity (dividing the graph in dis-connected components) and hence the clustering result. Since the MCL algorithm

needs a single input parameter (inflation) to control the granularity of the output clustering, we ran it for different inflation values until we achieved the desired number of clusters. Finally, in the Paroni Sterbini et al. dataset[22] it was not clear in advance whether the correct number of clusters present in the multidimensional space was three or four. Hence, we tested the clustering algorithms considering as output both three and four clusters' configurations, and we identified as the best solution the one that offered the highest accuracy.

**PC-corr network**. Furthermore, we investigated the effect of PPI on the microbiota of gastric fluid and gastric mucosa in dyspeptic patients, and the changes induced by *H. pylori* infection on the gastric mucosal microbiota, by means of the PC-corr approach[32]. PC-corr represents a simple algorithm that associates to any PCA segregation a discriminative network of features' interactions[32]. It is a method for linear multivariate-discriminative correlation network reverse engineering, that, thanks to its multivariate nature, can help to stress and squeeze out the underlying combinatorial and multifactorial mechanisms that generate the differences between the studied conditions[32]. Said what PC-corr is able to do, now we offer an intuitive explanation of how it works. PCA is one of the most employed approaches to unsupervisedly map linear dissimilarities (hidden in the high-dimensional space) into a visible space of data representation. When we notice that two or more groups of samples are separated along one of the axes of this representation space, generally the first question is to discover what are the features that are contributing more to this separation. This is easily achievable by analysing the PCA loading values that are associated to the axis along, which emerges the sample separation under investigation. But the loading values do not provide any information on how those features mutually interact. On the other hand, a correlation network between the features provide information on their associative relation but not on their contribution to the discrimination. PC-corr is an algorithm that is able to integrate together the discriminative information of the loadings with the combinatorial information of the correlations. Indeed, PC-corr offers as output a discriminative correlation network of features that can help to elucidate the possible associative mechanisms that are at support of the sample separation along a specific axis of PCA representation. Hence, for the studied datasets, it can be employed to point out the possible presence of bacterial alterations and their interplay, induced by a medical treatment (PPIs in dyspepsia) or infectious state (*H. pylori*).

**Bacteria-metabolite multilayer network construction and metabolite pathway analysis**. We used a recently realized bacteria-metabolite bipartite network, which is an open access resource[148] to infer the metabolic activity of the bacteria presented in the intersections of Figs. 4 and 5. The study[148] provided a large set of 9136 bacteria to metabolite interactions validated on experimental studies from mouse and human gastroenteric microbiota. It was available as a network, named NJC19, where node represented either bacteria or metabolites connected by several types of edges (e.g. production, consumption, degradation). In this dataset we restricted the analysis to interactions found on human bacteria. Since the dataset identified bacteria according to the taxonomic levels of species while our data referred to the genus level, we made a new form of the NJC19 network with edges starting from the bacteria genus to metabolites. When we did not find any interactions for specific bacteria, we discarded them from further analyses. Therefore, from the list of intersected bacteria from Fig. 4 (*Porphyromonas, Capnocytophaga, Streptophyta, Granulicatella, Clostridiales, Oribacterium, Veillonella, Bulleidia, Fusobacterium, Leptotrichia, Campylobacter, Prevotella*) we dropped *Streptophyta, Granulicatella, Oribacterium, Bulleidia* and *Prevotella*. Similarly, from the intersected bacteria of Fig. 5 (*Enhydrobacter, Methylobacterium, Catonella, Pseudomonas, Acinetobacter, Sphingomonas, Propionibacterium, Bulleidia*) we dropped *Bulleidia, Catonella, Sphingomonas* and *Enhydrobacter*. For the graph representation, we used the color code already applied in the previous figures for the bacteria according to the taxonomic order. While for metabolites we classified them in seven classes, assigning to each a different node shape and colour: vitamins, glycolysis, lipids, amino acids, carbohydrates, amines and miscellaneous. Furthermore, an enrichment analysis of metabolites (linked to the discriminative bacteria networks detected by PC-corr) has been conducted to unveil the metabolic pathways that might be associated to these bacteria perturbations. To this purpose, we used the framework provided by metaboloAnalyst suite[149]. Specifically, we performed the "Enrichment Analysis" against the KEGG database and we selected the significant pathways according to the Benjianini corrected p-values smaller than the significance level of 0.05. For the case of the *H. Pylori*-affected network, just few nodes were available and only one significant pathway was obtained from it with few metabolite hits. Therefore, the network was expanded by adding first neighbours metabolites—obtained from KEGG—from the current ones.

Finally, the metabolite layer network nodes were grouped according to the three most significant pathways in both the PPI- and *H. Pylori*-affected bacteria-metabolite networks. This was ensured according to the following procedure: a ranking was generated for the list of significant pathways in each of the two networks. Then, the nodes of each network were grouped according to the three pathways with the highest average ranking in the two networks, which in our study are: aminoacyl-tRNA biosynthesis; galactose metabolism; Alanine, aspartate and glutamate metabolism. A fourth group encapsulating the metabolites involved in the other significant pathways was also provided. Regarding the links considered in each bacteria and metabolite layer, the bacteria–bacteria associations were

maintained from Figs. 4 and 5, while edges between metabolites were obtained by the metabolite interaction involved in the significant enriched KEGG pathways.

The processing pipeline has been developed in the R environment[150] and by using the following packages: igraph[151], taxize[152], graphite[153], RCy3[154].

More information about the computing platforms concerning the analyses found in this study can be found in the Supplementary Note 9—Computing platforms adopted to implement the algorithms.

**Reporting summary**. Further information on research design is available in the Nature Research Reporting Summary linked to this article.

## Data availability
The data that support the findings of this study are available in: the MG RAST database accession code mgp5767 and mgp5732 for Amir3 and Amir4, respectively; the NCBI Sequence Read Archive (SRA) accession number SRP060417 for the Paroni Sterbini data; and the EBI short-read archive accession code PRJEB21104 for the Parsons data. In the case of the artificial microbial-like and Tripartite-Swiss-Roll datasets, they are available within the paper supplementary information Supplementary Data 13 and 14, respectively.

## Code availability
Codes for the PSI measure can be found in the biomedical-cybernetics git repository https://github.com/biomedical-cybernetics/projection-separability-indices. The code for MC-MCL clustering algorithm can be found in https://github.com/biomedical-cybernetics/minimum-curvilinear-Markov-clustering. The R or MATLAB code for PC-corr network analysis can be found in https://github.com/biomedical-cybernetics/PC-corr_net. The code to compute the trustworthiness can be found in https://github.com/biomedical-cybernetics/trustworthiness. The MCE code for nonlinear dimensionality reduction can be accessed from https://sites.google.com/site/carlovittoriocannistraci/5-datasets-and-matlab-code/minimum-curvilinearity-ii-april-2012.

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

## Acknowledgements

This work was supported by the TUD Forschungspool grant number F-004243-552-694-3430000. S.C. wrap-up postdoc period was supported by the Dresden International Graduate School for Biomedicine and Bioengineering (DIGS-BB), granted by the Deutsche Forschungsgemeinschaft (DFG) in the context of the Excellence Initiative. Computing support was offered by Centre for Information Services and High-Performance Computing (ZIH) of the TUD. P.S. is supported by Estonian Research Council Starting Grant PUT1130. C.D. is funded by the Research Grants—Doctoral Programs in Germany (DAAD), Promotion program Nr: 57299294. We thank Alexander Mestiashvili and the BIOTEC System Administrators for their IT support and Claudia Matthes, Gloria Marchesi and Sabine Zeissig for their administrative assistance.

## Author contributions

C.V.C. developed Minimum Curvilinearity (MCE), Minimum Curvilinear Markov Clustering (MC-MCL) and the Projection-based Separability Index (PSI). C.V.C. conceived all the study and the data analysis workflow with feedbacks from Mi.Sc. and S.W. G. C.D., S.C. and A.P. performed the computational analysis of the data and realized the figures under C.V.C. guidance with help of A.Z. for the bacteria-metabolite analysis. C.D., S.C., A.P. together with C.V.C. wrote the manuscript with valuable suggestions of P.S. and A.Z. F.P.S., L.M., G.C., G.I., B.P., Ma.Sa., G.G. and A.G. provided data and knowledge about the Paroni Sterbini et al. data cohort. B.N.P., U.Z.I. and M.P. provided data and knowledge about the Parsons et al. data cohort. All authors discussed the results and revised the manuscript. C.V.C. planned, directed and supervised the study. C.V.C. acquired funding.

## Funding

## Competing interests

The authors declare no competing interests.
