## [Peer Review File · Nature Communications]

Point by point reply to the Reviewers' comments

We are delighted to see such positive feedbacks from all the Reviewers. The recognition of our work is a high reward for our effort and commitment on this study. The improvements in the study could have not been done without the thoughtful work of the reviewers.

Reviewer #2 (Remarks to the Author):

I am satisfied with the text modifications and additional analyses provided by the authors. These contribute to make the presentation and findings clearer and stronger. I have no other comments.
Alban Ramette

REPLY: We are pleased to know that Dr. Ramette is satisfied with our modifications, and that thinks better of our findings after the first round of revision.

Reviewer #3 (Remarks to the Author):

The authors have provided very constructive replies to my comments. Several technical issues have been explained clearly and the addition of metabolomics data has strengthened the biological aspect of the manuscript. I have no more comments.

REPLY: We appreciate that the Reviewer found our comments constructive and that recognized the strengthened of the biological aspects of the article through the addition of the bacteria-metabolic network analysis.

Reviewer #4 (Remarks to the Author):

The authors have done a nice job addressing the previous critiques.

REPLY: we are contented with the Reviewer's response to our first round of revision

Reviewers' Comments:

Reviewer #2:

Remarks to the Author:

I am satisfied with the text modifications and additional analyses provided by the authors. These contribute to make the presentation and findings clearer and stronger. I have no other comments.

Alban Ramette

Reviewer #3:

Remarks to the Author:

The authors have provided very constructive replies to my comments. Several technical issues have been explained clearly and the addition of metabolomics data has strengthened the biological aspect of the manuscript. I have no more comments.

Reviewer #4:

Remarks to the Author:

The authors have done a nice job addressing the previous critiques.

Point by point reply to the Reviewers comments

In order to facilitate Editor's and Reviewers' check during revision, each point-by-point reply reports at the end also the exact text that we modified in the revised main manuscript file (whereas, for reason of space, revised text of supplementary notes, figures and tables should be consulted directly in the supplementary information file).

The revised text parts both in the main manuscript file and in the supplementary information file are highlighted by blue character colour, and in bold if it is embedded in the old text.

Reviewer #1 (Remarks to the Author):

The paper proposes a novel, interesting and important pipeline for analyzing metagenomic data. In particular, it introduces non-linear approaches for dimensionality reduction and multivariate pattern analysis with differential network analysis that reveals mechanisms of re-organizations of combinatorial microbial variations induced by a medical treatment or by an infection. As it is crucial to address non-linearity of systems-level data in biology and medicine, the findings can be of importance for design new therapies for infection, a timely and important topic.

This is a very good and well written manuscript that flows very well and is enjoyable to read. Even the complex computational methods are well written in a way accessible to wide scientific audience.

Two minor suggestions for improvement are:

#1 section "PC-corr network" could be expanded and written to the same level of detail and intuition as explanations of previous methods. Perhaps moving the sentence on lines 785-6 to that section and expanding a bit would address the point.

REPLY: We thank the Reviewer for pointing this out. We agree that the suggested modification will improve the manuscript.

ACTION: We modified the text and added more information about PC-corr. We report below the new text with modifications which is at Pag 20 of revised manuscript:

“Furthermore, we investigated the effect of PPI on the microbiota of gastric fluid and gastric mucosa in dyspeptic patients, and the changes induced by *H. pylori* infection on the gastric mucosal microbiota, by means of the PC-corr approach⁶². PC-corr is an algorithm that associates to any PCA segregation a discriminative network of features' interactions⁶². It is a method for linear multivariate-discriminative correlation network reverse engineering, that, thanks to its multivariate nature, can help to stress and squeeze out the underlying combinatorial and multifactorial mechanisms that generate the differences between the studied conditions⁶².

Said what PC-corr is able to do, now we offer an intuitive explanation of how it works. PCA is one of the most employed unsupervised approaches that translates the linear dissimilarities hidden in the high-dimensional space of our multivariate data into a visible 2- or 3D representation. When we notice that two or more groups of samples are separated along one of the axes (dimensions) of this representation, the first question is to discover which features mainly contribute to this separation. This is usually achievable by

analysing the PCA loading values of the principal component (i.e. the coefficients/coordinates assigned by that principal component to the original features) that is associated to the axis along which the samples separate into groups. Yet, while the loading values indicate which features have more weight in separating the samples, they do not provide any information on how such features mutually interact. A feature correlation network can indeed provide this information, but would omit their contribution to the discrimination of the samples. PC-corr is an algorithm that combines the discriminative information of the loadings with the combinatorial information of the correlations. Thus, PC-corr offers as output a discriminative correlation network of the features that drive the separation of the samples along a specific axis of the PCA representation, and this can help to elucidate the possible mechanisms that are behind that samples separation. Hence, for the studied datasets, it can be employed to point out the possible presence of bacterial alterations and their interplay, induced by a medical treatment (PPIs in dyspepsia) or infectious state (*H. pylori*). ”

#2 Rewrite the Abstract to explain the importance and the main methods a bit better. Perhaps using "PPIs" to mean Proton Pump Inhibitors is not the best choice, as it may be confused with protein-protein interactions, that's commonly associated with "PPIs".

REPLY: Thanks to the Reviewer for this comment. PPIs is also frequently used as acronym in medicine literature for proton pump inhibitor drugs, and we agree that the homonymy can potentially cause confusion but this is actually addressed by the fact that we explain the acronym already in the abstract

ACTION: we now specify the entire name in the abstract to avoid confusions:

“The stomach is inhabited by diverse microbial communities, co-existing in a dynamic balance. Long-term use of drugs such as proton pump inhibitors (PPIs), or bacterial infections such as *Helicobacter pylori*, cause significant microbial alterations. Yet, studies revealing how the commensal bacteria re-organize, due to these perturbations of the gastric environment, are in early phase and rely **principally** on linear techniques for multivariate analysis.

Here we disclose the importance of complementing linear dimensionality reduction techniques such as Principal Component Analysis and Multidimensional Scaling with nonlinear approaches **to unveil hidden patterns that with linear approaches remain unseen**. Then, we show the importance to complete multivariate pattern analysis with differential network analysis, to reveal mechanisms of re-organizations, which emerge from microbial variations induced by a medicament treatment (PPIs) or an infectious state (*H. pylori*). **Finally, we reveal metabolomic network alterations associated to the perturbed microbial communities.**”

Reviewer #2 (Remarks to the Author):

The key message in the study is that nonlinear approaches may be better than linear method to address complex, biological questions, especially to generate new insights about microbial community composition. There is a rich account in the literature (particularly in plant and animal community ecology, but also in microbial ecology over the last 10 years), which already shows this fact and provides several ways to model multimodal response data as a function of environmental, spatial or temporal gradients. For instance, unsupervised approach could be

done with correspondence analysis, and a supervised approach via canonical correspondence analysis to group samples and assess their ecological meaning given metadata is available to perform constrained analyses.

#1 Introduction. The authors ask whether "linear techniques are sufficient to bring out patterns in complex microbial data?" Line 142. The question and the approach should be reformulated, as the choice of statistical tools should be driven by a given experimental design, distribution type of data generated, and hypotheses to be tested, not the other way around (i.e. the performance of statistical tools cannot be assessed on a dataset without considering all those points; otherwise, the approach is purely exploratory and not hypothesis-driven).

If the hypothesis is a linear response of the species or community data to environmental gradients, linear models are the right approach conceptually.

REPLY: The first part of the article is focused on machine learning unsupervised detection of patterns. This is by definition data-driven and the methodology, rather than using the class labels of data to explore data difference, is to unsupervisedly discover patterns and compare them with the assumed labels. These unsupervised approaches are therefore by definition not hypothesis-driven but data-driven. Moreover, unsupervised, data-driven approaches are in principle not inferior to hypothesis-driven ones. Unsupervised approaches are especially useful to validate hypotheses or to generate new ones. Finally, unsupervised approaches are particularly useful when the number of samples is small in comparison to the features (and, in this scenario, overfitting by hypothesis driven approaches might be a problem) or when there is a class unbalance in the samples. Nonetheless, we agree and we thank the Reviewer for let us notice that we should clarify better the reason of our study in our introduction.

ACTION: we now modify part of the Introduction in the revised manuscript, adding text in which we clarify the reasons to conduct unsupervised analysis according to the procedure we describe in the study. Pag. 6:

“The common practice in unsupervised dimension reduction data analysis is to consider only the first two (or three, less used) dimensions of mapping, and the goal is to visually explore the distribution of the samples and the incidence of significant patterns ²⁵. **This type of analysis is advantageous to validate hypothesis or to generate new ones. In addition,** this procedure is particularly useful in case of studies with small size datasets ²³, **or for imbalance class samples**, to obtain unbiased (the labels are not used) confirmation of the separation between groups of samples for which diversity is theorized or expected.”

#2 Overall approach. The authors analyse data and find interesting patterns. As significant or biologically interesting as those patterns may be, there is no proof that those patterns may not have emerged due to chance alone, even if significant P values were found (i.e. not very likely but possible; see previous point).

REPLY: Thanks to the Reviewer for this comment that is useful to improve the quality of our study. We now provide further analysis and results that compute the extent to which the detected patterns can emerge at random according to different separability measures.

ACTION: We updated all PSI tables (PSI-ROC and PSI-PR) with a measure termed trustworthiness, which exploits a resampling technique based on label-reshuffling to build a null model. Then, the probability to get at random from the null model a separation equal or higher than the one detected by dimension reduction is computed. A paragraph was added at

the end of the method section “*Procedure to evaluate the performance of the dimension reduction algorithms*” with the respective clarification. We also explain how to compute the trustworthiness value with a new supplementary figure from the PSI values obtained by MCE applied on the Paroni Sterbini dataset (see below).

“In order to verify that the performances obtained by our evaluations using PSI on the DR techniques are not obtained by chance, we calculate a measure termed trustworthiness, which exploits a resampling technique based on label-reshuffling to build a null model (Suppl. Figure S2). The labels are reshuffled uniformly at random on the embedded points whose location is maintained unaltered in the reduced space. For each random reshuffling (the total number of reshuffling is a resampling parameter decided by the user, we adopted 1000 realizations), a PSI measure value is computed. The collection of all these values is used to draw the null model distribution. This distribution is employed to compute the probability to get at random a separation equal or higher than the one detected by using the original labels.”

Figure S2. Example of trustworthiness computation. *The figure reports the null model distribution obtained for the MCE embedding separation by reshuffling 1000 times the labels of the Paroni Sterbini dataset. (A) null model distribution for PSI-ROC; (B) null model distribution for PSI-PR. x-axis reports the respective PSI measure values; y-axis reports the density distribution values for the respective PSI measure. The vertical yellow lines denote the 95 percentile of the respective distributions. For the case of MCE embedding, the detected PSI-ROC and PSI-PR values (pointed by the arrows on the right tale of the distribution) are clearly significant (<0.05).*

#3 Therefore, there is a need to have a ground truth to evaluate the performance of the algorithms in the study, and it seems so far that there is no simulation or benchmarking of the proposed methods to see what patterns the methods can or cannot retrieve in simulated datasets with known structure.

REPLY: The Reviewer is right. However, the trustworthiness measure discussed above is based on a null model that, in absence of a ground truth, offers the opportunity to measure the extent to which a result similar or better than the one obtained with the original labels can be obtained by random labelling. This should be enough to reassure the Reviewer about the reliability of the provided results.

Nonetheless, in order to fully embrace the Reviewer concern, we made our best to include results also on benchmark data. Indeed, we run a simulation with synthetic “microbial-like” datasets generated with the R function `rmvzinegbin` from the library `SpiecEasi`. Specifically, we focused on the generation of a synthetic dataset with four different groups, 200 samples (50 by each class), 200 microbial features and highly sparse. The dataset was in general a challenge for the different dimensionality reduction techniques.

ACTION: We report in the study a Supplementary Table S4 with the results of this new analysis.

Table S4. Results of unsupervised analysis on the ‘microbial-like’ synthetic dataset. Best results of unsupervised dimension reduction techniques according to PSI-ROC and PSI-PR, which are indices for evaluation of sample separation in the space of the first two dimensions of embedding. HD (high dimension) indicates the separability in the high dimensional space (no dimension reduction) and it represents a reference to compare with the separability after dimension reduction. Results are ordered from the best (top) to the worst (bottom) performance. For each PSI value, the respective trustworthiness is reported.

Methods	PSI-ROC	Trust	PSI-PR	Trust
HD	1.00	0.0009	1.00	0.0009
MCE	0.99	0.0009	0.99	0.0009
MDStyc	0.97	0.0009	0.97	0.0009
MDSbc	0.96	0.0009	0.96	0.0009
PCA	0.85	0.0009	0.85	0.0009
nMDS	0.85	0.0009	0.85	0.0009
MDSwUF	0.83	0.0023	0.83	0.0020

Note: all PSI-ROC and PSI-PR values can be found in Supplementary Table S20

Abbreviations: HD: High Dimension; MCE: Minimum Curvilinear Embedding; MDSbc: Multidimensional Scaling with Bray-Curtis dissimilarity; MDSwUF: Multidimensional Scaling with weighted UniFrac distance; nMDS: Non-metric Multidimensional Scaling; MDStyc: Multidimensional Scaling with Theta-YC distance; PCA: Principal Component Analysis; PSI-ROC: Projection Separability Index measured by Area Under the Curve; PSI-PR: Projection Separability Index measured by Area Under the Precision Recall; Trust: Trustworthiness.

These results are now commented in the new revised version of the manuscript and a related section is added in the supplementary information methodology.

Text in supplementary information methodology:

“Artificial Dataset

In order to test and visualize how the algorithms could detect nonlinearity, we performed the analyses on two artificial datasets: (1) The Tripartite-Swiss-Roll dataset: an artificial dataset characterized by nonlinear structures and generated as discretization of the manifold associated to a Swiss-Roll function in a three-dimensional (3D) space. Indeed, it is a synthetic dataset obtained as the partition in three sections of a discrete Swiss-Roll manifold depicted in a three-dimensional space. It reproduces the typical nonlinearity (given by the Swiss-Roll shape) and the discontinuity (given by the tripartition of the manifold), that we do not see and that are often hidden in the multidimensional representation of our samples. See the illustration

in the original 3D-space of the Tripartite-Swiss-Roll dataset in Fig. S1A. This dataset is useful to introduce readers, not expert with nonlinear data analysis, to the basic concepts of nonlinear dimension reduction and therefore to facilitate their understanding of the new proposed methodologies for nonlinear dimension reduction. **And (2) A Microbial-like dataset: a synthetic dataset generated with the R function rmvzinegbin from the library SpiecEasi [1] that simulates a microbial OTU table. The dataset contains four different groups with 50 samples each and 200 features. Two type of noise were introduced with the aim to recreate non-linearity, strategy already applied - with a different purpose - in the study by Lo & Marculescu [2]. The first type of noise introduces 0 elements to counts that are non-zero, whilst the second type introduces a deviation/noise from the true non-zero count.”**

1. Kurtz, Z. D. et al. Sparse and compositionally robust inference of microbial ecological networks. *PLoS Comput. Biol.* 11, e1004226 (2015).
2. Lo, C. & Marculescu, R. MetaNN: accurate classification of host phenotypes from metagenomic data using neural networks. *BMC Bioinformatics* 20, 314 (2019).

New text in the Results section of the main article:

“Indeed, the Tripartite-Swiss-Roll is purposely created to reproduce a manifold that is nonlinear and discontinuous (broken in three parts) such as the results of MCE analysis of Paroni Sterbini et al. seems to be. Furthermore, to compare the different approaches in a more “realistic” scenario, a synthetic microbial-like dataset (which resamples the nonlinearities encountered in the Paroni Sterbini et al. data) is generated and analysed. MCE overcomes once again the other dimensionality reduction techniques and is very close to guarantee a separability equivalent to the one obtained in the high dimensional space (HD) (Supplementary Table S4). These results are similar to the ones obtained in the real datasets. As expected, MDS with weighted Unifrac distance is highly affected by the fact that phylogenetic information between synthetic features is not available and it is directly extracted from the OTU table. Interestingly, and opposite to what already shown in the real dataset, MDS with Theta-YC distance obtains great performances close to MCE.”

#4 The authors used the Tripartite-Swiss-Roll dataset, but how (dis)similar is this dataset to the typical microbial community data structure? It is not because the obtained clusters match with external (clinical) data that this is a definite proof (but only supporting arguments) that the patterns are indeed real or biologically meaningful.

REPLY: We used the tripartite Swissroll dataset as a proof of concept dataset that (as the first Reviewer in general remarked for our study) is extremely useful to clarify the origin of nonlinearity and multi-group crowding problem in a clear, intuitive and visual fashion. Nonetheless, now we have generated a synthetic dataset with microbial-like community structure to further support our investigation of nonlinearity, and the manuscript was revised accordingly (please refer to the previous answer).

#5 Line 236. Unsupervised analyses are "particularly useful to discover the presence of interesting sub-groups inside the studied cohort or to detect the influence of confounding factors." This is important to state here that if groups are identified by unsupervised analyses, those newly defined groups cannot be tested for significance anymore in the same study using

other statistical tools. Otherwise, the procedure is akin to fishing for significance (i.e. you first identify a pattern in the data and then test for significance of the pattern using the same data). The null hypothesis is then more likely to be rejected. The hypothesis of the existence of those groups should be confirmed in a new experiment, independent from the one used for defining the patterns.

REPLY: Thanks to the Reviewer for this comment. First, in order to evaluate the extent to which the proposed procedure is ‘fishing for significance’, now (as explained in #2) the detected sample pattern is compared against a null distribution generated by exploring the space of patterns that can emerge at random. Second, we agree with the Reviewer that confirmation on independent data would be important. Indeed, already in the previous version of the manuscript, we confirmed some patterns detected in Paroni Sterbini data using two independent datasets: Amir3 and Amir4, and we were able to show that there is a shared microbial network that is behind the detected separation. We believe that these are solid evidences at support of the detected patterns. Finally, addressing one of the Reviewer’s concerns, we showed that the unsupervised performance of MCE on real data could be confirmed on a synthetic microbial-like dataset that resampled the characteristic of the Paroni Stebini data. Altogether, we believe that we provided enough confirmations to reassure the Reviewer on the quality of our investigation and derived results.

#6 Table S5. Lines 557-558, 712. It seems that the authors compared P values between different approaches, as a tool to compare better performing approaches. Yet, P values are not measure of support for their respective hypotheses (e.g. Schervish 1996 Amm Stat). Instead of comparing p-values one should compare effect sizes. Therefore, P values should not be used beyond the simple reject/do not reject decision. See also R.J. Grissom & J.J. Kim (2012): "Effect Sizes for Research", Routledge, Taylor & Francis Group, New York, London.

REPLY: Thanks to the Reviewer for such valuable comment. We removed the p-value tables for comparing performances between dimensionality reduction (DR) methods. However, we would like to stress that P-values could also work as a metric for comparison, which is supported by the high Pearson correlations that the P-values and AUCs obtain for the different datasets:
Paroni Sterbini: -0.91

Amir 3: -0.86

Amir 4: -0.91

ACTION: now, according to the Reviewer’s request, for the comparison of the sample size effect, we show only PSI-ROC and PSI-PR . Indeed, the AUC-ROC curve (which is used in PSI-ROC) is also known as the common language effect size measure, which represents the probability that a randomly chosen positive instance will rank higher than a randomly chosen negative one [1-4]. For the case of AUPR, this is a trade-off between effect size measures sensitivity (a.k.a recall) and precision (a.k.a positive predictive values) [4].

1. Kraemer, H.C. Effect size. *Enycl. Clin. Psychol.* (2014), 10.1002/9781118625392.wbecp048
2. Kraemer H.C., Kupfer D.J. Size of treatment effects and their importance to clinical research and practice. *Biol Psychiatry*.59(11):990-99616368078 (2006).
3. Fawcett, T. An Introduction to ROC Analysis. *Pattern Recognition Letters*, vol. 27, no. 8, 861-874 (2006).

4. Hentschke, H. & Stüttgen, M.C. Computation of measures of effect size for neuroscience data sets. *Eur. J. Neurosci.* 34, 1887–1894 (2011).

#7 PC-corr. The authors developed MC-MCL method to deal with non-linear patterns in community data, but then use a PC-corr network, which is (Line 482) "a method for linear multivariate-discriminative correlation network reverse engineering". Why not also using non-linear methods to identify features (taxa) that make the difference between the patterns or groups? Would other already existing methods to identify taxa responsible for group differences be also needed to be compared to here?

REPLY: Thanks for this opportunity to improve the text of our manuscript. The reply to this concern was already in the previous manuscript version, in the section "Network analysis clarifies the effect of PPI-treatment on the gastric microbiota", Pag 36:

"In Amir4 (gastric fluid), PCA revealed that gastric fluid samples were separated into two groups according to PPI treatment along PC2 and their difference is significant (p -value < 0.01) (Suppl. Figure S7). Hence, we built the PC-corr network⁶² using the loadings of PC2 at cut-off 0.5 (Suppl. Figure S8).

Similarly for the Paroni Sterbini dataset (gastric mucosa), PCA (Suppl. Figure S9) could (significantly or close to significance) separate PPI-treated *H. pylori*-negative patients from untreated *H. pylori*-negative patients along PC2 and PC15 (p-value along PC2 = 0.014, p-value along PC15=0.054). Therefore we built the PC-corr network for both PC2 and PC15 discriminating dimension using 0.5 cut-off (Suppl. Figure S10, panel A and B). "

ACTION: However, in the revised version of the manuscript, in order to be more explicit, we further clarify the reason to use PC-corr for the analysis also in the previous paragraph. Pag. 36:

"Up to this point, in order to assess the emergence of nonlinear patterns in data, the application and performance of linear and non-linear dimensionality reduction algorithms has been compared. Special focus was on Paroni Sterbini dataset, where the presence or absence of *H. pylori* infection in addition to the medical treatment (or not) with PPI medicaments created a complex nonlinear scenario difficult to disentangle using linear transformations and even some nonlinear ones. Then, with the didactic help of the Tripartite-Swiss-roll dataset, we clarified that the origin of the Paroni Sterbini nonlinearity stays in the three-body problem. Indeed, considering pairwise comparisons of only two groups, the nonlinearity vanished. Based on these considerations, now we conduct only the two-group comparison of PPI treated/nontreated patients in which presence of *H. pylori* was negative, since these data are available both in Paroni Sterbini and Amir. Such simplification of the investigation enables the application of the above mentioned PC-corr algorithm, since, for the binary class problem both Paroni Sterbini and Amir4 datasets present a significant segregation measured by MW-pvalue when embedded by the linear algorithm PCA." ...

And later, in the section "Network analysis clarifies the effect of *H. pylori* infection on gastric mucosal microbiota" Pag 42, we modified as follows:

"Similar to the PPI treatment network analysis in the previous section, in order to investigate the influence of *H. pylori* infection on the gastric mucosal microbiota by means of PC-corr, we analysed: 1) Paroni Sterbini *et al.*²² considering only PPI-untreated dyspeptic patients, either infected (H+) or not by *H. pylori* (H-); and 2) Parsons *et al.*²⁹ restricting to PPI-

untreated patients from: i) normal stomach group with no evidence of *H. pylori* infection; ii) *H. pylori* gastritis group with evidence of *H. pylori* infection.” ...

#8 Line 581. Did the authors check whether PCA axes beyond the first two axes could also indicate treatment-related structure in the data?

REPLY: Thanks for this comment. The reply is that they cannot indicate.

ACTION: In the revised manuscript, we added a supplementary table S1 showing the level of segregation (by means of Mann-Whitney p-values) for all the dimensions of each DR (dimension reduction) applied in this study. From there we can see that for the different DR algorithms, the first 2 dimensions are the ones that indicate treatment-related structure in the data. We added a line in the article referring to this in sections “Results”, Page 24:

“we focused on the first two dimensions of embedding **as they are significantly related with the treatment/infection response (Suppl. Table S4)**”

and “Comparison of unsupervised analysis in three gastro-esophageal datasets”, Page 32:

“Table 1, top panel, shows the best results in performance of unsupervised dimension reduction (PCA, MDSwUF, MDSbc, NMDS, MCE, for details see ‘Methods - PCA, MDS (or PCoA) and LDA’ and ‘Methods - Minimum Curvilinear Embedding (MCE)’) according to the PSI (projection-based separability index) based on ~~the p-value of Mann-Whitney U test~~, AUC and the AUPR, on the three different datasets (for more details on the PSI see ‘Methods - Procedure to evaluate the performance of the dimension reduction algorithms’). **Just the space of the first two dimensions of embedding were here used since they are the ones related with the treatment/infection –related structures (Suppl. Table S1).**” ...

Reviewer #3 (Remarks to the Author):

#1 Ciucci et al. compared different linear and non-linear dimensionality reduction methods on multiple microbiome datasets, followed by PCA-based correlation network analysis to clarify the effect of PPI-treatment. Though advanced methodologies are always welcome to analyze the highly dimensional, complex microbial data, this study is rather technical. The approaches used were not really novel and the test data sets were very small. This makes the biological interpretation less convincing. Thus, in terms of its technical aspects and limited biological advance, I think it does not fit to the general readers of this journal.

REPLY: Thanks to the Reviewer for the critical feedback of our study, that we consider a spur to offer more evidences in favour of the value of the methodology presented.

First, we wish to stress that our study is innovative at two different levels. At the more general ‘methodological level’, we introduce a new computational data mining pipeline (Fig.1 of the manuscript) which explains how to overcome the limits of current multivariate analysis of small-size microbial data. At the more specific ‘technical level’, we propose new solutions in each of the three steps that composes this pipeline: dimension reduction, clustering and PC-corr networks. In the dimension reduction part, we innovate by illustrating the benefits to apply minimum curvilinear nonlinear machine learning methods for dimension reduction. This is a completely new technical way to perform nonlinear analysis in the microbial field. In the clustering part, we propose MC-MCL, which is the first nonlinear version of Markov clustering and represents a novel technical approach that was priced by the Reviewer 4 in her/his

comments. In the PC-corr part, we show how to extract valuable and robust information (that would otherwise be missed using standard procedure of analysis) across several (4 in total) small-size microbial datasets.

As a matter of fact, the Reviewer has also concerns about the value of the biological findings that come from such a small cohort. Although, in the previous manuscript version, we compensated this limitation by identifying conserved bacterial networks (representing bacterial communities perturbed across different datasets), the Reviewer's critical feedback motivated us to expand the methodology we presented in Fig.1 of the manuscript, to further highlight its value and potential. Thus, now we propose a fourth and fifth step that help the biomedical interpretation and increase the impact of the findings for the scientific community.

ACTION: To address the Reviewer's concerns, we add in the revised version of the manuscript new results, text and figures showing the bacteria-metabolite networks with significant pathway information, which support the findings made by means of PC-corr analysis. Indeed, we add two new steps (see new Fig.1 below) to the methodology: 4th) on how to obtain the bacteria-metabolite network and 5th) on how to perform the metabolic network pathway analysis. The fact that we find significant metabolic pathways associated to the discriminative bacterial networks, which are detected by PC-corr, is a key nontrivial finding that we hope might contribute to change the perception of the Reviewer about the biomedical impact of our study.

Figure 1. Flowchart of the data analysis. To answer the five questions under investigation in our study, we implemented a workflow based on machine learning tools. Following the flowchart shown in the figure, we analysed three 16S rRNA gene sequencing datasets with information on PPI use in dyspeptic patients; for one of the datasets (Paroni Sterbini et al.²²), patients were also determined to be positive or negative to *H. pylori* infection. Firstly, we performed unsupervised dimension reduction, both linear and nonlinear, in the first two dimensions of embedding. Nonlinear dimension reduction will show the presence of hidden patterns, in the form of sample groups. Secondly, nonlinear clustering was applied to confirm the well-possedeness of the hidden patterns found by nonlinear dimension reduction. Furthermore, our workflow ends with the network analysis. It starts with the use of the PC-corr algorithm, that reveals which combination of bacteria (features) are responsible for the identified differences between the groups of samples. A fourth dataset (Parsons et al.³².) is used only for the validation of the PC-corr network results and it contains information of PPI

treatment and *H. pylori* infection. From the consensus bacteria found in each PC-corr network, a bacteria-metabolite multilayer analysis that lastly end with the metabolite pathway enrichment analysis that introduces evidence to possible perturbed biological mechanisms.

Figure 8. PPI-affected bacteria-metabolite network in gastric environment of dyspeptic patients. A) Multilayer (bacteria-metabolite) network representation: the first layer is derived from Fig.6 and represents the consensus network (confirmed in two datasets: gastric mucosa from Paroni Sterbini et al. ²² and gastric fluid from Amir et al. ²¹) with PPI-affected bacteria nodes that present information on metabolite interaction in [1]. The second layer represents the network whose nodes are the metabolites in [1] interacting with the bacteria network in the first layer; different node shapes and colours refer to different metabolite classes (carbohydrates, amino acids, glycolysis, amines, miscellaneous). B) In depth visualization of the bacteria-metabolite network interactions. The metabolites are grouped according to their involvement in significant pathways. For discernibility, the metabolites are arranged according to three significant pathways ($p < 0.05$ after Benjamini correction as result of a metabolite pathway enrichment analysis) and a fourth group that encloses altogether nodes associated to other significant pathways (please refer to the method section: Bacteria-metabolite multilayer network construction and metabolite pathway analysis); note that only metabolites present in

significant pathways are here displayed. For more information, please refer to figure S17 and table S18. The bacteria node stroke color is associated to the phyla information as in Figure 6, whereas the different colours in the inner fill are associated to the different pathways and their extent is proportional to the number of metabolites that the bacterium connects with in the different displayed pathways.

A

B

Figure 9. *H. pylori*-affected bacteria-metabolite network in gastric environment of dyspeptic patients. A) Multilayer (bacteria-metabolite) network representation: the first layer is derived from Fig.7 and represents the consensus network (confirmed in two different datasets of gastric mucosa: Paroni Sterbini et al. ²² and Parsons et al. ²) with *H. pylori*-affected bacteria nodes that present information on metabolite interaction in [1]. The second layer represents the network whose nodes are the metabolites in [1] interacting with the bacteria network in the first layer; different node shapes and colours refer to different metabolite classes (carbohydrates, amino acids, glycolysis, lipids, vitamins, miscellaneous). B) In depth visualization of the bacteria-metabolite network interactions. The metabolites are grouped according to their involvement in significant pathways. For discernibility, the metabolites are arranged according to three significant pathways ($p < 0.05$ after Benjamini correction as result of a metabolite pathway enrichment analysis) and a fourth group that encloses altogether nodes associated to other significant pathways (please refer to the method section: Bacteria-metabolite multilayer network construction and metabolite pathway analysis); note that only metabolites present in significant pathways are here displayed. For more information, please refer to figure S18 and table S19. The bacteria node stroke color is associated to the phyla information as in Figure 7, whereas the different colours in the inner fill are associated to the different pathways and their extent is proportional to the number of metabolites that the bacterium connects with in the different displayed pathways.

“Bacteria-metabolite multilayer network construction and metabolite pathway analysis

We used a recently realized bacteria-metabolite bipartite network which is an open access resource [1] to infer the metabolic activity of the bacteria presented in the intersections of figures 6 and 7. The study [1] provided a large set of 9136 bacteria to metabolite interactions validated on experimental studies from mouse and human gastroenteric microbiota. It was available as a network, named NJC19, where node represented either bacteria or metabolites connected by several types of edges (e.g. production, consumption, degradation). In this dataset we restricted the analysis to interactions found on human bacteria. Since the dataset identified bacteria according to the taxonomic levels of species while our data referred to the genus level, we made a new form of the NJC19 network with edges starting from the bacteria genus to metabolites. When we did not find any interactions for specific bacteria, we discarded them from further analyses. Therefore, from the list of intersected bacteria from figure 6 (*Porphyromonas*, *Capnocytophaga*, *Streptophyta*, *Granulicatella*, *Clostridiales*, *Oribacterium*, *Veillonella*, *Bulleidia*, *Fusobacterium*, *Leptotrichia*, *Campylobacter*, *Prevotella*) we dropped *Streptophyta*, *Granulicatella*, *Oribacterium*, *Bulleidia* and *Prevotella*. Similarly, from the intersected bacteria of figure 7 (*Enhydrobacter*, *Methylobacterium*, *Catonella*, *Pseudomonas*, *Acinetobacter*, *Sphingomonas*, *Propionibacterium*, *Bulleidia*) we dropped *Bulleidia*, *Catonella*, *Sphingomonas* and *Enhydrobacter*. For the graph representation, we used the color code already applied in the previous figures for the bacteria according to the taxonomic order. While for metabolites we classified them in 7 classes, assigning to each a different node shape and colour: vitamins, glycolysis, lipids, amino acids, carbohydrates, amines and miscellaneous. Furthermore, an enrichment analysis of metabolites (linked to the discriminative bacteria networks detected by PC-corr) has been conducted to unveil the metabolic pathways that might be associated to these bacteria perturbations. To this purpose, we used the framework provided by metaboloAnalyst suite [2]. Specifically, we performed the “Enrichment Analysis” against the KEGG database and we selected the significant pathways according to the Benjamini corrected p-values smaller than the significance level of 0.05. For the case of the *H. pylori*-affected network, just few nodes were available and only one significant pathway was obtained from it with few metabolite hits. Therefore, the network was expanded by adding first neighbours metabolites – obtained from KEGG – from the current ones.

Finally, the metabolite layer network nodes were grouped according to the three most significant pathways in both the PPI- and *H. pylori*-affected bacteria-metabolite networks. This was ensured according to the following procedure: a ranking was generated for the list of significant pathways in each of the two networks. Then, the nodes of each network were grouped according to the three pathways with the highest average ranking in the two networks, which in our study are: aminoacyl-tRNA biosynthesis; galactose metabolism; Alanine, aspartate and glutamate metabolism. A fourth group encapsulating the metabolites involved in the other significant pathways was also provided. Regarding the links considered in each bacteria and metabolite layer, the bacteria-bacteria associations were maintained from figures 6 and 7, whilst edges between metabolites were obtained by the metabolite interaction involved in the significant enriched KEGG pathways.

The processing pipeline has been developed in the R environment [3] and by using the following packages: *igraph* [4], *taxize* [5], *graphite* [6], *RCy3* [7].”

“Our study is innovative at two different levels. At the more general ‘methodological level’, we introduce a new computational data mining pipeline (Fig.1) which explains how to overcome the limits of current multivariate analysis of small-size microbial data. At the more specific ‘technical level’, we propose innovative solutions in each of the 5 steps that composes this pipeline: dimension reduction, clustering, PC-corr networks, multilayer bacteria-metabolite networks and metabolic network pathways analysis. In the dimension reduction section, we innovate by illustrating the benefits to apply minimum curvilinear nonlinear machine learning methods for dimension reduction. This is a completely new technical way to perform nonlinear analysis in the microbial field. In the clustering section, we propose MC-MCL, which is the first nonlinear version of Markov clustering and represents a novel nonlinear clustering approach. In the PC-corr section, we show how to extract valuable and robust information (that would otherwise be missed using standard procedure of analysis) across several (4 in total) small-size microbial datasets. In the fourth and fifth step we clarify how to enhance the biomedical interpretation with the aim to increase the impact of the findings on the scientific community.”

Results Page 46

“Bacteria-metabolite multilayer network analysis associates possible metabolic pathways perturbations

The relation between bacteria and metabolites is fundamental both to deepen the understanding of mechanisms associated to diseases dysfunction and drugs action, and to foster their biomedical interpretation [8-10]. For this reason, we made a quantum leap in our investigation from bacteria to metabolites and we built two bacteria-metabolite multilayer networks: one (Fig. 8) was derived from the PPI-affected bacteria network in Fig. 6, the other (Fig. 9) was derived from the *H. pylori*-affected bacteria network in Fig.7. The methodological procedure to build those multilayer networks is provided in the Methods (see section: Bacteria-metabolite multilayer network construction). Remarkably, by applying metabolic pathway enrichment analysis, we found that the metabolite layer of the PPI-affected (Fig.8B) and *H. pylori*-affected (Fig.9B) networks contain metabolites significantly involved ($p < 0.05$ after Benjamini correction) in important pathways (see full list in Suppl. Tables S18 and S19) associated with obesity [11], symptomatic atherosclerosis [12], functional dyspepsia [13], gestational diabetes mellitus [14], Wilson’s disease [15], among others. To simplify the visualization and interpretation (for the methodology of selection see Method section: Bacteria-metabolite multilayer network construction) we displayed the three most significant and relevant pathways in both PPI-affected (Fig.8B) and *H. pylori*-affected (Fig.9B) networks. Interestingly, the bacteria *Porphyromonas* and *Fusobacterium* are highly contributing for possible perturbations on the Aminoacyl-tRNA biosynthesis pathway for alterations produced by PPI (Figure 8B), whilst *Methylobacterium* does it on the *H. Pylori* infection side (Figure 9B). Besides, N-Acetylneuraminic acid (Suppl. Figure S17) is a sialic acid that has been associated also with pathogenic enteric bacteria [16,17] and tumors [18]. Overproduction of nitrites and nitrates by the observed anaerobic bacteria have been already observed in diverse parts of the gastrointestinal tract in patients suffering from migraine [19], intestinal dysbiosis and colorectal cancer [20,21], an effect already associated with the use of PPIs such as omeoprazole [22].”

Discussion

“Precisely, from the system point of view the obtained microbial differential networks pinpointed marked bacteria-bacteria interactions and modules affected by PPI treatment in the gastric environment in dyspepsia and by *H. pylori* infection in the gastric mucosa. **Moreover, we elucidated via bacteria-metabolite multilayer networks, possible metabolic alterations produced by the perturbed bacterial communities and the respective metabolic pathways involved in those changes. The fact that we find significant metabolic pathways associated to the discriminative bacterial networks, which are detected by PC-corr, is a nontrivial finding that suggests the reliability and impact of the integrated machine learning/network biology methodology we propose. However, some limitations frequently present in integrative systems biology also affect our study. For instance, when we adopt protein interaction networks in drug repositioning [23] or in disease analysis [24], we are aware that further information such as the contextualization of the network to the peculiar organ, tissue, cell or cell-compartment would allow more accurate results. The same is valid for our study, where we have to adopt a generic bacteria-metabolite gut network, because it is the most updated resource currently available in the field. This means that when – hopefully in future - more specialized bacteria-metabolite networks will be available for the gastric mucosa/fluid and even in specific areas of the stomach, then our analysis - such as many other omic analysis in integrative network biology - will benefit of this quantum leap in the data quality and contextualization. Hence, we suggest that our findings can be an important starting point to design new therapies that consider not only *H. pylori* infection but also the directly associated microbial alterations as well as the indirect alterations due to the drugs used for *H. pylori* eradication such as PPIs.”**

1. Roktaek Lim, Josephine Jill T. Cabatbat, Thomas L. P. Martin, Haneul Kim, Seunghyeon Kim, Jaeyun Sung, Cheol-Min Ghim & Pan-Jun Kim, Large-scale metabolic interaction network of the mouse and human gut microbiota. *Scientific Data* 7, 204 (2020).
2. Pang, Z., Chong, J., Li, S. and Xia, J. MetaboAnalystR 3.0: Toward an Optimized Workflow for Global Metabolomics. *Metabolites* 10(5) 186 (2020).
3. R Core Team (2017). R: A language and environment for statistical computing. R Foundation for Statistical Computing, Vienna, Austria. URL <https://www.R-project.org/>.
4. Csardi G, Nepusz T (2006). “The igraph software package for complex network research.” *InterJournal, Complex Systems*, 1695. <http://igraph.org>.
5. Scott Chamberlain, Eduard Szoecs, Zachary Foster, Zebulun Arendsee, Carl Boettiger, Karthik Ram, Ignasi Bartomeus, John Baumgartner, James O'Donnell, Jari Oksanen, Bastian Greshake Tzovaras, Philippe Marchand, Vinh Tran, Maëlle Salmon, Gaopeng Li, and Matthias Grenié. (2020) taxize: Taxonomic information from around the web. R package version 0.9.95. <https://github.com/ropensci/taxize>.
6. Gabriele Sales, Enrica Calura, Duccio Cavalieri & Chiara Romualdi, graphite - a Bioconductor package to convert pathway topology to gene network. *BMC Bioinformatics* volume 13, Article number: 20 (2012).
7. Gustavsen, A. J, Pai, Shraddha, Isserlin, Ruth, Demchak, Barry, Pico, R. A (2019). “RCy3: Network Biology using Cytoscape from within R.” *F1000Research*. doi: 10.12688/f1000research.20887.3
8. Amato, S. M. et al. The role of metabolism in bacterial persistence. *Front. Microbiol.* 5, 70 (2014). doi.org/10.3389/fmicb.2014.00070

9. Li Z, Quan G, Jiang X, et al. Effects of Metabolites Derived From Gut Microbiota and Hosts on Pathogens. *Front Cell Infect Microbiol.* 2018;8:314. (2018). doi:10.3389/fcimb.2018.00314
10. Vojinovic, D., Radjabzadeh, D., Kurilshikov, A. et al. Relationship between gut microbiota and circulating metabolites in population-based cohorts. *Nat Commun* 10, 5813 (2019). doi.org/10.1038/s41467-019-13721-1
11. Del Chierico F, Abbatini F, Russo A, Quagliariello A, Reddel S, Capoccia D, Caccamo R, Ginanni Corradini S, Nobili V, De Peppo F, Dallapiccola B, Leonetti F, Silecchia G, Putignani L. Gut Microbiota Markers in Obese Adolescent and Adult Patients: Age-Dependent Differential Patterns. *Front Microbiol* 9: 1210, 2018. doi:10.3389/fmicb.2018.01210.
12. Karlsson F.H., Fak F., Nookaew I., Tremaroli V., Fagerberg B., Petranovic D., Backhed F., Nielsen J. Symptomatic atherosclerosis is associated with an altered gut metagenome. *Nat. Commun.* 2012;3:1245. doi: 10.1038/ncomms2266
13. L. Luo, M. H. Hu, Y. Li, Y. X. Chen, S. B. Zhang, J. H. Chen, Y. Y. Wang, B. Y. Lu, Z. Y. Xie and Q. F. Liao, Association between metabolic profile and microbiomic changes in rats with functional dyspepsia. *RSC Adv.*, 2018, 8, 20166–20181
14. Ma, S.; You, Y.; Huang, L.; Long, S.; Zhang, J.; Guo, C.; Zhang, N.; Wu, X.; Xiao, Y.; Tan, H. Alterations in Gut Microbiota of Gestational Diabetes Patients During the First Trimester of Pregnancy. *Front. Cell. Infect. Microbiol.* 2020, 10, 58
15. Cai, X., Deng, L., Ma, X. et al. Altered diversity and composition of gut microbiota in Wilson's disease. *Sci Rep* 10, 21825 (2020). doi.org/10.1038/s41598-020-78988-7
16. Severi E, Hood DW, Thomas GH. "Sialic acid utilization by bacterial pathogens". *Microbiology.* 153 (9): 2817–2822. doi:10.1099/mic.0.2007/009480-0 (2007).
17. Vimr ER, Kalivoda KA, Deszo EL, Steenbergen SM. "Diversity of microbial sialic acid metabolism". *Microbiol Mol Biol Rev.* 68 (1): 132–153. doi:10.1128/mubr.68.1.132-153.2004 (2004).
18. Xiaoman Zhou, Ganglong Yang and Feng Guan. Biological Functions and Analytical Strategies of Sialic Acids in Tumor. *Cells* 2020, 9(2), 273. doi.org/10.3390/cells9020273.
19. Gonzalez A, Hyde E, Sangwan N, Gilbert JA, Viirre E, Knight R. 2016. Migraines are correlated with higher levels of nitrate-, nitrite-, and nitric oxide-reducing oral microbes in the American Gut Project Cohort. *mSystems* 1(5):e00105-16. doi:10.1128/mSystems.00105-16.
20. Jun Kobayashi, Effect of diet and gut environment on the gastrointestinal formation of N-nitroso compounds: A review. *Nitric Oxide*, 73, 66-73 (2018).
21. Roisin Hughes, Ian R. Rowland, Metabolic Activities of the Gut Microflora in Relation to Cancer. *Microbial Ecology in Health and Disease*, 12(2), 179-185 (2000).
22. Verdu E, Viani F, Armstrong D, et al. Effect of omeprazole on intragastric bacterial counts, nitrates, nitrites, and N-nitroso compounds. *Gut.* 35(4):455-460. doi:10.1136/gut.35.4.455 (1994).
23. Claudio Durán, Simone Daminelli, Josephine M Thomas, V Joachim Haupt, Michael Schroeder, Carlo Vittorio Cannistraci, Pioneering topological methods for network-based drug–target prediction by exploiting a brain-network self-organization theory, *Briefings in Bioinformatics*, Volume 19, Issue 6, (2018), Pages 1183–1202.

24. Muscoloni, A.; Abdelhamid, I.; Decano, J.L.; Souza, E.; Maiorino, E.; Aikawa, M.; Silverman, E.K.; Sharma, A.; Cannistraci, C.V. Hyperedge Entanglement in High-Order Multilayer Networks. *Preprints* 2020, 2020120500.

Major comments:

#2 The authors applied different dimensionality reduction methods, including linear approaches (PCA, MDSbc) and nonlinear approaches (MDSwUF, NMDS, MCE). The first major conclusion is that MCE outperforms other methods and linear technique will fail to bring out the patterns in the microbial datasets related to PPI-treatment. Firstly, based on Fig 3 and Fig 4, none of these methods separated PPI-treated and nontreated groups, even for MCE.

REPLY: Thanks to the Reviewer for this comment, which can help us to clarify why MCE indeed separated PPI-treated and non-treated groups. Actually, the different DR methods were able to separate PPI-treated (blue points) and non-treated H positive groups (green points) but with difficulties to discern PPI-treated (blue points) and non-treated H negatives (red points). The only method that was able to achieve the separation of all these groups was MCE (see figure below). In particular, MCE was able to discern PPI-treated (blue points) and non-treated H negatives (red points) along the second dimension.

ACTION: To evidence it better, in the new revised manuscript, we introduce a density plot of the three groups on the right side of the MCE figure 3. The current arrows on the left side are used only for this reply to the Reviewers to stress the pics of density at each group. In addition, as you can notice, all three groups are clearly separated in the 2D space.

Figure 3. MCE, a topological machine learning for nonlinear and hierarchical dimension reduction. A) Results on the Paroni Sterbini et al.²² dataset. The shown best MCE result is based on PSI-PR projection-based separability index (PSI) for the three different labels (P-treated, untreated H+ and untreated H-), evaluated in the 2D embedding space under the DCS normalization. The PSI-ROC and PSI-PR are reported as overall estimators of separation between the groups in the 2D reduced space. Blue dots represent PPI-treated samples, while red and green dots are the untreated samples which resulted either negative (red) or positive (green) to the *H. pylori* test (histological observation and urease test). B) The curves in three

different colours (red, blue and green) highlight the different distributions of the three groups on the second dimension.

The clustering algorithms also confirm the three cluster arrangement, where their performances increase when detecting three clusters instead of four as discussed in the paper Page 30:

“In addition, the hypothesis of three clusters seems more congruous than four clusters, because both MC-MCL and MCL decrease their accuracies in detecting four clusters.”

#3 The presence of *H. pylori* drives the difference of the microbial community, instead of PPI treatment. This is also confirmed by the PCA plot (Fig3A) that a couple of PPI+ samples (with *H. pylori*) are closer to HP+ samples, while PPI- (without HP) were more closer to HP-.

REPLY: There is a significant segregation (P-value < 0.05) for H+ vs H- samples considering treated and non-treated cases for the PCA plot (P-value: 0.004) and also for the MCE plot (P-value: 0.017). Nonetheless, these results do not imply what the Reviewer suggests: “the presence of *H. pylori* drives the difference of the microbial community, instead of PPI treatment”. Instead, this segregation arises easily due to the already clear segregation of the non-treated PPI groups with H+ and H- (red and green points). Indeed, if what the Reviewer suggests were the case, then the segregation between H+ and H- samples would be evident as well inside the PPI treated group. However, the P-values are not anymore significant for this case (P-value PCA: 0.46 & P-value MCE: 1) and no evident segregation arises neither by eyes, already supported by the Supplementary Figure S6.

ACTION: we now introduce in the revised manuscript this explanation in result section, Page 31:

“It might be argued that the presence of *H. pylori* only drives the difference of the microbial community, instead of PPI treatment. However, if this were the case, then the segregation between H+ and H- samples would be evident as well inside the PPI treated group. However, the P-values are not anymore significant for this case (P-value PCA: 0.46 & P-value MCE: 1) and no evident segregation arises neither by eyes, as supported by the Suppl. Figure S6.”

#4 Notable, there were only 4 PPI+ patients. MCE (Fig 4) indeed yielded a better separation between PPI treated and HP- groups, suggesting the microbial shift towards HP+ like community. The improvement is marginal: AUC increased from 0.92 (PCA, MDSuWF and NMDS) to 0.967 (MCE).

REPLY: Thanks to this concern of the Reviewer, we realized that the formula used to evaluate the overall PSI indices (as average of the pairwise) could be improved considering not only the average but also the dispersion. This would allow to reward methods that offer robust separation across all pairwise estimations.

We designed a new formula to compute the PSI measures when evaluating more than two groups. In general, when certain groups are overlapped and there is one group clearly segregated from the rest, this solitary group will increase the PSI value since it obtains a perfect segregation from the rest of the groups, giving a boost to the final PSI performance. In order to address this, instead of computing the final PSI value as a simple mean of the pairwise-group PSIs, now also the standard deviation of the pairwise-group PSI values is taken into account. The new formula is: $E/(1+\delta)$, where E is the mean of the pairwise PSI values and δ their standard deviation. Thus, the standard deviation works as a penalization in case of outliers PSI values, ensuring that the

overall PSI is high only when all pairwise PSI values are close to the mean. This means that a good embedding should provide homogeneous separation of all group pairs, which is indeed what we want to quantify with a global (in the sense that considers all group pairs) measure.

All Paroni Sterbini PSI performances are now updated through the manuscript and the methodology of the PSI section was accordingly updated. As a consequence, now the new AUC is 0.85 for PCA and NMDS and 0.91 for MCE. We believe that these new values appropriately quantify as performance what we visually perceive looking at the 2D embedding of the respective DR methods.

However, let us pose by hypothesis that the Reviewer considers “marginal” also this difference of AUC (from 0.85 to 0.91) detected using the new PSI formula. We want to stress that in general offering an AUC-ROC result that is higher than 0.9 is considered relevant in all scientific literature. Furthermore, as suggested by Ammirati et al., the same level of increase becomes more significant when being close to perfect segregation. This becomes evident when “quantifying the improvement in terms of the distance from the exact predictor”. As a didactic example, let us compare the current improvement of 0.06 (0.85 – 0.91) against a case with the same improvement but closer to randomness (0.50 – 0.56). In the former the relative improvement in respect to the exact predictor is 40% (computed as $(0.91-0.85)/(1-0.85)*100$), whereas in the latter is 12% (computed as $(0.56-0.50)/(1-0.50)*100$).

ACTION: We now introduce these notions and respective results in the revised manuscript, in the Result section “Gastric tissue dataset unsupervised analysis” Page 26:

“Non-centred MCE (Figure 4A, DCS normalization) was the best performing technique, with a **PSI-ROC of 0.91 and PSI-PR of 0.96** (Table 1) (for details see Suppl. Table S2). It even outperforms the nonlinear methods NMDS (Sammon Mapping) and MDSwUF, since MCE is automatically able to **unsupervisedly infer from data the underlying** (hierarchical) phylogenetic relationship among the bacteria. **MCE does not receive in input any phylogenetic information but directly infers it** from the bacterial abundance of the dataset by performing a hierarchical embedding, as already shown in the study of Alanis-Lobato et al.³⁹ (see ‘*Methods- MCE to unsupervisedly infer and visualize phylogenetic (hierarchical) relations*’). **The gain in performance compared with the rest of the dimensionality reduction techniques is relevant.**

Indeed, the PSI-ROC improvement from 0.85 (PCA and NMDS) to 0.91 is not trivial. We want to stress that in general offering an AUC-ROC result that is higher than 0.9 is considered relevant in all scientific literature. Furthermore, as suggested by Ammirati et al., the same level of increase becomes more significant when being close to perfect segregation. This becomes evident when “quantifying the improvement in terms of the distance from the exact predictor”. As a didactic example, let us compare the current PSI-AUC improvement of 0.06 (0.85 – 0.91) against a case with a same hypothetical improvement but closer to randomness (0.50 – 0.56). In the former the relative improvement in respect to the exact predictor is 40% (computed as $(0.91-0.85)/(1-0.85)*100$), whereas in the latter is 12% (computed as $(0.56-0.50)/(1-0.50)*100$). Similarly for PSI-PR, MCE (PSI-PR=0.96) relative improvement from PCA (PSI-PR=0.91) in respect to the perfect predictor is 56% (computed as $(0.96-0.91)/(1-0.91)*100$)”

1. E Ammirati, CV Cannistraci, NA Cristell, V Vecchio, AG Palini, P Tornvall. Identification and Predictive Value of Interleukin-6⁺ Interleukin-10⁺ and Interleukin-6⁻ Interleukin-

10⁺ Cytokine Patterns in ST-Elevation Acute Myocardial Infarction. *Circulation research* 111 (10), 1336-1348 (2012).

#5 and P-value decreased from 0.009 to 0.004. This could be due to the small sample size. Moreover, MCE used different distance matrix, combined with polygenetic distance.

REPLY: We are sorry for this misunderstanding. First, as for request of Reviewer 2 we now omit the PSI P-value performances in the revised manuscript. In any case, the new PSI formula (please refer to the answer of the previous comment) works just for PSI metrics where the maximum possible value (perfect segregation) is 1. Second, the statement of the Reviewer “Moreover, MCE used different distance matrix, combined with polygenetic distance” is never stated in the manuscript because it is not true. Certainly, we compare MCE (which infers hidden phylogenetic information directly from data) with MDS weighted Unifrac, which indeed uses a phylogenetic tree to compute the distances. What we stated in the previous manuscript version Page 26 is that MCE is able to (unsupervisedly) infer hierarchy (thanks to the MST) and therefore reconstruct the phylogenetic relationship among the bacteria.

ACTION: We now rephrase this paragraph to make it more understandable for the audience. “..., since it is automatically able to **unsupervisedly infer from data the underlying** (hierarchical) phylogenetic relationship among the bacteria. **MCE does not receive in input any phylogenetic information but directly infers it** from the bacterial abundance of the dataset by performing a hierarchical embedding, as already shown in the study of Alanis-Lobato et al. ⁴¹ (see ‘*Methods- MCE to unsupervisedly infer and visualize phylogenetic (hierarchical) relations*’).”

#6 Thus it is not conclusive that the observed improvement is due to dimensionality reduction algorithm or due to different distance calculation. Authors should use the same distance matrix to compare the performance of different methods.

REPLY: As we explain in the reply to the previous comment, we indeed use for MCE and all other dimension reduction techniques non-phylogenetic-driven distance matrices obtained directly from bacteria abundance (OTU tables), with the only exception of MDSwUF that - by definition -requires the integration of phylogenetic information. Hence, the gain in performance of MCE is clearly due to the algorithm itself.

#7 Moreover, there are many other well-known non-linear dimensionality reduction methods, such as t-SNE. How does t-SNE perform?

REPLY: Our study is on nonlinear parameter-free unsupervised dimension reduction. This is not a case, but the result of a specific choice as we reported already in the previous manuscript version. We believe that advanced nonlinear data analysis needs adaptiveness and automatization, whereas methods such as t-SNE and Isomap are ‘sponsored’ as unsupervised but, in reality, nobody clarifies how to tune the parameters that are 2 in case of t-SNE and 1 for Isomap. This is the reason why they were considered out of the scope of the current study, because in particular with 2 parameters to tune in small size data, overfitting is a relevant issue. We do not want to contradict the Reviewer, but we are not aware of any ‘oracle’ algorithm that can suggest how to tune two parameters (without using the labels) preventing overfitting in small size data. Actually, to the best of our knowledge, there is not yet any conventional or commonly accepted solution for it.

ACTION: To address the Reviewer’s request, in the revised version of the study, we perform the dimension reduction analysis also introducing t-SNE and Isomap. We apply a supervised procedure in which the labels are used to supervisedly tune the internal parameters of these methods and we select the solution which offers the best performance. The results are shown hereunder for supplementary figure S3. Within these results, MCE outperforms Isomap and t-SNE for both PSI-ROC and PSI-PR values. We now discuss these two non-linear methods in the introduction and mentioned this figure in the results section.

Figure S3. Nonlinear dimension reduction by t-SNE and Isomap applied to the Paroni Sterbini dataset. The plots report the best t-SNE and Isomap results based on PSI-ROC and PSI-PR for the separation of the three different groups (PPI-treated [blue points], untreated H+ [green points] and untreated H- [red points]), evaluated in the 2D embedding space. A) Best t-SNE embedding obtained with parameters dimension 19 and perplexity 14; B) Best Isomap embedding obtained with parameter k nearest neighbors of 21. Note that the sample labels were used in order to supervised select the best input parameters for each algorithm.

New paragraph in the Introduction section:

“This procedure is particularly useful in case of studies with small size datasets ²³, or for **imbalance class samples**, to obtain unbiased (the labels are not used) confirmation of the separation between groups of samples for which diversity is theorized or expected.

In addition, we will provide an analysis with two nonlinear algorithms for dimensionality reduction often used in literature, namely Isomap [1] and t-SNE [2,3]. These methods, although unsupervised, need hyperparameters optimization. Indeed, Isomap needs as input a parameter related to ‘k’ number of neighbours to construct a network, whereas t-SNE needs the perplexity and number of dimensions (or components). Different values of these parameters may lead to different results, which represent a challenge in an unsupervised scenario where automatic and label-free selection of the best solution is wished. This is the reason why this study will focus mainly on parameter-free dimensionality reduction techniques, whereas Isomap and t-SNE results will be shortly

considered for a specific dataset in the result section.”

New paragraph in the Results section, Page 28:

“ Thus, MCE provides an ordering of the groups along the second dimension that is related to pH increment (from H- to P&H+). Furthermore, we contrast MCE performance on this challenging dataset versus two baseline algorithms for nonlinear dimension reduction: t-SNE and Isomap. As we stressed in the introduction these algorithms require optimal tuning of parameters (two for t-SNE and one for Isomap). We believe that advanced nonlinear data analysis needs adaptiveness and automatization, whereas methods such as t-SNE and Isomap, although in principle are unsupervised, in practice are applied in a supervised manner and the hypothesized class labels are used to learn their best parameter tuning. Unlikely, in small size datasets, parameter tuning is a relevant issue that may cause overfitting, especially with more than one parameter such as in the case of t-SNE and, to the best of our knowledge, there is not yet any commonly accepted solution for this. Here, with the mere intention to provide a proof of concept that allows to compare MCE with other nonlinear dimension reduction methods, we apply a supervised procedure in which the labels are used to supervisedly tune the internal parameters of these methods and we select the solution which offers the best performance. The results are shown in Suppl. figure S3. t-SNE (PSI-ROC: 0.90, PSI-PR: 0.94) and Isomap (PSI-ROC: 0.87, PSI-PR: 0.94) performances are lower than MCE performances, displaying difficulty to resolve the difference between treated and untreated samples, mostly for the cases of treated patients (blue points) and untreated patients without *H. pylori* infection (red points). This indicates that in principle adaptive parameter-free algorithms such as MCE may also outperform more complex algorithms under difficult scenarios such as for this particular case.”

1. J. B. Tenenbaum, V. de Silva, and J. C. Langford, “A global geometric framework for nonlinear dimensionality reduction.,” *Science*, vol. 290, pp. 2319–23, 2000.
2. K. Bunte, S. Haase, M. Biehl, and T. Villmann, “Stochastic neighbor embedding (SNE) for dimension reduction and visualization using arbitrary divergences,” *Neurocomputing*, vol. 90, pp. 23–45, Aug. 2012.
3. L. van der Maaten and G. Hinton, “Visualizing Data using t-SNE,” *J. Mach. Learn. Res.*, vol. 9, no. Nov, pp. 2579–2605, 2008.

#8 Nevertheless, the authors had concluded that non-linear based MCE is better.

REPLY: Giving all the results that we provide, including now nonlinear techniques such as t-SNE and Isomap for Paroni Sterbini, we cannot conclude the opposite. In any case, the indices are higher for MCE. However, even in a hypothetical scenario that MCE offered comparable or slightly lower results than t-SNE and Isomap, you should take in consideration that MCE is a parameter-free algorithm, therefore its performance would be impressive, since the other nonlinear techniques require parameter tuning and MCE not.

#9 It puzzles me that authors then chose to use PCA-based correlation methods for network analysis, which is linear-based method.

REPLY: We thank the Reviewer for this comment. However, the reply was already addressed in the previous version of the manuscript in the Results section “Gastric tissue dataset unsupervised analysis”, Page 29. Indeed, we extensively provided examples by reproducing the nonlinear effect with the tripartite swiss-roll in order to point out what was the reason of nonlinearity. As demonstrated, the nonlinearity was raised in Paroni Sterbini when analysing the three clusters problem, but for binary cluster segregation, a linear approach was more than enough (Figure 4).

“Moreover, because of the discovered major nonlinear complexity in the Paroni Sterbini gastric biopsy dataset, we wanted to verify whether it was generated by multi-grouping (three-body interaction problem associated to the presence of three hidden clusters). To do so, we applied PCA to three subsampled versions of the dataset (with the best normalization originally found for the complete dataset), each corresponding to the combination of two groups (Fig. 4A-C), and PCA could find significant separation (p-values <0.02 and AUC, AUPR > 0.80). To further confirm that the presence of multiple sample groups generates the data complexity, we did the same for the Tripartite Swiss-Roll (Fig. S5A-C), where we recovered the discrimination, even though two comparisons overlap to some extent (Fig. S5A and C). Furthermore, to have another confirmation that the PPI-treated samples are not separable for *H. pylori* infection, we analysed the dataset considering exclusively the PPI-treated samples. The result is that no internal separation related to *H. pylori* infection emerges within the PPI-treated patients, as shown by the best MCE result (Suppl. Figure S6).”

Hence, this justifies the use of PC-corr algorithm in the study. Nevertheless, we also expanded the text in the Results section to provide further clarifications. Please refer to the answer #7 of the Reviewer 2.

#10 It is not clearly described how this correlation network reflects the microbial changes due to PPI treatment.

REPLY: This concern was addressed in two sections of the article. First in the methods section “PC-corr network”. Then in the result section “Network analysis clarifies the effect of PPI-treatment on the gastric microbiota” in Page 37. For the first scenario we explain that PC-corr is not a correlation network but rather a method for linear multivariate-discriminative correlation network inference. Then, for the second case, we address how we got the microbial changes due to PPI from two datasets.

ACTION: However, thanks to the Reviewer’s comment now we improve the text in the manuscript. Page 37:

“Subsequently, to investigate how PPI is affecting the microbiota in the gastric environment, we considered the conserved **PC-corr** network **as an indication of bacteria behavior robustness**. It is obtained as the union of the two PC-corr networks (obtained for PC2 and PC15) derived from the Paroni Sterbini gastric mucosa dataset intersected with the PC-corr network derived from the Amir4 gastric fluid dataset. The resulting conserved network displays the bacteria with same trend in the two datasets, i.e. either increased or decreased **abundance for patients** with PPI-treatment, respectively in red and black colour, as emphasized by the violet circle at the centre of Figure 5. Figure 6 is the same as Figure 5 but here the nodes are coloured according to phylum-level taxonomy. The conserved network which arises at the overlap between the two PC-corr networks (union of Paroni Sterbini networks intersected with

the Amir4 network) is statistically significant (p -value=1.00e-04), as a result of the statistical test based on trying to obtain the same conserved network by random resampling the bacteria in the two networks (Suppl. Figure S11), implying the difficulty of generating this intersection simply at random (since this intersection lies to the right of the critical value at the 0.05 level in the distribution of overlap)”

Other specific comments:

#11 The method section is way too long and not in a good order. Line 227: The part of “Data exploration and visualization” should be moved down, together with the paragraph of PAA, MDS and LDA (Line 270).

REPLY: Thanks to the Reviewer for his comment. We actually now re-ordered the manuscript accordingly. We even moved the sections with Swiss-roll dataset to the supplementary information.

#12 Line 387: the polygenetic information is instead directly inferred from bacterial abundance, differently from MDSwUF. I don’t understand this. How polygenetic information was calculated from bacterial abundance? How distance was calculated.

REPLY: MCE can automatically infer the phylogenetic abundance because it is a hierarchical method. Please refer to our reply to your comment #5 provided above.

#13 Line 436: put zero to the negative values. How many negative values were generated and were replaced by zero. What is the impact on the analysis?

REPLY: For the clustering approaches, the idea of putting the negative values to 0 is because those nodes in the network are actually even more different (anti-similar) than what would we find at random (correlation 0). Indeed, any correlation that is lower than 0 tends to overcome an intrinsic threshold of random similarity, hence correlations lower than 0 can be interpreted as less significant than random. This is a procedure commonly used to implement MCL [1].

1. Stijn van Dongen, Graph Clustering by Flow Simulation, PhD thesis, University of Utrecht, May 2000. (<http://www.library.uu.nl/digiarchief/dip/diss/1895620/inhoud.htm>)

#14 Labels of PPI+ and PPI- in the figures are very confusing to readers. They seem to refer PPI treated (PPI+) and non-treated group (PPI-). Actually, they refer to PPI treated group with and without H. pylori respectively, while HP+ and HP- were PPI non-treated group. The authors choose clearer and self-explaining labels.

REPLY: Thanks to the Reviewer for his concern. We now changed the samples labels for the following ones:

PPI+ → P&H+

PPI- → P&H-

HP+ → H+

HP- → H-

Reviewer #4 (Remarks to the Author):

Summary:

In the manuscript by Ciucci and colleagues, the authors sought to utilize conventional unsupervised machine learning techniques in conjunction with more nonlinear analysis techniques to understand the responses of the gastric microbiota to specific perturbations. The group created and validated a nonlinear clustering technique combining the weighted graphs of Markov clustering with the network embedding properties of Curvilinear Embedding. The approaches presented here represent a much needed evolution in the standard analyses of microbial community data as intuitively many interactions within these consortia will likely not follow linear relationships. Generally speaking, their arguments are well justified and supported, however several areas of improvement are present. The authors spend large amounts of text on simply demonstrative examples of principles that do not contribute to the central theme of validating their MC-MCL approach and presenting novel findings about the structure of the gastric microbiota during perturbation. The authors also incorrectly use certain vocabulary that is common within the microbiome field. This usage could lead to incorrect assumptions about the biology they are capturing in their models, or an incomplete understanding of the biological assumptions inherent to some of the computational techniques employed here. These and other issues are discussed in more detail below.

REPLY: We thank the Reviewer for evaluating that the approaches presented in our study represent a much needed evolution in the standard analyses of microbial community data. This was one of the main intentions of our study and we feel rewarded that the Reviewer recognized it. Meanwhile, the Reviewer noted that several areas of improvement are present, and we are convinced that her/his concerns will help to improve our study.

Major Concerns:

#1 Metagenomes cannot be obtained from 16S rRNA gene amplicon sequencing alone. The term metagenome refers to the collective gene content of all organisms found within the environment of interest. This vocabulary must be corrected as true metagenomic studies require more robust sequence curation and analysis pipelines for handling assembly/annotation for large amounts of shotgun read data. Failing to make this distinction is dishonest to the form and interpretation of analyses performed in this study, and will mislead readers who are more familiar with these terms.

REPLY: We thank the Reviewer for this important comment. It is also our interest that any reader understands that our metagenomic data refer to 16S rRNA amplicon sequencing and not to shotgun sequencing. We want to reassure the Reviewer that there was not any dishonest purpose and that although one of the biomedical Authors raised this issue during internal article revision, since the manuscript was finally edited by computational people and we should integrate the corrections of many experts, we corrected this terminology in some points, but we neglected to correct it in some other points of the text. We commit to pay more attention during this revision process.

ACTION:

We modified the text accordingly, using now the term 16S rRNA amplicon.

#2 The authors spend a great deal of time on demonstrating where older approaches fail (PCA, MDS, LDA) with test datasets, and I wonder why they just didn't show that they fail on datasets where MC-MCL succeeds toward the second half of the results. Additionally, I understand the value of introducing the Swiss-Roll dataset as a lesson for readers, but it seems that the subsequent figures that compare both types of approaches more than demonstrates the point the authors are trying to make. Overall, shortening the results section by eliminating (or greatly reducing) the primarily proof-of-concept examples may benefit the general readability of the final manuscript.

REPLY: Thanks for this comment. We moved the proof-of-concept examples to the Suppl. Info. However, this part is still important for instance to reply to questions such as the one of Reviewer 3, because it explains a way in which nonlinearity can arise in data.

#3 It is confusing why they would introduce the PC-corr method at all in the current study, as it seems they should be focusing energy on validating their new approach. This appears to be something that should be an entire separate future study and removed from the current text entirely.

REPLY: thanks to the Reviewer for this concern, that indirectly points towards a common dilemma in interdisciplinary research. While from the mere computational standpoint we agree with the Reviewer suggestion, from the computational biomedicine standpoint we still believe that the PC-corr analysis is fundamental in the study, since it gives the insights on: 1) the microbial differentiation between patients with and without PPI-treatment relatedness; and 2) which bacteria community are affected by the presence of *H. pylori* infection.

These types of analyses can result in an extraordinary gain in knowledge and understanding of the biological mechanisms behind the perturbations. Indeed, thanks to the PC-corr analysis, we are now able to include in the revised study two bacteria-metabolite interaction network (see reply #1 to Reviewer 3, and the new Figure 8, Figure 9, Supplementary Figure S17, Supplementary Figure S18 and Supplementary Table S18 and S19). Thanks to it, we can now visualize the dynamics of the compounds produced/consumed/degraded by bacteria and their role in certain pathways. The text in the manuscript has been accordingly modified (please refer to the reply #1 to Reviewer 3).

#4 Line 360: Pearson correlation assumes constant variance and incorrect usage may result in overcalling correlation results. I would think that Spearman would be more appropriate here.

REPLY: Thanks to the Reviewer for the nice suggestion. We have added now MCE using Spearman correlation. Nevertheless, the results are maintained or worse than what we already found by means of Pearson correlation.

ACTION: We have added all Spearman-based values in the supplementary information table S2.

#5 Line 372: Weighted Unifrac distance, while used in many studies, incorrectly assumes phylogenetic relatedness of bacteria simply by similarity of the current section of the 16S gene being analyzed. This disregards the possibility of convergent evolution of sequences, and thus fails more frequently the less well-characterized the environment of interest is. A more accurate, but still Euclidean, algorithm would be Theta-YC.

Yue & Clayton. (2005). A Similarity Measure Based on Species Proportions. *Communication in Statistics- Theory and Methods* 34(11):2123-2131.

REPLY: Thanks for the Reviewer's suggestion. The Theta YC distance obtains better performances than the Bray-Curtis approach but – in this study – lower than the weighted Unifrac distance.

ACTION: We include and compare now in our study the MDS Theta-YC against the already discussed approaches (please see below the updated Table 1 and Table 2).

Table 1. Results of unsupervised analysis on the real datasets. Best results of unsupervised dimension reduction techniques (top panel) and of clustering (bottom panel).

(Top panel): Best results of unsupervised dimension reduction techniques according to the PSI indices for sample separation in the space of the first two dimensions of embedding. HD (no dimension reduction) represents the reference results to see how good the separability present in the high dimensional space is preserved by dimension reduction techniques. Results are ordered from the best (top) to the worst (bottom) method. For the Paroni Sterbini dataset, we show the results for three different labels (PPI-treated, untreated H+ and untreated H-). For the Amir datasets, the PSI measures were computed for two groups, identified by the presence or absence of PPI treatment. For each PSI value, a respective trustworthiness was calculated.

(Bottom panel): Best results of clustering (highest accuracies, regardless of the normalization and type of correlation) MCL and MC-MCL, in each of the three studied datasets (Paroni Sterbini, Amir3 and Amir4), and the mean performance (mean of the highest accuracies) across all the datasets.

For Paroni Sterbini dataset, we show the results for three clusters (PPI-treated, untreated H+ and untreated H-) and in brackets the results for four clusters (P&H+, P&H-, untreated H+ and untreated H-). Instead, for Amir datasets, the accuracies were computed for two groups,

identified according to the presence or absence of PPI treatment.

Dimension Reduction	PSI-ROC							
	Method	Paroni Sterbini	Trust	Amir3	Trust	Amir4	Trust	mean
	HD	0.88	0.0036	0.95	0.0009	0.98	0.0009	0.94
	MDSwUF	0.84	0.0089	1.00	0.0009	0.88	0.0329	0.90
	MCE	0.91	0.0036	0.88	0.0329	0.91	0.0009	0.90
	PCA	0.85	0.0063	0.91	0.0009	0.86	0.0169	0.87
	MDStyc	0.84	0.0076	0.88	0.0009	0.84	0.0249	0.85
	nMDS	0.85	0.0036	0.86	0.0169	0.84	0.0089	0.85
	MDSbc	0.81	0.0183	0.86	0.0089	0.84	0.0189	0.84
PSI-PR								
Method	Paroni Sterbini	Trust	Amir3	Trust	Amir4	Trust	mean	
HD	0.94	0.0009	0.96	0.0009	0.99	0.0009	0.96	
MDSwUF	0.88	0.0036	1.00	0.0009	0.90	0.0089	0.93	
MCE	0.96	0.0009	0.89	0.0089	0.92	0.0039	0.92	
PCA	0.91	0.0039	0.90	0.0009	0.88	0.0089	0.90	
MDStyc	0.88	0.0116	0.90	0.0009	0.88	0.0089	0.89	
MDSbc	0.86	0.0116	0.89	0.0009	0.90	0.0009	0.88	
nMDS	0.90	0.0036	0.87	0.0089	0.87	0.0009	0.88	
Clustering	Accuracy	Paroni Sterbini	Amir3	Amir4	Mean performance			
	MC-MCL	0.71 (0.58)	0.81	0.75	0.76			
	MCL	0.67 (0.63)	0.69	0.75	0.70			

Note: all the P-values, AUC and AUPR can be found in Supplementary Table S2, while all the accuracies can be found in Supplementary Table S17.

Abbreviations: HD: High Dimension; MCE: Minimum Curvilinear Embedding; MDSbc: Multidimensional Scaling with Bray-Curtis dissimilarity; MDSwUF: Multidimensional Scaling with weighted UniFrac distance; NMDS: Non-metric Multidimensional Scaling; MDStyc: Multidimensional Scaling with Theta-YC distance; PCA: Principal Component Analysis; MCL: Markov Clustering; MC-MCL: Minimum Curvilinear Markov Clustering; PSI-ROC: Projection Separability Index measured by Area Under the Curve; PSI-PR: Projection Separability Index measured by Area Under the Precision Recall; Trust: Trustworthiness.

Table 2. Ranked performance of unsupervised dimension reduction techniques on the real datasets. The table shows the ranked performance of unsupervised dimension reduction techniques according to the PSI indices for sample separation (PSI-ROC and PSI-PR) in the space of the first two dimensions of embedding, for the three studied datasets (Paroni Sterbini, Amir3 and Amir4). Each rank is related to the results obtained in Table 1, top panel. The results are ordered by the mean performance (fourth column) from the best (top) to the worst (bottom)

method.

PSI-ROC					PSI-PR				
Method	Paroni Sterbini	Amir3	Amir4	mean	Method	Paroni Sterbini	Amir3	Amir4	mean
HD	2	2	1	1.67	HD	2	2	1	1.67
MCE	1	4	2	2.33	MCE	1	5	2	2.67
MDSwUF	5	1	3	3.00	MDSwUF	5	1	3	3.00
PCA	3	3	4	3.33	PCA	3	3	5	3.67
nMDS	3	6	5	4.67	MDStyc	5	3	5	4.33
MDStyc	5	4	5	4.67	MDSbc	7	5	3	5.00
MDSbc	7	6	5	6.00	nMDS	4	7	7	6.00

Abbreviations: HD: High Dimension; MCE: Minimum Curvilinear Embedding; MDSbc: Multidimensional Scaling with Bray-Curtis dissimilarity; MDSwUF: Multidimensional Scaling with weighted UniFrac distance; NMDS: Non-metric Multidimensional Scaling; MDStyc: Multidimensional Scaling with Theta-YC distance; PCA: Principal Component Analysis; PSI-ROC: Projection Separability Index measured by Area Under the Curve; PSI-PR: Projection Separability Index measured by Area Under the Precision Recall.

Minor Concerns:

#6 The entire section which discusses the computing platforms used in this study belongs in the Methods section.

REPLY: Maybe the Reviewer did not notice, but the discussion of the computing platforms used in this study was already present in the previous manuscript in the Methods as the last sub-heading of the section.

#7 Line 239: Typically, the standard abbreviation for features within datasets associated with the microbiome is p, instead of m.

REPLY: Thanks to the Reviewer for this concern

ACTION: We have adjusted the abbreviation for the features to the standard p.

#8 Flora generally refers to photosynthetic life, a more appropriate term instead of microflora is simply microbiota.

REPLY: Thanks to the Reviewer for pointing out the inappropriate usage of the term flora.

ACTION: We have changed it to microbiota through the article.

Reviewers' Comments:

Reviewer #1:

Remarks to the Author:

The paper proposes a novel, interesting and important pipeline for analyzing metagenomic data. In particular, it introduces non-linear approaches for dimensionality reduction and multivariate pattern analysis with differential network analysis that reveals mechanisms of re-organizations of combinatorial microbial variations induced by a medical treatment or by an infection. As it is crucial to address non-linearity of systems-level data in biology and medicine, the findings can be of importance for design new therapies for infection, a timely and important topic.

This is a very good and well written manuscript that flows very well and is enjoyable to read. Even the complex computational methods are well written in a way accessible to wide scientific audience.

Two minor suggestions for improvement are:

1. section "PC-corr network" could be expanded and written to the same level of detail and intuition as explanations of previous methods. Perhaps moving the sentence on lines 785-6 to that section and expanding a bit would address the point.
2. Rewrite the Abstract to explain the importance and the main methods a bit better. Perhaps using "PPIs" to mean Proton Pump Inhibitors is not the best choice, as it may be confused with protein-protein interactions, that's commonly associated with "PPIs".

Reviewer #2:

Remarks to the Author:

The key message in the study is that nonlinear approaches may be better than linear method to address complex, biological questions, especially to generate new insights about microbial community composition. There is a rich account in the literature (particularly in plant and animal community ecology, but also in microbial ecology over the last 10 years), which already shows this fact and provides several ways to model multimodal response data as a function of environmental, spatial or temporal gradients. For instance, unsupervised approach could be done with correspondence analysis, and a supervised approach via canonical correspondence analysis to group samples and assess their ecological meaning given metadata is available to perform constrained analyses.

Introduction. The authors ask whether "linear techniques are sufficient to bring out patterns in complex microbial data?" Line 142. The question and the approach should be reformulated, as the choice of statistical tools should be driven by a given experimental design, distribution type of data generated, and hypotheses to be tested, not the other way around (i.e. the performance of statistical tools cannot be assessed on a dataset without considering all those points; otherwise, the approach is purely exploratory and not hypothesis-driven).

If the hypothesis is a linear response of the species or community data to environmental gradients, linear models are the right approach conceptually.

Overall approach. The authors analyse data and find interesting patterns. As significant or biologically interesting as those patterns may be, there is no proof that those patterns may not have emerged due to chance alone, even if significant P values were found (i.e. not very likely but possible; see previous point). Therefore, there is a need to have a ground truth to evaluate the performance of the algorithms in the study, and it seems so far that there is no simulation or benchmarking of the proposed methods to see what patterns the methods can or cannot retrieve in simulated datasets with known structure. The authors used the Tripartite-Swiss-Roll dataset, but how (dis)similar is this dataset to the typical microbial community data structure? It is not

because the obtained clusters match with external (clinical) data that this is a definite proof (but only supporting arguments) that the patterns are indeed real or biologically meaningful.

Line 236. Unsupervised analyses are "particularly useful to discover the presence of interesting sub-groups inside the studied cohort or to detect the influence of confounding factors." This is important to state here that if groups are identified by unsupervised analyses, those newly defined groups cannot be tested for significance anymore in the same study using other statistical tools. Otherwise, the procedure is akin to fishing for significance (i.e. you first identify a pattern in the data and then test for significance of the pattern using the same data). The null hypothesis is then more likely to be rejected. The hypothesis of the existence of those groups should be confirmed in a new experiment, independent from the one used for defining the patterns.

Table S5. Lines 557-558, 712. It seems that the authors compared P values between different approaches, as a tool to compare better performing approaches. Yet, P values are not measure of support for their respective hypotheses (e.g. Schervish 1996 *Am Stat*). Instead of comparing p-values one should compare effect sizes. Therefore, P values should not be used beyond the simple reject/do not reject decision. See also R.J. Grissom & J.J. Kim (2012): "Effect Sizes for Research", Routledge, Taylor & Francis Group, New York, London.

PC-corr. The authors developed MC-MCL method to deal with non-linear patterns in community data, but then use a PC-corr network, which is (Line 482) "a method for linear multivariate-discriminative correlation network reverse engineering". Why not also using non-linear methods to identify features (taxa) that make the difference between the patterns or groups? Would other already existing methods to identify taxa responsible for group differences be also needed to be compared to here?

Line 581. Did the authors check whether PCA axes beyond the first two axes could also indicate treatment-related structure in the data?

Reviewer #3:

Remarks to the Author:

Ciucci et al. compared different linear and non-linear dimensionality reduction methods on multiple microbiome datasets, followed by PCA-based correlation network analysis to clarify the effect of PPI-treatment. Though advanced methodologies are always welcome to analyze the highly dimensional, complex microbial data, this study is rather technical. The approaches used were not really novel and the test data sets were very small. This makes the biological interpretation less convincing. Thus, in terms of its technical aspects and limited biological advance, I think it does not fit to the general readers of this journal.

Major comments:

The authors applied different dimensionality reduction methods, including linear approaches (PCA, MDSbc) and nonlinear approaches (MDSwUF, NMDS, MCE). The first major conclusion is that MCE outperforms other methods and linear technique will fail to bring out the patterns in the microbial datasets related to PPI-treatment. Firstly, based on Fig 3 and Fig 4, none of these methods separated PPI-treated and nontreated groups, even for MCE. The presence of *H. pylori* drives the difference of the microbial community, instead of PPI treatment. This is also confirmed by the PCA plot (Fig3A) that a couple of PPI+ samples (with *H. pylori*) are closer to HP+ samples, while PPI- (without HP) were more closer to HP-. Notable, there were only 4 PPI+ patients. MCE (Fig 4) indeed yielded a better separation between PPI treated and HP- groups, suggesting the microbial shift towards HP+ like community. The improvement is marginal: AUC increased from 0.92 (PCA, MDSuWF and NMDS) to 0.967 (MCE) and P value decreased from 0.009 to 0.004. This could due to the small sample size. Moreover, MCE used different distance matrix, combined with polygenetic distance. Thus it is not conclusive that the observed improvement is due to dimensionality

reduction algorithm or due to different distance calculation. Authors should use the same distance matrix to compare the performance of different methods.

Moreover, there are many other well-known non-linear dimensionality reduction methods, such as t-SNE. How does t-SNE perform?

Nevertheless, the authors had concluded that non-linear based MCE is better. It puzzles me that authors then chose to use PCA-based correlation methods for network analysis, which is linear-based method. It is not clearly described how this correlation network reflects the microbial changes due to PPI treatment.

Other specific comments:

The method section is way too long and not in a good order. Line 227: The part of "Data exploration and visualization" should be moved down, together with the paragraph of PAA, MDS and LDA (Line 270).

Line 387: the polygenetic information is instead directly inferred from bacterial abundance, differently from MDSwUF. I don't understand this. How polygenetic information was calculated from bacterial abundance? How distance was calculated.

Line 436: put zero to the negative values. How many negative values were generated and were replaced by zero. What is the impact on the analysis?

Labels of PPI+ and PPI- in the figures are very confusing to readers. They seem to refer PPI treated (PPI+) and non-treated group (PPI-). Actually, they refer to PPI treated group with and without *H. pylori* respectively, while HP+ and HP- were PPI non-treated group. The authors choose clearer and self-explaining labels.

Reviewer #4:

Remarks to the Author:

Summary:

In the manuscript by Ciucci and colleagues, the authors sought to utilize conventional unsupervised machine learning techniques in conjunction with more nonlinear analysis techniques to understand the responses of the gastric microbiota to specific perturbations. The group created and validated a nonlinear clustering technique combining the weighted graphs of Markov clustering with the network embedding properties of Curvilinear Embedding. The approaches presented here represent a much needed evolution in the standard analyses of microbial community data as intuitively many interactions within these consortia will likely not follow linear relationships. Generally speaking, their arguments are well justified and supported, however several areas of improvement are present. The authors spend large amounts of text on simply demonstrative examples of principles that do not contribute to the central theme of validating their MC-MCL approach and presenting novel findings about the structure of the gastric microbiota during perturbation. The authors also incorrectly use certain vocabulary that is common within the microbiome field. This usage could lead to incorrect assumptions about the biology they are capturing in their models, or an incomplete understanding of the biological assumptions inherent to some of the computational techniques employed here. These and other issues are discussed in more detail below.

Major Concerns:

Metagenomes cannot be obtained from 16S rRNA gene amplicon sequencing alone. The term metagenome refers to the collective gene content of all organisms found within the environment of

interest. This vocabulary must be corrected as true metagenomic studies require more robust sequence curation and analysis pipelines for handling assembly/annotation for large amounts of shotgun read data. Failing to make this distinction is dishonest to the form and interpretation of analyses performed in this study, and will mislead readers who are more familiar with these terms.

The authors spend a great deal of time on demonstrating where older approaches fail (PCA, MDS, LDA) with test datasets, and I wonder why they just didn't show that they fail on datasets where MC-MCL succeeds toward the second half of the results. Additionally, I understand the value of introducing the Swiss-Roll dataset as a lesson for readers, but it seems that the subsequent figures that compare both types of approaches more than demonstrates the point the authors are trying to make. Overall, shortening the results section by eliminating (or greatly reducing) the primarily proof-of-concept examples may benefit the general readability of the final manuscript.

It is confusing why they would introduce the PC-corr method at all in the current study, as it seems they should be focusing energy on validating their new approach. This appears to be something that should be an entire separate future study and removed from the current text entirely.

Line 360: Pearson correlation assumes constant variance and incorrect usage may result in overcalling correlation results. I would think that Spearman would be more appropriate here.

Line 372: Weighted Unifrac distance, while used in many studies, incorrectly assumes phylogenetic relatedness of bacteria simply by similarity of the current section of the 16S gene being analyzed. This disregards the possibility of convergent evolution of sequences, and thus fails more frequently the less well-characterized the environment of interest is. A more accurate, but still Euclidean, algorithm would be Theta-YC.

Yue & Clayton. (2005). A Similarity Measure Based on Species Proportions. *Communication in Statistics- Theory and Methods* 34(11):2123-2131.

Minor Concerns:

The entire section which discusses the computing platforms used in this study belongs in the Methods section.

Line 239: Typically, the standard abbreviation for features within datasets associated with the microbiome is p , instead of m .

Flora generally refers to photosynthetic life, a more appropriate term instead of microflora is simply microbiota.